

# The transport history of African biomass burning aerosols arriving in the remote Southeast Atlantic marine boundary layer and their impacts on cloud properties

Huihui Wu[1], Fanny Peers[2,a], Jonathan W Taylor[1,b], Chenjie Yu[1,c], Steven J Abel[3], Paul A. Barrett[3], Jamie Trembath[4], Keith Bower[1], Jim M. Haywood[2,3] and Hugh Coe[1]

[1]Department of Earth and Environmental Sciences, University of Manchester, Manchester, UK
[2]College of Mathematics, Engineering and Physical Science, University of Exeter, UK
[3]Met Office, Fitzroy Road, Exeter, EX1 3PB, UK
[4]Facility for Airborne Atmospheric Measurements (FAAM), FAAM, Cranfield, UK

[a]Now at: Laboratoire d'Optique Atmosphérique, Université de Lille, Villeneuve-d'Ascq, France
[b]Now at: Research IT, University of Manchester, Manchester, UK
[c]Now at: Université Paris Cité and Univ Paris Est Créteil, CNRS, LISA, F-75013 Paris, France

*Correspondence to*: Hugh Coe (hugh.coe@manchester.ac.uk)

**Abstract.** The transport of African biomass burning aerosols (BBAs) and their impacts on cloud formation and properties over the Southeast Atlantic (SEA) remain one of the largest sources of uncertainty in understanding climate effects across this region. In this study, vertical structures of thermodynamics, aerosol properties and cloud microphysics were characterized around Ascension Island during the CLARIFY-2017 (CLoud-Aerosol-Radiation Interactions and Forcing for Year 2017; August–September 2017) aircraft campaign, providing insights into the relationship between transported African BBAs and clouds in the marine boundary layer (MBL) over the remote SEA. The biomass burning (BB)-impacted MBL exhibited substantially enhanced aerosol number concentration ($N_a$, 0.1 – 3 µm) compared to the clean MBL around Ascension Island, leading to generally increased cloud droplet number concentration ($N_d$) but smaller cloud effective radius ($R_e$) in BB-impacted clouds compared to clean clouds. The cloud-layer mean $N_d$ values were observed to be strongly correlated with aerosols below the cloud (sub-$N_a$) but were more weakly associated with aerosols immediately above the cloud. The increase in the sub-$N_a$ is caused by entrained BBAs from above-cloud to below-cloud regions along long-range transport pathways and/or at the place of observation. We also explored possible simplifications to establish relationships between $N_d$ and sub-$N_a$ or $R_e$ from in-situ measurements. Similar droplet activation fractions were observed in the clean and moderately BB-impacted (sub-$N_a$ < 700 cm$^{-3}$) clouds, while a greater variability was noted in more polluted clouds. The relationship between $N_d$ and $R_e$ remained consistent regardless of the levels of BB influence. Backward simulations were conducted using UK Met Office's Numerical Atmospheric Modelling Environment (NAME), to track the sources and pathways of air parcels reaching Ascension Island. NAME results indicate that air parcels arriving in the Ascension Island MBL can originate from both the boundary layer (BL) and free troposphere (FT) during long-range transport, and entrainment mixing from the FT into the MBL over the SEA is likely to occur. BB pollution in the Ascension Island MBL could occur, when FT air parcels, primarily originating from the





African continent (20°S – 0°N), carry BB smoke. By coupling NAME simulations with SEVIRI (Spinning Enhanced Visible and Infrared Imager) retrievals (aerosol and cloud fields) along the simulated transport path, the study suggests that efficient entrainment of African air parcels from the FT into the MBL occurs multiple days over the SEA before reaching the Ascension Island MBL, mainly in the region to the west of 0°E for examined cases. This study provides important aerosol and cloud parameterizations for climate models, and also provides observational constraints for evaluating the effects of transported BBAs on clouds and their subsequent radiative forcing over the SEA. Furthermore, the identified BBAs entrainment region

may provide additional constraints for refining the vertical transport processes of African BBAs in models, thereby improving the representation of their vertical structures over the remote SEA.

## 1 Introduction

Clouds cover roughly two-thirds of the Earth's surface on an annual basis. They can modify Earth's top of the atmosphere

(TOA) radiation budget by reflecting incoming shortwave radiation, thereby enhancing the planetary albedo, and by contributing to the greenhouse effect (Loeb et al., 2009). The dominant cloud type by area globally is stratocumulus cloud (Sc), which has been extensively studied (Wood, 2012). These clouds are climatically important, as they reflect a significant amount of solar radiation and exert only a small radiative effect in the longwave, leading to a net cooling effect. Previous simulations have reported that even small variations in their microphysical properties can significantly impact Earth's energy

balance, subsequently affect large-scale atmospheric teleconnections, underscoring the crucial role of these clouds in Earth's radiation budget and climate (Jones et al., 2009). The southeast Atlantic (SEA) Ocean region is home to one of the largest Sc decks in the world, due to the low sea-surface temperatures and high low-tropospheric stability resulting from the prevailing subsidence (Seager et al., 2003). A particular interest of the semipermanent Sc decks over the SEA is their interplay with seasonal biomass burning aerosols (BBAs) that are emitted from African wildfires (Painemal et al., 2014), with the prevalence

and severity of African wildfire events expected to increase in coming years (Fuzzi et al., 2015). Every year from July to October, wildfires across central and southern Africa account for about one-third of global carbon emissions from biomass burning (BB) (Roberts et al., 2009). The elevated smoke is frequently transported westward over the SEA by the easterly jet which is associated with the northern branch of the deep anticyclone over southern Africa (Adebiyi and Zuidema, 2016). Previous observations and model studies also indicate that the subsiding smoke layer in the free troposphere (FT) may entrain

into the Sc cloud top during its transport from land over the ocean (Painemal et al., 2014; Das et al., 2017). Consequently, the general transport pattern of BB smoke from central and southern Africa wildfires may lead to BBAs being present both above and within the marine boundary layer (MBL) over the SEA.

A distinctive feature of these BBAs is their strong absorption of solar radiation over the SEA (Zuidema et al., 2018; Wu et al., 2020), modifying the radiative balance differently compared to non-absorbing particles such as sulfate and sea salt in

marine environments. During the fire season (July−September), transported African BBAs have been reported to induce a strong warming direct radiative effect (DRE) over the SEA (monthly mean = ~15−35 W m$^{-2}$) (De Graaf et al., 2014). The



thermodynamic changes induced by BBAs heating can further provoke cloud adjustment (known as the semi-direct effect), leading to an increase or decrease in cloud coverage depending on the relative vertical locations of the BB layers and the cloud deck (Boucher et al., 2013). In the smoky MBL, BBA heating tends to reduce the sub-cloud relative humidity (RH) and liquid water content, thereby decreasing the cloud cover (Zhang and Zuidema, 2019). When BBAs are located above clouds, their heating strengthens the cloud-top temperature inversion and stabilizes the low atmosphere. This stabilization will weaken the entrainment process, inducing the moistening of the MBL and the preservation of cloud cover over the SEA (Deaconu et al., 2019; Herbert et al., 2020). Previous simulations presented an overall negative semi-direct effect of BBAs over the SEA during the fire season, counteracting their DRE (Wilcox et al., 2012). The indirect effect of BBAs on MBL clouds over the SEA, by serving as cloud condensation nuclei (CCNs), has received less attention than their DRE and semi-direct effects. However, recent observational studies have demonstrated that the SEA BBAs can be in contact with the underlying clouds (Painemal et al., 2014; Gupta et al., 2021), contrary to previous assumptions that they are well separated in this region. Recent simulations show that the indirect effect of BBAs over the SEA can significantly enhance the cloud brightness over the SEA and cause a substantial cooling, by reducing cloud droplet size (Twomey effect) and affecting cloud lifetime (Gordon et al., 2018; Lu et al., 2018). The indirect effect of BBAs over the SEA is suggested to play a dominant role in determining the total radiative forcing at the TOA, compared with their DRE and semi-direct effects (Lu et al., 2018). Therefore, realistically representing the response of clouds to BBAs is vital for accurate model assessments of BBA-cloud-radiation interactions over the SEA.

Satellite-based observations have been employed in this region to characterize aerosol and cloud spatial and temporal variations, and to statistically evaluate cloud responses to BBAs (Costantino and Bréon, 2010, 2013; Painemal et al., 2014). Passive sensor data (i.e. aerosol and cloud properties) from A-Train satellites coupled with CALIOP (Cloud-Aerosol Lidar with Orthogonal Polarization) estimates of aerosol and cloud altitudes from CALIPSO (Cloud-Aerosol Lidar and Infrared Pathfinder Satellite Observation) satellite, are commonly used. These satellite-based observations have been used to examine the vertical distance between the smoke layer base and the cloud top, which plays a vital role in the impact of BB layers on Sc radiative properties. When cloud and aerosol layers nearly overlap, empirical relationships between $N_d$ or $R_e$ and CCN (or aerosol number concentration) were estimated to quantify the cloud response to BBAs. In contrast, $N_d$ or $R_e$ and aerosols are generally uncorrelated when smoke and clouds are vertically well separated. However, CALIOP tends to overestimate the base of the smoke layer and imply an unrealistic separation between BBAs and underlying clouds, since absorbing smoke particles attenuate the lidar beam and thereby CALIOP is not able to detect the lower portions of BBA layers (Rajapakshe et al., 2017). Satellites are also unable to resolve the exact amounts of BBAs activated in the clouds, which directly impacts the magnitude of the cloud microphysical effects. More accurate in-situ measurements of aerosol-cloud relationship are needed to assess the effects of BBAs on Sc over this region.

Another important uncertainty in simulating radiative effects of BBAs over the SEA is the vertical distribution of transported BBAs. Recent models tend to show BBAs layers descending rapidly when off the western coast of the African continent, resulting in BBAs layers that are too low in altitude over the SEA (Das et al., 2017; Gordon et al., 2018). The levels to which the aerosol plumes subside, and the steepness of this descent vary amongst the models. The underestimated smoke



layer height in models would lead to an underestimation of above-cloud aerosols and an overestimation of MBL aerosol loadings (Shinozuka et al., 2020) on the eastern side of the SEA cloud deck and the converse on the western side. The misrepresentation of aerosol vertical distribution would cause a significant diversity in simulated climate forcing over the SEA, as the radiative effects of aerosols are dependent on their vertical distribution and locations with respect to clouds (Samset et al., 2013). This unrealistically rapid descent of BBA layers may be attributed to an overestimated subsidence in the model-simulated large-scale vertical velocities. Recent studies suggest that most climate models underpredict BBAs absorption over the SEA, likely causing the exaggerated mixing of BBAs into the MBL (Mallet et al., 2021). This is because models that underestimate aerosol absorption may fail to capture a "self-lofting" process, where heating induced by the absorbing BBAs can elevate the associated plume layer to higher altitudes (Johnson et al., 2023). Consequently, the entrainment process of elevated FT smoke into the MBL is poorly represented in most models, despite its importance in determining the MBL smoke layer in transport regions over the SEA. To reliably simulate the vertical distributions of BBAs, it is necessary to determine the efficient entrainment region and improve the representation of aerosol entrainment processes. The resolution of models is also assumed to affect the simulation of vertical processes (Das et al., 2017). A test of high-resolution meteorological fields in models, supporting the resolution of small-scale convection transport, is needed to study the plume transport history.

To address the aforementioned issues, aircraft in-situ measurements are essential to provide unique constraints on the vertical distribution of long-range transported African BBAs for climate models over the SEA. Aircraft observations with continuous vertical sampling are also the most reliable source for accurately characterizing the correlations between aerosols and clouds in real time. Recent aircraft observations such as the ORACLES (ObseRvations of Aerosols above CLouds and their intEractionS; September 2016, August to September 2017, and October 2018) and the Aerosol Radiation and Clouds in southern Africa (AEROCLO-Sa) projects were conducted over the SEA, mostly within westward of the Africa continent and eastward of 0° E, which characterized the vertical distributions of African BB plume and clouds, as well as the distinctive MBL cloud responses to BBAs below cloud (Formenti et al., 2019; Redemann et al., 2021). However, aircraft observations over the remote SEA are needed to provide a broader-scale picture of BBA-cloud interactions in this region. In this study, we present updated aircraft observations of clouds and BBAs from the CLARIFY-2017 (CLoudAerosol-Radiation Interactions and Forcing for Year 2017; August–September 2017) campaign, which was based around Ascension Island in the remote SEA (Haywood et al., 2021). We characterize the vertical structures of aerosol properties and cloud microphysics and assess microphysical effects of BBAs on cloud properties over Ascension Island. We also conduct backward simulations using a high-resolution modelling environment, to investigate the sources of air parcels arriving at Ascension Island and the long-range transport history of air parcels from Africa to the remote SEA. We combine air parcel analysis and SEVIRI (Spinning Enhanced Visible and Infrared Imager) satellite observations, to identify the efficient entrainment regions where FT air parcels from Africa are likely to enter the MBL over the SEA and demonstrate their impacts on cloud properties along transport.



## 2 Methodology

The CLARIFY campaign was conducted using the UK FAAM (Facility for Airborne Atmospheric Measurements) BAe-146 Atmospheric Research Aircraft and characterized the aerosol-cloud system centered around Ascension Island (7.96° S, 14.35° W) in the remote SEA. A total of 28 scientific flights (designated flight labels from C028 to C055) took place between 16th August and 7th September 2017. The aircraft was equipped with a range of instruments to measure aerosol and cloud properties, as well as meteorological variables (e.g. temperature (T), potential temperature ($\theta$), RH, total water mixing ratio ($q_t$)). Each flight consisted of a series of straight and level runs (SLRs) at varying altitudes and vertical profiles to characterize aerosols. On cloudy days, saw-tooth and stepped profiles were made to characterize cloud microphysics from cloud base to top. A more detailed overview of the project is provided by Haywood et al. (2021). Tracks of flights used in this study are shown in Fig. 1. Transit flights, C040-41 are not included since the aircraft was predominately at high altitudes. Flights with mainly cloud-free samplings or focusing on specific events such as pocket of open cell (POC) (C052-C054) are also not included in this study. The main aerosol and cloud measurements used in this study are described below.

### 2.1 Aerosol measurements

Aerosol size distributions were measured at 1-Hz resolution via two wing-mounted Passive Cavity Aerosol Spectrometer Probes (PCASP1 and PCASP2), which resolved number concentrations in 30 diameter bins between 0.1 and 3 µm. The two PCASPs were size-calibrated using di-ethyl-hexylsebacate (DEHS) and polystyrene latex spheres (PSL) with known size and refractive index (Rosenberg et al., 2012). A refractive index of $1.54 - 0.027i$ was assumed for ambient aerosol to determine the bin sizes by using Mie scattering theory. The refractive index was obtained by the method reported by Peers et al. (2019), where the aerosol model is tuned using the refractive index to best represent the PCASP measurements. Aerosol number concentration in the accumulation mode ($N_a$, 0.1 – 3 µm) was obtained by integrating the PCASP distribution. The lognormal fitted count median diameter (CMD) was also calculated from the PCASP aerosol size distribution. Barrett et al. (2022) reported that the $N_a$ and particle size distributions of submicron aerosols from the two PCASPs on FAAM are comparable. In this study, we used the aerosol size distribution measurements from the PCASP2.

A model 3786-LP water-filled condensation particle counter (CPC) on board can detect aerosol particles larger than 3 nm (Hering et al., 2005) at 1-Hz resolution, and it can provide condensation nuclei number concentration ($CN_3$, > 3 nm) at an accuracy of ±12 %. Cloud Condensation Nuclei (CCN) number concentrations were measured at 1-Hz resolution by a Droplet Measurement Technologies (DMT) dual-column CCN counter (Roberts and Nenes, 2005). The CCN data was analyzed when measured at a supersaturation (*SS*) of ~0.2 %. The relative uncertainty in the supersaturation measurements is ±10 %, and the uncertainty associated with the flow calibration and counting efficiency of the particle counter is typically ~6 % (Trembath, 2013). The aerosol measurements reported here were corrected to standard temperature and pressure (STP, 273.15 K and 1013.25 hPa). Note that the PCASP, CPC and CCN measurements inside clouds can be unreliable and thus in-cloud (criteria see Sect. 2.2) data was removed. We also obtained the above-cloud aerosol optical thickness (AOT) across the SEA region



(20° W − 15° W; 30° S − 0° N), which was retrieved from the SEVIRI sensor aboard the geostationary Meteosat-10 satellite. The retrieval methods and corrections are described in Peers et al. (2019, 2021). The available SEVIRI aerosol field retrievals are from 6:12 to 17:57 (UTC), with a time resolution of 15 min.

## 2.2 Cloud measurements

Measurements of the cloud droplet number size distribution were made by a Cloud Droplet Probe (CDP) at 1-Hz resolution, with the operation and calibration of the CDP described in Lance et al. (2010). In brief, the CDP is a forward-scattering optical particle counter which can detect particles in 30 diameter bins between 2 and 52 μm. When a cloud droplet passes through the laser beam, the forward scattered light is collected over a 1.7 to 14° solid angle. Then, the light is equally distributed by an optical beam splitter, where one beam is sampled by a qualifier photodetector which recognizes a countable particle, and the other by the size detector. A 10-point glass bead calibration spanning the instrument's detection range was performed before each flight day. The nominal bead size was corrected for the differences between the refractive indices of glass and water, and the water-corrected size was used to calibrate the instrument's sizing response (Barrett et al., 2021).

The CDP measurements were corrected to STP (273.15 K and 1013.25 hPa). We calculated $N_d$, $R_e$, and liquid water content (LWC) from the CDP's cloud droplet spectrum as follows:

$$N_d = \int n(r) \, dr \approx \sum_1^m n(r_i)$$

$$R_e = \frac{\int r^3 \, n(r) \, dr}{\int r^2 \, n(r) \, dr} \approx \frac{\sum_1^m r^3 \, n(r_i)}{\sum_1^m r^2 \, n(r_i)}$$

$$LWC = \frac{4\pi}{3} \, \rho_{water} \int r^3 \, n(r) \, dr \approx \frac{4\pi}{3} \, \rho_{water} \sum_1^m r^3 \, n(r_i)$$

where $n(r)$ is the number of cloud droplets in a particular size bin, $r_i$ is the middle radius value for each of the size bins, and $\rho_{water}$ is the density of liquid water. An LWC value over 0.01 g m$^{-3}$ for 1 Hz measurements was used to define the low threshold for the presence of cloud. To eliminate the inclusion of optically thinner clouds, a threshold of $N_d > 5$ cm$^{-3}$ and bulk LWC > 0.02 g m$^{-3}$ was used to perform statistical cloud sample analysis. The duration of efficient cloud sampling in each flight using these criteria is listed in Table S1 and the areas are also highlighted in Fig. 1. Of relevance to this work is the saw-tooths and stepped profiles obtained between the sub-cloud and above-cloud regions. A vertical profile that sampled a continuous cloud layer was used to determine the cloud base height ($Z_B$) as the lowest altitude with $N_d > 5$ cm$^{-3}$ and bulk LWC > 0.02 g m$^{-3}$, and the cloud-top height ($Z_T$) was identified as the highest altitude satisfying these criteria. The liquid water path (LWP) was also integrated from each profile sampled.

We also obtained remotely sensed cloud optical thickness (COT), $R_e$ and cloud-top height (COT) across the SEA region (20° W − 15° W; 30° S − 0° N), which was also retrieved from the SEVIRI sensor aboard the geostationary Meteosat-10 satellite, following Peers et al., (2019, 2021). The $N_d$ was calculated assuming an adiabatic-like vertical stratification (Painemal et al., 2012):





$$N_d = 1.4067 \times 10^{-6} \left[ cm^{-\frac{1}{2}} \right] COT^{-\frac{1}{2}} R_e^{-\frac{5}{2}}.$$

Only data from liquid clouds in the MBL (cloud top height less than 3000 m) are included in this analysis. The available SEVIRI cloud field retrievals are from 6:12 to 17:57 (UTC), with a time resolution of 15 min.

## 2.3 NAME description

In this study, we used the UK Met Office Numerical Atmospheric Modelling Environment (NAME) (Jones et al., 2007) to conduct backward air parcel simulations. A certain amount of hypothetical tracer particles (hereafter referred to as "air

parcels") (~$2.7 \times 10^{-7}$ g m$^{-3}$) were released from a $2° \times 2° \times 300$ m box centered around the Ascension Island observation site (14.35°W, 7.96°S, 341 m) and their pathways were tracked backwards for 7-days using the reanalysis products of three-dimensional gridded (3D) meteorological fields derived from the UK Met Office's global Numerical Weather Prediction (NWP) model, the Unified Model (MetUM) (Brown et al., 2012). These fields are updated every 3 hours and have a high resolution of 0.14° longitude by 0.1° latitude. The meteorological fields have 59 vertical levels up to an approximate height of

29 km. NAME was chosen as an appropriate model for this study because it uses high-resolution meteorological data of approximately 10 km × 10 km, it can predict dispersion over distances ranging from a few kilometers to the whole globe, and it has been used successfully in similar NAME studies looking at the air parcel pathways (Panagi et al., 2020). A recent study by Haywood et al (2021) also shows that the high-resolution meteorological fields from the NWP model do reasonably well in representing aerosol vertical profiles over Ascension Island during the CLARIFY period.

In this study, we output instantaneous 3D footprints from NAME every 3 hours during a 7-days backward dispersion simulation. The instantaneous 3D footprints show the air parcels passing through each grid box at specified backward times (3-hourly). The footprint unit is represented by mass concentration (g m$^{-3}$), based on a known quantity of air parcels at release. We analyzed the instantaneous 3D footprints in two ways: 1) The vertical grids of each instantaneous 3D footprint were integrated within each horizontal grid to get the column-integrated horizontal footprints at 3-hourly backward times, and all

the 3-hourly column-integrated horizontal footprints were further integrated during the 7-days simulation to provide the spatial horizontal distribution of original air parcels arriving at Ascension Island over the past 7 days (see results in Sect. 4.1). 2) The horizontal grids of each instantaneous 3D footprint were integrated within each vertical layer, and the vertical distributions of air parcels passing through the horizontal area were calculated at 3-hourly backward times (see results in Sect. 4.1). The 3D meteorological parameters, e.g. temperatures and humidity, were output every 3 hours during each 7-days backward

simulation. The 7-days back-trajectories were also performed using the NAME, releasing a certain amount of hypothetical tracer particles from Ascension Island observation site (14.35°W, 7.96°S, 341 m). The trajectory output is every 15-min and is in the form of air parcel location (latitude, longitude and altitude). The corresponding meteorological fields (e.g. wind fields and temperatures) along the back trajectory were output.

The BL depth from the MetUM is determined to be the height of the surface mixed layer to which BL turbulence extends

(Lock et al., 2020), which is employed in NAME. Over the SEA, stratocumulus clouds play a role in producing a strong




capping inversion, and the presence of a strong capping inversion inhibits turbulent mixing between the cool BL air and warmer and drier overlying FT air. When getting to Ascension Island, the trade wind inversion height is typically higher than the BL depth defined by the MetUM, as the clouds tend to be decoupled from the surface mixed layer, which has been also observed during CLARIFY (e.g. Abel et al., 2020; Haywood et al., 2021). Here, we calculated the inversion height over the SEA, using the outputs of vertical gradients of meteorological fields. The inversion height ($z_i$) is quantified as the height at which the vertical gradient of θ is the largest, and there is also a steep decrease in humidity. Instead of using the BL depth from the MetUM, the calculated inversion height was employed to analyze FT and BL products of NAME outputs separately.

## 3 In-situ observations

During CLARIFY-2017, BB pollutions were observed solely in the MBL, or solely in the FT, or in both the MBL and FT. The classification was introduced in previous CLARIFY studies (Wu et al., 2020; Haywood et al., 2021), which was based on carbon dioxide (CO) and aerosol concentrations. Fig. 2 shows complex vertical structures of $N_a$ (0.1 – 3 μm) measured by the PCASP during CLARIFY-2017. The BB pollution conditions of the 17 flights used in this study are listed in Table S1. Simultaneous field observations on Ascension Island by Zuidema et al. (2018) are also consistent with the dates when the aerosol vertical distributions and MBL pollution conditions changed during CLARIFY. From 16th to 20th Aug, BBAs were observed to exist predominantly in the MBL, and the FT was mainly clean. From 21st to 25th Aug, the MBL changed to be much cleaner, and the BB pollution existed predominantly in the FT. From 26th to 31st Aug, the MBL was BB-impacted again, and the BBAs were observed in both the MBL and FT. Here, we characterized thermodynamic, aerosol and cloud properties under clean and BB-impacted MBL conditions separately.

### 3.1 Vertical structures of thermodynamics, aerosol and cloud

Over the SEA, there is typically a strong thermodynamic inversion at the top of the MBL (e.g. Lock et al., 2000). It is suggested that inversion height ($z_i$) is an important parameter for understanding the BL structure. Here, we estimated the $z_i$ as the height at which the measured vertical gradient of liquid water potential temperature ($θ_l$) is the largest (Zheng et al., 2011), and the $θ_l$ was calculated from the T, θ and water mixing ratios. Fig. 3 shows the vertical profiles of $θ_l$ (Fig. 3a) and $q_t$ (Fig. 3b) for the flights used in this study. The height scale was normalized by the $z_i$, to exclude the variation of $z_i$ and compare the MBL structure on different days. During CLARIFY, we observed similar vertical structures of thermodynamic parameters between flights. The MBL generally showed a surface layer up to the lifting condensation level (LCL), followed by another layer that extends to the base of the inversion. The surface layer was fairly well-mixed with relatively constant $θ_l$ and $q_t$, mostly rising from the sea surface up to an altitude of ~ 300 to 800 m ($z/z_i$ = ~ 0.2 – 0.5). On some days, the upper layer above the LCL was relatively well-mixed, suggesting the presence of a stratocumulus layer that was decoupled from the surface. On some other days, the upper layer was conditionally unstable (mostly happened in the BB-impacted MBL), suggesting the presence of a stratocumulus-over-cumulus MBL or a cumulus-capped MBL. The BL decoupling parameters $α_q$ and $α_θ$, were



also calculated from the respective profiles following the method in Zheng et al. (2011), which represent the relative differences of the $q_t$ and $\theta_l$ between the surface and the upper part of the BL respectively. It is suggested that a value of $\alpha_q$ over ~0.3 indicates a decoupled BL. The average values of $\alpha_q$ and $\alpha_\theta$ in this study were calculated to be $0.35 \pm 0.11$ and $0.32 \pm 0.09$

respectively, and the upper part of the MBL was clearly drier than the lower surface layer, suggesting a generally decoupled MBL over Ascension Island area during CLARIFY. The strong inversion layer occurred at the top of the MBL ($0.9 < z/z_i <$ 1.0), with the average $\theta_l$ increase of approximately 4–9 K across BL top and a sharp decrease in humidity ($q_t$). Above the inversion layer, the air was much drier compared to the MBL and is regarded as being in the FT. FT humidity was higher on days with the presence of FT BB plumes, as the water vapor signal in the FT is reported to have a robust positive correlation

with BB plume strength over the SEA (Pistone et al., 2021).

In this study, the profiles of aerosol properties (CMD, CCN/CN$_3$ and CCN/N$_a$ ratios) were summarized under clean and BB-impacted MBL conditions separately. Wu et al. (2020) also presented the vertical variations of aerosol compositions during CLARIFY. The CMD (Fig. 3c) of aerosols in the BB-impacted MBL was 10 – 15 % lower than in the FT BB plumes, which is probably attributable to some processes occurring in the MBL such as aerosol removal by drizzle (Wu et al., 2020). It is

previously reported by Wu et al. (2020) that the BB-impacted MBL had higher sulfate mass fractions (~ 26%) than the FT BB plumes (~ 11%) around Ascension Island, due to the mixing with marine emissions such as marine sulfate. The corresponding profiles of CCN/CN$_3$ (Fig. 3d) and CCN/N$_a$ (Fig. 3e) ratios under the BB-impacted condition were relatively constant with altitude, suggesting that vertical variations of aerosol size and chemical composition did not apparently affect the BBA ability to become CCN. The CCN activation fractions of highly aged African BBAs in this study are close to transatlantic African

BBAs observed at high level over Amazon area during the dry season (CCN/CN$_{20}$ = $0.83 \pm 0.06$, at $SS$ = 0.5%) (Holanda et al., 2020), but are generally higher than fresher BBAs sampled over the African continent (CCN/N$_a$ = 0.68, at $SS$ = 0.3%) (Ross et al., 2003). In the clean MBL, Wu et al. (2020) presented an average aerosol size distribution, with a CMD of ~30 nm in the Aitken mode and a CMD of ~160 nm in the accumulation mode. The aerosols in the clean MBL were smaller than in the BB-impacted MBL during CLARIFY. We calculated the average ratio of N$_a$/CN$_3$ in the clean MBL to be $0.40 \pm 0.26$, also

suggesting that the majority of particles were in the size below 0.1 μm. The submicron aerosols from marine emissions in the clean MBL were previously reported to be dominated by sulfate (~ 60%) and organic (~ 24%). The average CCN/N$_a$ in the clean MBL ($0.83 \pm 0.20$) was close to the BB-impacted MBL ($0.82 \pm 0.17$), suggesting that the CCN activated fractions of clean and BB-impacted MBL aerosols in the accumulation size range are close although the clean MBL aerosols contained more sulfate with high hygroscopicity than the BB-impacted MBL. The average CCN/CN$_3$ in the clean MBL ($0.38 \pm 0.18$)

was much smaller than in the BB-impacted MBL ($0.76 \pm 0.10$), as the dominant small particles (< 0.1 μm) in the clean MBL are not sufficiently large to act as CCN (Dusek et al., 2006).

Figs. 3f-h show the vertical profiles of cloud properties (N$_d$, LWC and R$_e$), using a height scale normalized by $z_i$. In this study, we focus on sawtooth profiles that contain a continuous cloud layer from the cloud base to the cloud top. In Flight C032 with deep cumulus clouds, stepped profiles and SLRs were carried out and the average vertical profiles of cloud properties are

provided. It should be noted that there were broken cumulus clouds sampled below the continuous stratocumulus cloud layer



on some days, which are not included in this study. In total, there are 31 and 12 sufficient profiles sampled in the BB-impacted and clean MBL respectively. The profile of $N_d$ (Fig. 3f) remains relatively constant, with sometimes an increase near the top. The LWC (Fig. 3g) first increased with height from the cloud base and then decreased near the top of the MBL due to droplet evaporation. The profile of in-cloud $R_e$ ($N_d > 5$ cm$^{-3}$ and bulk LWC $> 0.02$ g m$^{-3}$) generally presented enhanced values from the cloud base to the top (Fig. 3h). Larger ranges of $N_d$ and LWC values, but a smaller range of $R_e$ were found in the BB-impacted MBL clouds than in the clean MBL clouds.

## 3.2 Relationship between aerosol and cloud properties

In this section, we examine the relationship between aerosols and cloud microphysics over Ascension Island in the remote SEA. Here, we use $N_a$ as the metric instead of CCN concentrations to establish the aerosol impacts on cloud microphysics, as $N_a$ is not dependent on cloud supersaturations. Furthermore, the aerosols in the size above 0.1 μm ($N_a$) behaved as the dominant CCN, under both BB-impacted and clean conditions (as shown in Sect. 3.1).

### Aerosol-$N_d$ relationship

Previous studies suggest several mechanisms for the activation of aerosols into cloud droplets that typically occur: 1) via updrafts carrying aerosols to the cloud base (so-called primary activation) or 2) via entrainment through turbulent mixing at the cloud top (so-called secondary activation) (e.g. Hoffmann et al., 2015; Slawinska et al., 2012). Here, we analyzed the cloud-layer mean $N_d$, $N_a$ below the cloud (sub-$N_a$) and above the cloud (above-$N_a$), to examine the relative importance for both activation mechanisms. In each profile with a continuous cloud layer, we averaged the $N_a$ within 200m above the cloud top to obtain the above-$N_a$, and within 200m below the cloud base to obtain the sub-$N_a$. Fig. 4a and 4b show the relationship between cloud-layer mean $N_d$ and $N_a$ concentrations for all the analyzed profiles, in terms of sub-$N_a$ and above-$N_a$. The $N_d$ exhibited a significantly positive correlation with sub-$N_a$, with a Pearson correlation coefficient (r) of 0.89 for all the profiles and a r of 0.77 for BB-impacted MBL profiles. However, the correlation between $N_d$ and above-$N_a$ was relatively weak (r = 0.4 for all the profiles), especially for the flights with clean FT but high $N_a$ in the BB-impacted MBL (dashed black box in Fig. 4b). These results show that the influence of above-$N_a$ on cloud properties was weaker than sub-$N_a$ at the place/time of observation, indicating that primary activation of $N_a$ near cloud base played a greater role as compared to secondary activation of $N_a$ entrained at cloud top. Previous studies similarly reported a higher positive correlation between $N_d$ and below-cloud aerosols than above-cloud aerosols, e.g. over the SEA BB-impacted region near African western offshore (Diamond et al., 2018) and over the Pacific BB-impacted region (Mardi et al., 2019). The disconnectedness between $N_d$ and above-$N_a$ is partly due to the gap between FT aerosol layers and the cloud layer (Mardi et al., 2019). Fig. S1a shows the relationship between sub-$N_a$ and the distance between the bottom of FT BB layers and the cloud top (referred to as Aerosol Base to Cloud Top, AB2CT). The bottom of the FT BB layer was defined as the lowest altitude of the plume where $N_a$ exceeded 500 cm$^{-3}$ (Gupta et al., 2021). A relatively negative correlation was observed between sub-$N_a$ and AB2CT. Separated profiles (AB2CT > 100 m) all occurred within the clean MBL, with a low range of sub-$N_a$ (56 – 125 cm$^{-3}$) (Fig. S1a). The large gap observed on these days separated





the elevated BB layers from the cloud layer and hence weakened the effects of above-$N_a$ on cloud microphysical properties. Fig. S1b shows the relationship between sub-$N_a$ and above-$N_a$ for only contact profiles (AB2CT < 100 m). Some contact

profiles occurred within the clean MBL, but presenting higher sub-$N_a$ (172 – 315 cm$^{-3}$) than separated profiles (56 – 125 cm$^{-3}$) within the clean MBL. Example vertical structures of $N_a$, LWC, and $\theta_l$ in separate and contact profiles within the clean MBL are shown in Fig S2a and Fig. S2b respectively. In the separate profile (Fig. S2a), $N_a$ remained low and relatively constant throughout the entire MBL column (Fig S2a), whereas in the contact profile (Fig. S2b), $N_a$ was slightly elevated below the cloud compared to the surface well-mixed layer. These results suggest that the proximity of FT BB plumes to the cloud top

allows them to impact the sub-cloud $N_a$ budget at the place of observation. Similar results have also been reported in other SEA studies near the western offshore of Africa (Gupta et al., 2021, 2022). The interconnectedness between sub-cloud and above-cloud regions for these contact profiles is attributed to the entrainment process of BBAs from the cloud top to below the cloud at the place of observation, as suggested in previous studies (Diamond et al., 2018). Most contact profiles (34 out of 42) occurred within the BB-impacted MBL, which had a broad range of sub-$N_a$ (212 – 1183 cm$^{-3}$). For these contact profiles

within the BB-impacted MBL, the increase in sub-$N_a$ is not only related to the entrainment process at the place of observation, but also previously entrained African BBAs from above-cloud to below-cloud regions over the SEA during long-range transport (Haywood et al., 2021). It should be noted that the entrainment process is time-dependent and relies on factors such as the entrainment rate and the time duration since the aerosol and cloud layer have come into contact (Diamond et al., 2018), which would prevent an instantaneous correlation and consequently lead to the weak correlation between $N_d$ and above-$N_a$ at

the place/time of observation. It is also important to highlight some observation days that exhibited a clean FT but a BB-impacted MBL (dashed box shown in Figs. 4b and S1b). The increase in amounts of MBL BBAs on these days would be predominantly due to previously entrained African BBAs along the long-range transport. This suggests that previously entrained BBAs during long-range transport are more likely to impact the sub-cloud $N_a$ budget than the entrainment of local FT layers at the place of observation, further resulting in a weak correlation between $N_d$ and above-$N_a$.

During CLARIFY, the BB-impacted MBL had substantially enhanced sub-$N_a$ (212 – 1183 cm$^{-3}$) compared to the clean MBL (56 – 315 cm$^{-3}$). The main contributor to the sub-$N_a$ in the clean MBL was local marine emissions. The increase in the sub-$N_a$ is indicated to be the entrained BBAs from above-cloud to below-cloud regions during long-range transport or/and at the place of observation. More details on the long-range transport and entrainment processes of African BBAs are discussed in Sect. 4. The enhanced sub-$N_a$ in the BB-impacted MBL led to a substantially enhanced range of $N_d$ as compared to the clean

MBL (Fig. 4a), showing the importance of transported BBAs that affect cloud formation in this region. The relationship between $N_d$ and sub-$N_a$ follows a similar pattern in the clean or moderately BB-impacted clouds (sub-$N_a$ < 700 cm$^{-3}$), while a greater variation was observed in more polluted clouds (sub-$N_a$ > 700 cm$^{-3}$). The larger variability in the response of $N_d$ to sub-$N_a$ under more polluted conditions may be partly due to the variability in the MBL updraft velocity. A recent study suggests that in a more polluted MBL regime, defined as the velocity-limited regime, droplet formation is more sensitive to fluctuations

in updraft velocity than to fluctuations in aerosol concentration (Kacarab et al., 2020). When a high concentration of aerosols or CCN is present, there is a strong competition between them for water vapor. The increase in updraft velocity may augment



supersaturation levels, thereby playing a more important role in driving droplet formation than the increase in aerosol concentration under polluted conditions (Kacarab et al., 2020). Additionally, differences in the strength of cloud-top entrainment processes may contribute to observed bias in the response of $N_d$ to sub-$N_a$. Adiabatic LWC (aLWC) was calculated

for each cloud profile, and the ratio of LWC to aLWC (LWC/aLWC) was averaged over the cloud layer to assess the extent of cloud-top entrainment mixing or/and the presence of drizzle (Gupta et al., 2021). In Fig. 4a, some contact profiles (blue dashed box) outline the central relationship between $N_d$ and sub-$N_a$, displaying lower droplet activation fractions than other contact profiles. These contact profiles had average LWC/aLWC values close to 1, suggesting negligible mixing between cloudy and non-cloudy air. Most of the other contact profiles had lower average LWC/aLWC values (0.34 – 0.83), indicating

greater entrainment mixing of aerosols from above-cloud into the cloud layer at the place of observation, which may promote additional droplet nucleation above the cloud base and higher $N_d$ compared to those contact profiles with negligible mixing.

For applications in remote sensing retrievals and climate models, parameterizations of $N_d$ based on $N_a$ have been suggested using various schemes, e.g. linear regression (Hegg et al., 2012) or exponential power law (Twohy et al., 2013) depending on in-situ measurements. We quantified the $N_d$ response to sub-$N_a$ by a frequently used power law relationship ($N_d$

$\sim \alpha N_a^\beta$). The exponent $\beta$ can be thought of as the fractional increase in $N_d$ in response to the fractional increase in $N_a$ (> 0.1 µm) (McComiskey and Feingold, 2008). Data from BB-impacted MBL profiles, analyzed using bootstrap resampling, yielded median values (95% confidence interval) of $\alpha$ = 3.18 (0.56 – 13.44) and $\beta$ = 0.71 (0.42 – 0.92). For clean MBL profiles, the values were $\alpha$ = 2.17 (0.22 – 7.32) and $\beta$ = 0.74 (0.42 – 1.03). It should be noted that the data from clean MBL profiles exhibit small-scale variation, which may limit their reliability in quantifying the power law relationship. When combined, all the data

in this study from clean and BB-impacted MBL profiles yielded $\alpha$ = 1.29 (0.45 – 3.24) and $\beta$ = 0.81 (0.64 – 0.96). The results suggest a small difference in the response of $N_d$ to $N_a$ between BB-impacted and clean MBL profiles, likely due to their similar CCN ability of MBL aerosols under two conditions (as shown in Sect 3.1). Our observations are different to a previous study, which reported higher droplet activation fractions for the cleaner MBL compared to the BB-impacted MBL over the Pacific Ocean (Mardi et al., 2019). Furthermore, the $\beta$ value for BB-impacted MBL cases in this study (0.71 (0.42 – 0.92)) is higher

than that reported for BB-impacted areas off the California coast of North America (0.26 (0.15 – 0.42)) in Mardi et al. (2019). A factor that may explain the differences in droplet activation fraction between the studies is the variability in aerosol properties. The CCN ability of transported African BBAs in this study is broadly higher than the aged BBAs from Western/Northern American wildfires (CCN/CN = 0.11 – 0.62, at $SS$ = 0.2 – 0.5%) in previous work (Pratt et al., 2011; Zheng et al., 2020). The observed differences in the $N_a$-$N_d$ relationship between these studies suggest that the droplet formation

sensitivity to aerosol concentration is varying in different BB-impacted regions, thus different parameterizations of $N_d$ based on $N_a$ should be applied to different BB environments for simulating regional radiative forcing.

### $R_e$ relationship with $N_a$ and $N_d$

The $R_e$ is one of the key variables that determine the radiative properties of liquid water clouds, and the parameterization of $R_e$ in climate models is critical for assessing global radiative forcing. Fig. 5a shows the relationship between cloud-layer



mean $R_e$ and sub-$N_a$ for all the analyzed profiles, and Fig. S3a is for cloud-top-layer $R_e$ and sub-$N_a$. LWP values integrated

from the analyzed cloud profiles were in a larger range for the BB-impacted MBL (11.1 – 161.3 g m$^{-2}$) than the clean MBL

(25.1 – 63.0 g m$^{-2}$). Although the LWP is indicated to be generally higher in the BB-impacted MBL than the clean MBL, the

intrusion of BBAs led to a significantly smaller range of $R_e$ as compared to the clean MBL, consistent with the Twomey effect

(Twomey, 1977). The relationship between $R_e$ and sub-$N_a$ shows a greater variation than that between $N_d$ and sub-$N_a$, likely

due to the influence of additional atmospheric factors such as the MBL thermodynamic structure, cloud depth, cloud-top

entrainment process, etc. Some profiles displayed smaller fractional decrease in $R_e$ in response to the fractional increase in $N_a$

(> 0.1 μm) compared to the other profiles, as shown in Fig. 5a and S3a. These profiles were mainly observed on days with

larger temperature differences between the surface and upper layers of the MBL relative to the other profiles, potentially

supporting stronger updrafts and thereby the development of thicker cloud layers and larger cloud droplets (Wood, 2012). A

generally positive relationship between the cloud-top-layer $R_e$ and cloud-layer depth is shown in Fig. S3b. Cloud-top

entrainment processes may also explain the variability in the response of $R_e$ to sub-$N_a$, since the mixing of warm and dry FT

air into the clouds would reduce LWC and limit condensational growth, thereby constraining the droplet size (Wood, 2012).

Fig. S3c shows the relationship between cloud-layer mean $R_e$ and average LWC/aLWC ratio for all the analyzed profiles.

Cloud layers with negligible entrainment mixing (average LWC/aLWC values, ~ 1) present a generally larger range of $R_e$

compared to those with greater entrainment mixing (lower average LWC/aLWC, < 0.83). Herbert et al. (2020) conducted

large-eddy simulations and reported diurnal cycles of cloud thickness and LWP over the SEA, with both reaching their

maximum at dawn and minimum shortly after midday. Thus, variations in the observation times of the day, which ranged from

9:00 to 16:00 (UTC time) during the campaign, may also contribute to the observed variability in the response of $R_e$ to sub-$N_a$.

Various studies have generated a power law parameterization between $R_e$ and specific cloud water content (the ratio of

LWC/$N_d$) ($R_e \sim a\,(LWC/N_d)^b$), for use in climate models. Fig. 5b shows the relationship between cloud-layer mean $R_e$ versus

the average ratio of LWC/$N_d$ for all analyzed profiles. BB-impacted and clean MBL profiles yielded similar exponent (b)

values of (0.34 ± 0.01) and (0.33 ± 0.01) respectively, which are also close to the exponent (~0.33) validated in previous

aerosol-cloud studies including Amazonia BB areas (e.g. Reid et al., 1999; Lu et al., 2008). A key issue in the use of ($R_e \sim a$

$(LWC/N_d)^b$) is the speciation of the pre-factor "a" (Liu and Daum et al., 2000), which is also similar for the BB-impacted

(77.06 ± 7.53) and clean MBL profiles (69.54 ± 4.95). When combined, all the data in this study from clean and BB-impacted

MBL profiles yielded a = 68.54 ± 3.05 and b = 0.33 ± 0.01. Field observations from this and previous work have tested similar

power law schemes, suggesting that the parametrizations of $R_e \sim a\,(LWC/N_d)$ may exhibit consistent skill for varying degrees

of BB pollution influence.



## 4 Transport history and aerosol-cloud along transport

As discussed in the previous section, CLARIFY measurements (Fig. 2, Table S1) show complex aerosol vertical structures and different MBL conditions with BB-impacted and clean environments. In this section, we modelled 7-days backward dispersion and trajectories using the UK Met Office NAME, to investigate sources of air parcels arriving at Ascension Island

and to examine entrainment history over the SEA. Three case studies were chosen in this study and the release times are marked in Fig. 2: Case 1 released tracers on 18 August 2017 (12:00 UTC); Case 2 released tracers on 21 August 2017 (12:00 UTC), when the MBL transferred from BB-impacted to clean; Case 3 released tracers on 26 August 2017 (12:00 UTC), when the MBL transferred from clean to BB-impacted. We also link NAME simulations to SEVIRI retrievals (cloud and aerosol fields) to investigate the effects of aerosols on cloud properties along the simulated transport.

**4.1 Air parcel history**

Fig. 6a$_1$, a$_2$ and a$_3$ show the integrated backward-dispersion fields over the 7-days simulation for the three cases, representing the horizontal footprint of the air parcels reaching the MBL over Ascension Island area. Using the calculated BL height, we divided the integrated dispersion fields over the whole column into the FT (Fig. 6b$_1$, b$_2$ and b$_3$) and BL (Fig. 6c$_1$, c$_2$ and c$_3$) separately. The dispersion fields in these cases show that BL air parcels (Fig. 6c) were generally transported from the

southeast to northwest over the ocean and arose from a clean source region. In Case 2, which was characterized as a clean MBL over Ascension Island, the dispersion fields (Fig. 6b$_2$) suggest that FT air parcels also mainly arose from clean oceanic regions and hence were free from BB pollution. In Cases 1 and 3 (Fig. 6b$_1$ and 6b$_3$), air parcels originating in the FT were mostly from the African continent (20°S – 0°N), which would be influenced by BB plumes from seasonal wildfires (see Fig. 1, the spatial distribution of MODIS-detected fires for August 2017), and the MBL was observed to be BB-impacted over

Ascension Island. The pollution conditions observed over Ascension Island are consistent with our air parcel analysis using NAME backward simulations for three cases. The backward-dispersion results indicate that air parcels arriving in the MBL over Ascension Island area can originate from both the BL and FT during long-range transport, and BB pollution occurs when FT air parcels are mainly from the African continent (20°S – 0°N). The sources of MBL BB pollution indicated by NAME simulations are consistent with previous studies (Gordon et al., 2018; Zuidema et al., 2018; Haywood et al., 2021).

For BB-impacted MBL cases (Cases 1 and 3), the instantaneous horizontal fields (midday-step footprint in Figs. S4 and S5) show that FT air parcels from the African continent moved westward toward the SEA region. To arrive in the MBL over Ascension Island, there should be African continental air parcels mixing from the FT into the MBL over the SEA. To further investigate this mixing process of FT air parcels from the African continent, we analyzed the vertical distributions of air parcels that were dispersed into the FT region (covering the horizontal area of 20°S – 0°N, 15°W – 12°E) along the backward-

simulation time, for two BB-impacted MBL cases (Fig. 7). The corresponding exchange rates (black dashed lines in Figs. 7a and 7b) and the cumulative exchange amounts (black dashed lines in Figs. 6Sa and 6Sb) of FT air parcels along the backward-simulation time were also calculated, to indicate the mixing conditions of African air parcels from the FT into the MBL over



the SEA. The simulation results suggest that FT air parcels from African sources spanning a wide range of altitudes were generally descending over the SEA before reaching the release area and the MBL was deepening further offshore, which has been also identified in previous work (e.g. Das et al., 2017). FT air parcels can be entrained into the MBL when meeting the BL top. FT air parcels are suggested to be mostly mixed into the MBL in the last ~2 days for Case 1 (Figs. 7a and 6Sa) and in the last ~3 days for Case 3 (Figs. 7b and 6Sb) before arriving at Ascension Island. The mixing rates of FT air parcels into the MBL are suggested to vary between two BB-impacted MBL cases, there is a marked maximum in the exchange rate of FT air parcels for Case 1, whereas Case 3 shows a steadier and continuous mixing. Fig. S7 shows the estimated entrainment rate ($\omega_e$) at $z_i$ along the back trajectory from NAME simulations for Cases 1 and 3, coupled with the air parcel height and $z_i$. The $\omega_e$ was estimated as the sum of the subsidence rate at $z_i$ ($\omega_s$ at $z_i$) and the rate of change of the MBL height (Wood and Bretherton, 2004). The subsidence rate was regarded as the average downward vertical wind at $z_i$. For a given $z_i$ and FT particle concentration, the rate of increase in MBL particle concentration is suggested to be dependent on the $\omega_e$ (Wood and Bretherton, 2004; Diamond et al., 2018). In Fig. S7, when the trajectory height was mainly within or close to $z_i$, the entrainment rates in Case 1 were generally higher than Case 3, which likely accounts for the stronger mixing in Case 1 than Case 3. NAME results from BB-impacted MBL cases suggest that efficient mixing of FT air parcels into the MBL would occur multiple days over the SEA during their westward movement from the African continent to the Ascension Island area. Once FT air parcels that could carry BB plumes from the African continent are efficiently mixed into the MBL, they would be rapidly diluted with cleaner MBL air parcels. Subsequently, they were transported by the MBL's south-easterly winds to arrive at Ascension Island. Combined with the instantaneous fields of BB-impacted MBL cases, the main area where the efficient mixing occurred and the mixed air parcels could be transported to Ascension Island area is suggested to be located west of 0°E in both cases.

## 4.2 SEVIRI aerosol and cloud fields along transport

To confirm FT BB pollution conditions along the simulated transport pathway, we performed a statistical analysis of the SEVIRI-retrieved above-cloud AOT (hereafter referred to as AOT) co-located with NAME horizontal footprints, at 3-hourly backward step. Examples of the co-location relationship between AOT and NAME instantaneous horizontal footprints are shown in Fig. S8. The parts of the scene that are co-located with the contemporaneous horizontal footprints from NAME simulations are highlighted. The air-density-weighted average AOT was calculated for the co-located areas at 3-hourly backward step. Figs. 7c and 7d show the air-density-weighted average AOT along the FT and BL air parcel transport respectively, for three cases. High AOT values suggest that the presence of African BBAs is co-located with air parcels along the simulated transport. For BB-impacted MBL cases (Cases 1 and 3), the AOT co-located with FT air parcels (black lines in Fig. 7c) remained mostly high during the periods of efficient mixing indicated by NAME simulations. This provides evidence for the possibility of BBA entrainment from the FT into the MBL during these efficient mixing periods, if the bottom of BB layers is close to the cloud top. Along BL transport from southeast to northwest over the SEA, the AOT co-located with BL air parcels (black lines in Fig. 7d) was enhanced when BL air parcels moved close to the efficient mixing periods of FT air parcels. This provides further evidence that the entrained BBAs from the FT into the MBL could be subsequently transported



by the low-level south-easterly winds to arrive at the Ascension Island area. In comparison, the AOT in the clean-MBL case (Case 2) was continuously low along the simulated transport (red lines in Fig. 7c and 7d), demonstrating a negligible contribution of polluted air parcels to the MBL over Ascension Island. Based on NAME simulations and AOT along the

simulated transport, the occurrence of BB-impacted MBL over Ascension Island is suggested to be related to two factors. Firstly, African BB plumes aloft need to be transported to the efficient mixing areas identified in this study, which largely depends on the strength of the African easterly jet. Furthermore, the transported BB plumes can be entrained into the MBL in areas of efficient mixing over the SEA, which depends on the altitude of the African easterly jet and the distances between BB layers and the BL top. The significance of the efficient mixing areas identified in this study lies in the potential for the entrained

BBAs in these regions to be transported by MBL south-easterly winds to the vicinity of Ascension Island. A meteorological overview over the SEA during August 2017 was provided in a recent study (Ryoo et al., 2022), which supports this study. The African easterly jet was reported to be mostly at a low FT level (~ 3 km; ~ 700 hPa), but slow-moving and weak in early August. With this, BB plumes aloft were not likely transported as far as Ascension Island, leading to the observed clean FT, such as in Case 1. However, the low-level easterly jet was transporting African BB plumes in the low FT, allowing the

possibility of BBAs entrainment into the MBL during early periods of efficient mixing, examples of which are identified in the NAME simulations. These previously entrained BB plumes could be transported by the MBL south-easterly winds and contribute to the polluted MBL over Ascension Island, such as in Case 1. In mid-August, the low-level African easterly jet was reported to be suppressed, while a strong mid-level easterly jet gradually developed above 600 hPa. This would transport African BB plumes at higher altitudes, implying a disconnection between BB layers and the BL top. The apparent separation

between the bottom of BB layers and the cloud top was also observed over Ascension Island from CLARIFY in-situ measurements, as illustrated in Sect 3.2. This disconnection would suppress the entrainment of African BBAs into the MBL during long-range transport, resulting in a relatively clean MBL over Ascension Island, such as in Case 2. In late August, strong easterly winds at a low FT (700 hPa) were reported to become apparent from 20 August, together with another easterly jet aloft above 600 hPa. The strong low-level easterly winds would allow African BBAs transported to both the FT and the

MBL over Ascension Island, such as in Case 3.

We analyzed average SEVIRI-retrieved $N_d$ and $R_e$ along cloud-runs in each CLARIFY flight, which show relatively good agreement with in-situ measurements from the CDP (Fig. S9). The overall trends of $N_d$ and $R_e$ compare well between SEVIRI-retrievals and in-situ measurements around Ascension Island, both showing enhanced $N_d$ and reduced $R_e$ in the BB-impacted MBL than the clean MBL. There is a worse comparison of $R_e$ on some days, since the CDP measures cloud droplets below 50

μm and larger-size particles out of the CDP detect range may lead to discrepancies, especially for low $N_d$ cases (Abel et al., 2020). Here, we extend the analysis of SEVIRI-retrieved cloud properties ($N_d$ and $R_e$) to their evolution along the simulated BL transport from southeast to northwest over the SEA, before arrival at Ascension Island. Comparisons between BB-impacted and clean MBL cases allow the investigation of the impacts of entrained BBAs on a broad cloud field. Similar to the AOT, examples of the co-location relationships between $N_d$ or $R_e$ and NAME instantaneous horizontal footprints at a given time step

are shown in Fig. S8. The air-density-weighted average $N_d$ and $R_e$ were also calculated for these co-located areas at 3-hourly





backward step. Figs. 8a and 8b show the air-density-weighted average $N_d$ and $R_e$ respectively along the BL air parcel transport for three cases. For the clean MBL case (Case 2), the $N_d$ was consistently low throughout the passage of BL air parcels over the SEA, even reducing before approaching Ascension Island, while the $R_e$ increased substantially. The evolution of clouds over the SEA is related to other meteorological factors such as the evolution of CTH (Painemal et al., 2014). The air-density-

weighted average CTH was calculated along BL air parcels with the backward-simulation time (Fig. 8c), which was increasing in Case 2 (red line in Fig. 8c) from the southeast to northwest over the ocean. As the MBL deepened together with enhanced CTH, condensational growth yields the observed larger droplets (Painemal et al., 2014). The deeper MBL also tends to be decoupled, and the decoupled MBL is likely associated with an increased occurrence of drizzle due to the presence of larger droplets (Jones et al., 2011), which may result in lower $N_d$ when approaching Ascension Island. These processes would

contribute to the increasing $R_e$ and slightly reduced $N_d$ observed along the BL transport for the clean MBL case. For BB-impacted MBL cases (Cases 1 and 3), the $N_d$ was largely enhanced when FT air parcels from the African continent were suggested to efficiently entrain into the MBL during ~2–3 days prior to arrival at Ascension Island. Although the deepened MBL (black lines in Fig. 8c) with accompanying processes of condensational growth should promote increasing $R_e$ (as in the clean MBL case), the $R_e$ was relatively similar along the BL transport for BB-impacted MBL cases. This suggests that the

enhanced $N_d$ from entrained BBAs promoted a reduction in cloud droplet size, which offsets the growth effects from the deepened MBL. Comparisons between BB-impacted and clean MBL cases show that BBA entrainment along the transport path would lead to significantly enhanced $N_d$ but reduced $R_e$, modifying cloud fields in broad efficient mixing areas that persist through the SEA region. This modification of cloud properties, which is co-located with the efficient entrainment areas identified through NAME simulations, further supports the previous analysis of the mixing history of African air parcels. This

modification of cloud properties due to entrained BBAs also has significant implications for the radiative properties and microphysical processes of clouds over the remote SEA once impacted by broad areas of efficient mixing between the FT and MBL when BBA is present in the FT air.

By coupling NAME simulations with SEVIRI retrievals (aerosol and cloud fields) along the simulated transport path, we have been able to identify the efficient entrainment areas where African FT air parcels mix into the MBL before reaching

Ascension Island. Efficient African BBA entrainment from the FT into the MBL may occur multiple days prior to arrival at Ascension Island and mainly in a region of west of ~0°E over the SEA for the examined cases in this study. Previous model studies have highlighted uncertainty in accurately simulating the vertical distribution of transported African BBAs over the SEA and their subsequent climate effects (Das et al., 2017; Shinozuka et al., 2020). These studies emphasized that the transported BBAs are predicted to descend to unrealistically low altitudes as they move westward off the African coast. One

potential explanation for this sudden subsidence over the SEA is the inadequate representation of a "self-lofting" process in models, where absorption-induced heating from BBAs can lift the associated plume layer to higher altitudes (Johnson et al., 2023). The insufficient simulation of this self-lofting process is likely due to the large underestimation of the single scattering albedo (SSA) and subsequent absorption of transported BBAs in most models (Mallet et al., 2021). Another potential explanation for the misrepresentation of vertical transport and hence vertical distribution of BBAs is related to the resolution



of models, as small-scale processes such as turbulence and self-lofting are not explicitly resolved in coarse-resolution models (Das et al., 2017). To improve the simulation of vertical distributions of African BBAs over the remote SEA, it is essential to reduce model biases in aerosol properties (e.g. SSA) and vertical processes. Improving the representation of these processes is complex as reducing bias in one property may lead to greater biases in another. The identified efficient entrainment areas in this study could provide further constraints for testing and refining predictions related to BBA transport processes and vertical

distributions over the SEA. Furthermore, the indicated transport and mixing history from NAME simulations are supported by the evolution of aerosol/cloud fields from SEVIRI retrievals, suggesting that high-resolution meteorological fields from UM models could provide a reasonable simulation of the plume descent, which could be employed in future regional model simulations.

## 5 Conclusions and implications

In this study, we have characterized the vertical structures of aerosol and cloud properties around Ascension Island in the remote SEA during the CLARIFY-2017 aircraft campaign and investigated the response of clouds to observed aerosols within the MBL. We also conducted backward simulations (for both clean and BB-impacted MBL cases) using the UK Met Office's NAME to assess the sources and transport pathways of air parcels before arrival at Ascension Island. NAME simulations allow a discrimination of air parcels in different layers, and this has been used to derive the entrainment history of African air parcels

from the FT into the MBL over the SEA.

    During the campaign, the BB-impacted MBL had substantially enhanced $N_a$ compared to the clean MBL around Ascension Island. This resulted in broader ranges of $N_d$ and LWC values, but a smaller range of $R_e$ within BB-impacted cloud profiles compared to clean cloud profiles. A stronger relationship was observed between the cloud layer-mean $N_d$ and sub-cloud aerosol properties (sub-$N_a$), while the $N_d$ was only weakly associated with smoke immediately above the cloud. This

suggests that the primary activation of sub-cloud $N_a$ was more important in determining $N_d$ values than the secondary activation of $N_a$ entrained at the cloud top at the time of observation. The observed increase in the sub-$N_a$ is a result of entrained BBAs from above-cloud to below-cloud regions along long-range transport or/and the place of observation. We also explored possible simplifications to establish relationships between $N_d$ and sub-$N_a$ or $R_e$ from in-situ measurements. Similar droplet activation patterns ($N_d \sim$ sub-$N_a$) were observed in the clean and moderately BB-impacted (sub-$N_a$ < 700 cm$^{-3}$) clouds, likely due to the

comparable CCN ability of MBL aerosols under the two conditions. However, a greater variability was noted in more BB-polluted clouds (sub-$N_a$ > 700 cm$^{-3}$), possibly influenced by additional atmospheric factors such as the cloud-top entrainment and updraft velocity variations. Comparisons between this study and other BB regions indicate that the droplet formation sensitivity to aerosol concentration is varying across different BB-impacted areas, likely due to the variations in CCN ability of BBAs. Thus, region-specific parameterizations of $N_d$ based on $N_a$ are necessary for accurately simulating regional radiative

forcing. The relationship between $N_d$ and $R_e$ remained consistent regardless of the levels of BB influence. This study extends previous aerosol-cloud interaction field studies to a wider region over the SEA, which provides important aerosol and cloud



parameterizations in climate models for benefiting the assessment of the impacts of transported BBAs on clouds and their subsequent climate effects over the SEA. Furthermore, the disconnectedness between cloud properties and the above-cloud aerosol layer indicates that satellite-based observations of aerosol-cloud interaction in this region may be subject to large uncertainty, as they are unable to resolve the aerosol properties within and below the cloud layer. This underscores the importance of airborne in-situ measurements in this region to provide direct observations of aerosol-cloud interaction.

NAME results show that air parcels arriving at Ascension Island can originate from both the FT and BL, indicating the presence of entrainment mixing from the FT into the MBL over the SEA. For the clean MBL case, both FT and BL air parcels were predominantly from the oceanic region (south of 30°S), transporting air from the southeast to northwest over the SEA before reaching Ascension Island. For BB-impacted MBL cases, FT air parcels were transported from the African continent (20°S – 0°N) which could bring BB smoke, while BL air parcels followed similar pathways as in the clean MBL case. NAME analysis coupled with SEVIRI-retrievals (aerosol and cloud fields) along the simulated transport suggests that efficient entrainment of African BB smoke from the FT into the MBL occurred multiple days before arrival at Ascension Island, which is mainly in the region to the west of 0°E for the examined cases. One significance of the efficient mixing areas identified in this study lies in the potential for the entrained BBAs to be further transported by MBL south-easterly winds to the vicinity of Ascension Island. The analysis of SEVIRI-retrieved cloud fields along the simulated transport path also indicates significant modifications in cloud droplet concentrations and sizes within the broad efficient mixing areas of African BB smoke. The efficient entrainment areas identified in this study can provide an important constraint for refining the transport processes and vertical distributions of African BBAs over the SEA. More case studies should be conducted to validate this methodology and examine entrainment patterns across different months and years, ultimately to characterize a climatological area for entrainment over the SEA.

*Data availability.* Airborne measurements are available from the Centre for Environmental Data Analysis https://catalogue.ceda.ac.uk/uuid/38ab7089781a4560b067dd6c20af3769.

*Author contributions.* H.C. and J.H. designed the research; J.W.T., S.J.A., P.B., J.T. and K.B. performed field experiments; H.W., J.W.T. and J.T. prepared in-situ datasets; F.P. provided the SEVIRI dataset; H.W. and C.Y. performed NAME analysis; H.W. analyzed combined datasets and led the manuscript writing.

*Competing interests.* The authors declare no competing interests.

*Acknowledgements.* This work was mainly supported by the UK Natural Environment Research Council (NERC) Large Grant NE/L013584/1 via project CLARIFY (Cloud-Aerosol-Radiation Interactions and Forcing for Year 2017). The first author has also received funding from Horizon Europe programme under Grant Agreement No. 101137680 via project CERTAINTY (Cloud-aERosol inTeractions & their impActs IN The earth sYstem). The staff of Airtask, Avalon Engineering and FAAM are



thanked for their thoroughly professional work, before, during and after the deployment. The NAME group in the UK Met Office are thanked for their instructions on backward simulations.

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



**Figures**

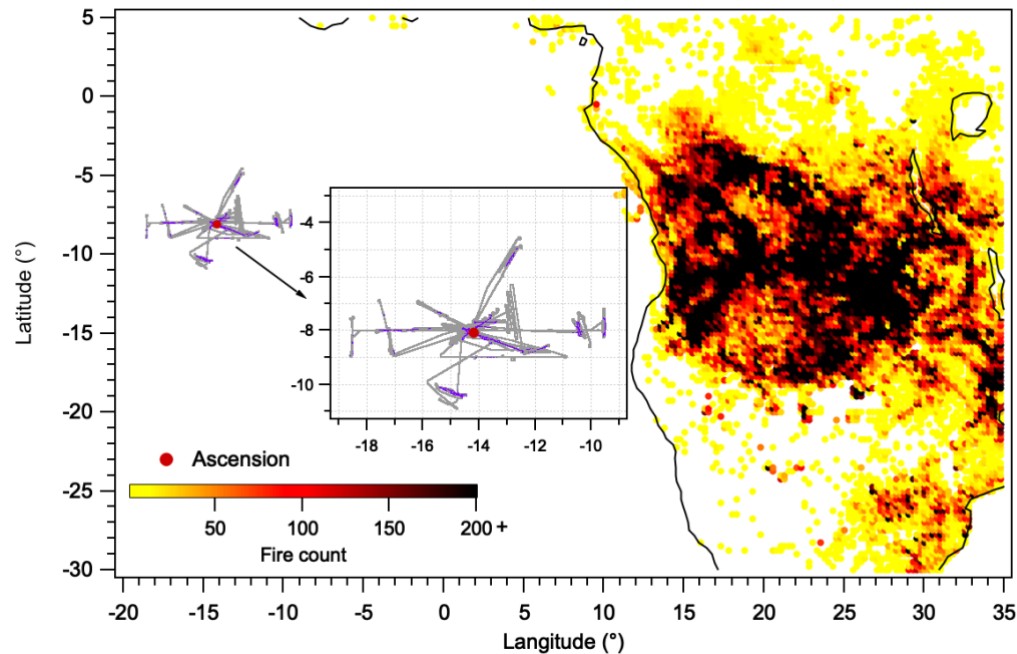

Figure 1: Tracks of the CLARIFY flights used in this study, cloud sampling periods during tracks ($N_d > 5$ cm$^{-3}$ and bulk LWC > 0.02 g m$^{-3}$) are marked in purple colour. The integrated spatial distribution of MODIS-detected fire counts in August 2017 are shown over the African continent.


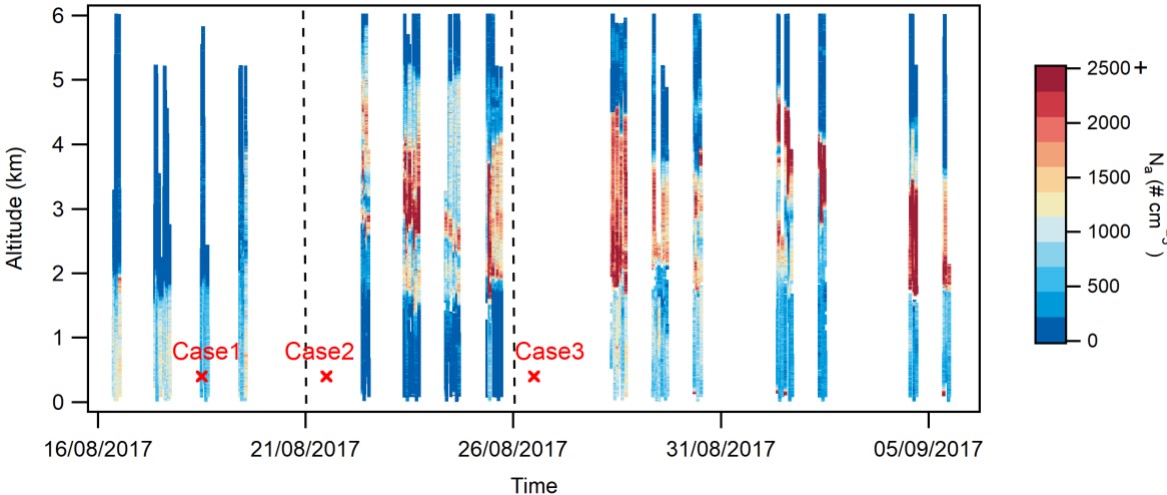

Figure 2: Time series plots showing vertical profiles of aerosol number concentration ($N_a$) derived from the PCASP, for all CLARIFY flights. The red markers represent the release time for the three NAME cases.



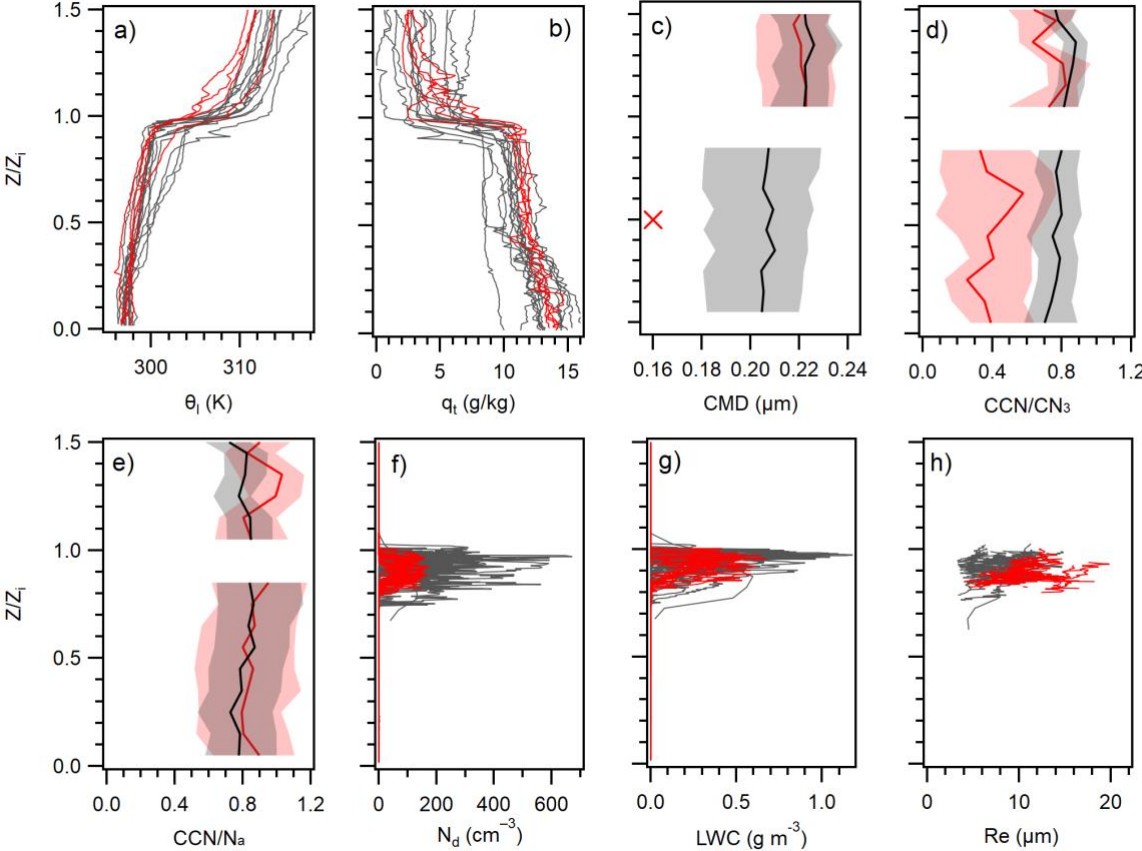

**Figure 3: a, b)** Average vertical profiles of a) liquid water potential temperature ($\theta_l$, K), b) total water mixing ratio ($q_t$) for each flight used in this study. c-e) Summarized profiles of c) aerosol count median diameter derived from the PCASP (CMD, μm) and d,e) the ratio of CCN (~ 0.2%) to condensation nuclei ($D_a > 3$ nm from the CPC) (CCN/CN$_3$) and accumulation aerosol concentration ($D_a > 0.1$ μm from the PCASP) (CCN/$N_a$). Solid lines represent median values, and shades represent 10% and 90% values. The red cross marker in Figure 3c represents the average CMD in accumulation mode from measurements within the clean MBL.  f-h) Vertical profiles of f) liquid water content (LWC, g m$^{-3}$), g) cloud droplet number concentration ($N_d$, # cm$^{-3}$) and h) cloud effective radius ($R_e$, μm) in sampled cloud layers. The y-axis uses a height scale normalized by inversion height ($z_i$). Red represents measurements from flights with the clean MBL, and black represents measurements from flights with the BB-impacted MBL.



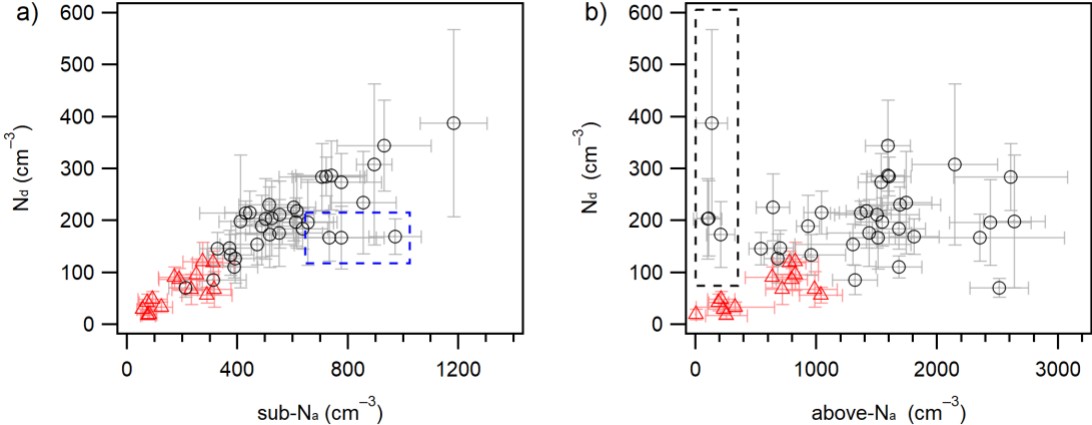

**Figure 4: Relationships between cloud-layer mean $N_d$ and $N_a$ for all profiles, in terms of a) sub-$N_a$ and b) above-$N_a$. The markers and error bars represent the average values and standard deviation for each profile. Black represents the BB-impacted MBL, and red represents the clean MBL. The dashed black box represents the sampled profiles from flights with clean FT but high $N_a$ in the BB-impacted MBL. The blue dashed box represents some contact profiles that outline the central relationship between $N_d$ and sub-$N_a$ and had average LWC/aLWC values close to 1.**

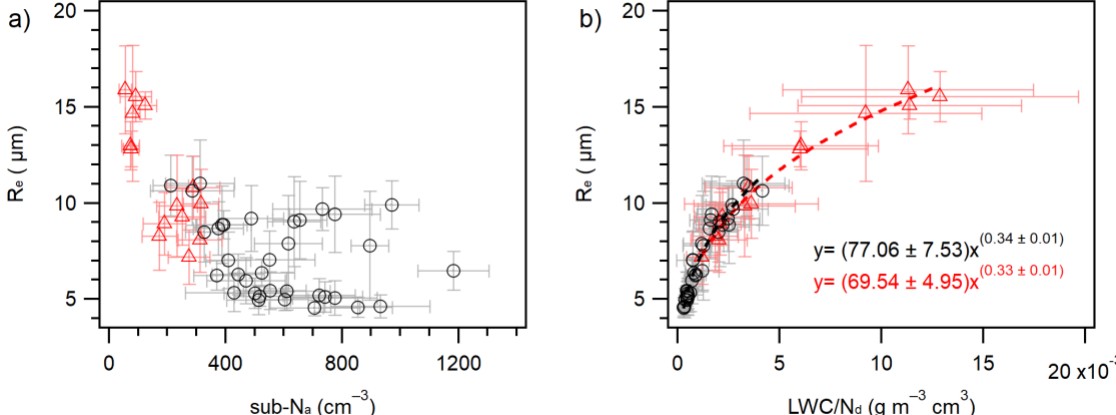

**Figure 5: a) Relationship between cloud-layer mean $R_e$ and sub-$N_a$ for all analyzed profiles. b) Relationship between cloud-layer mean $R_e$ and the average ratio of LWC/$N_d$ for all analyzed profiles. The markers and error bars represent the average values and standard deviation for each profile. Black represents the BB-impacted MBL, and red represents the clean MBL.**



**Figure 6: The 7-days backward-dispersion of three cases from NAME simulations. Top horizontal panels are Case 1 released on 18 August 2017 at 12:00 UTC, middle horizontal panels are Case 2 released on 21 August 2017 at 12:00 UTC, bottom horizontal panels are Case 3 released on 26 August 2017 at 12:00 UTC. Left panels represent the integrated air parcel dispersion results through the whole column. Middle vertical panels represent air parcel dispersion results attributed to FT. Right panels represent air parcel dispersion results attributed to BL. All plots are shown in the same colour scale. The black boxes represent the release locations of Ascension Island. The red boxes in Fig. b₁ and b₃ represent the horizontal grids (covering the area of 20°S – 0°N, 15°W – 12°E) used for integrating within each vertical layer to derive the vertical distribution of dispersed air parcels in Fig. 7.**

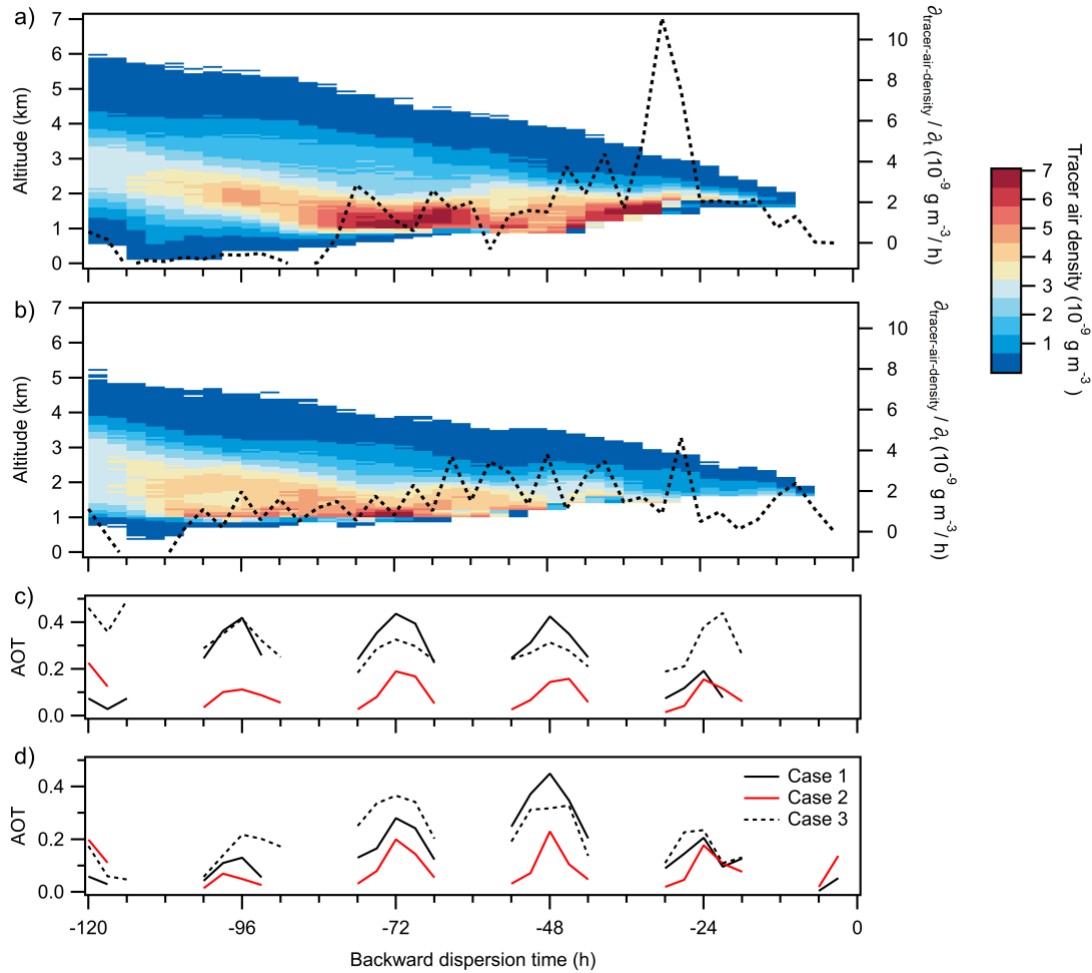

885

**Figure 7: a-b) The 3-hourly time series of the vertical distribution of air parcels dispersed into the FT region identified in the horizontal area of 20°S – 0°N, 15°W – 12°E (see red box in Fig. 6), in terms of backward dispersion time from 3 to 120 h. The right axis (dashed black lines) are the corresponding 3-hourly exchange rates of air parcels dispersed between the FT and MBL, calculated by dividing the increase in dispersed FT air parcels by the dispersion time along backward simulations. a) is for Case 1 and b) is for Case 3. c-d) The air-density-weighted average AOT from SEVIRI co-located and co-temporaneous with c) FT and d) BL air parcels along the backward-simulation time, for three cases.**





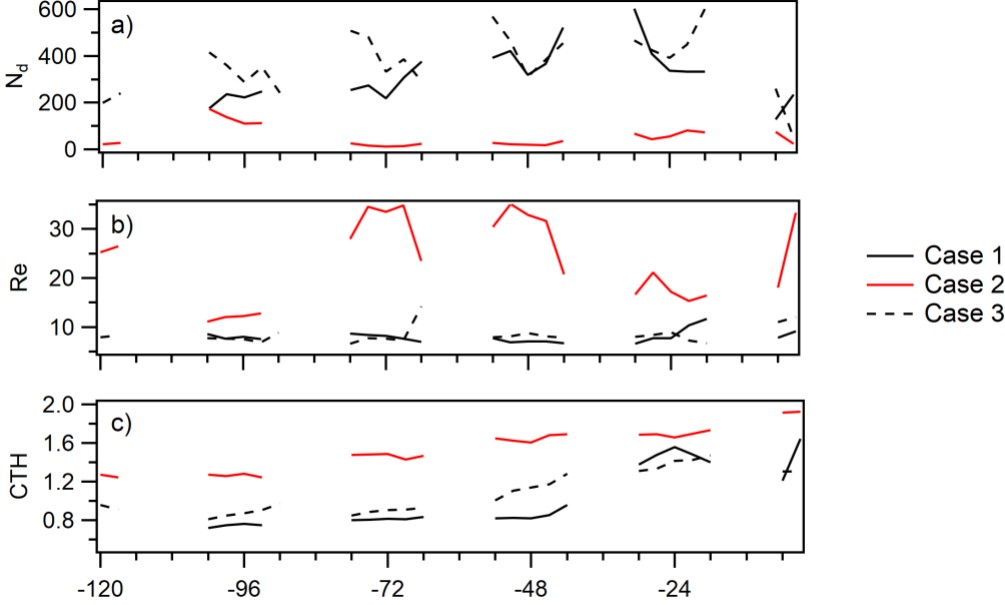

**Figure 8: The air-density-weighted average a) $N_d$, b) $R_e$, c) CTH from SEVIRI co-located and co-temporaneous with BL air parcels along the backward-simulation time, for three cases.**