# Peer review of "The transport history of African biomass burning aerosols arriving in the remote Southeast Atlantic and their impacts on cloud properties"

_EGUsphere, 2024_

## Referee Comment (RC2)

**Review of Wu et al., 2025: The transport history of African biomass burning aerosols arriving in the remote Southeast Atlantic marine boundary layer and their impacts on cloud properties**

This paper combines in-situ data from the CLARIFY-2017 campaign to satellite data and backtrajectory analyses to investigate aerosol-low-level cloud interactions around the Ascencion Island in the remote South-East Atlantic. The authors show that cloud droplet number concentrations are increased, and droplet radii are decreased by biomass burning aerosols, especially if the aerosols are in the sub-cloud layer. This agrees with published literature that highlights the important role of the respective positions of aerosol and cloud layers for ACI in this region, even though the exact functional relationship between aerosol and cloud properties seem to differ compared to other biomass burning regions. The authors then carry out backtrajectory analyses to estimate the exact timing of FT air entrainment into the marine boundary layer for 3 different case studies and identify efficient mixing regions 2-3 days upwind of the Ascencion Island, which could be used to constrain climate models that struggle to accurately simulate ACI in the SEA. I think these results are interesting and can be published after major revisions.

**Major comments:**

The novelty of the paper is not immediately obvious to me, but I believe this could be fixed with some reorganization of its content.
-  Many of the figures in the first half of the paper are reminiscent of the figures in Haywood et al., 2021. Since there are a lot of figures in the supplementary material, maybe it would make sense to remove redundant figures and instead move some of the supplementary material to the main text.
- I also wonder whether the instrument descriptions have previously been published in data papers, in which case it might not be necessary to have such a long method section.
- Since it has been a few years since the CLARIFY 2017 campaign, it might be helpful to summarize for the reader the findings that have already been published using the campaign data, and mention questions that have yet to be answered, including the one(s) this study wants to address, which would make the novelty obvious.
- The paper could be more succinct if the authors removed some redundancy in the explanations. For example, there are long paragraphs dedicated to reviewing the literature in the results section, but the relevance of the cited studies to this study might be more convincing if summarized into 1 or 2 sentences only.
- It might also be helpful to make the abstract more succinct.
- On Ln 540, you mention that your study has important implications for the radiative effects of aerosols in the region, but have you looked at how TOA radiative fluxes from SEVIRI change along the study period, depending on the aerosol load and location ?
- Section 4.2: In my opinion, this section needs to be rewritten.
        - I am concerned about the AOT analyses. Since AOT is a vertically integrated value, the analysis of AOT associated to either MBL or FT air parcels is misleading. For instance, an increase in AOT for the MBL air parcels could simply mean that the MBL is located under a polluted FT, but the MBL itself could still be pristine (and vice versa, for a clean FT over a polluted MBL). Could you address this caveat more explicitly?
        - Ln 490-510 is a very lengthy explanation mixing literature review and some interpretations. Some of the literature review could be moved to the introduction, which would allow the interpretations to stand out more convincingly.

**Minor comments :**

- some choices of words are a bit vague and confusing, for instance. For example, on Ln 28 : « a greater variability was noted in more polluted clouds» is imprecise. Same on Ln 352 (« a greater

variation» ). Do you mean that the linear fit is weaker ? Another example on Ln 295 : « Larger ranges of Nd and LWC values … a smaller range of Re ». Do you mean that Nd and LWC values are larger on average and Re values are smaller on average, or do you mean that there is a larger/smaller standard deviation in the observed distributions (to me, « range » suggests the latter). There are several other occasions where « range » is used in this way in the text (e.g., Ln 404, 572). Could you use more precise mathematical language to help readers better understand your point ?

- Ln 50 : « affecting », not «  ». The last part of the sentence (starting with « underscoring... ») sounds redundant and could probably be removed.

- Ln 120 : remove «  », correct « AfricaN continent »

- Ln 125 : This study is obviously focused on stratocumulus clouds, and the study area is the Ascension island, but traditionally, Sc are thought to form closer to the coast (e.g., areas defined by Klein and Hartmann, 1993), and then transition to cumulus clouds as they move westwards to Ascension Island, which might leave some readers confused. It would be good to address this. Since you actually already have a list of cloud types observed during the campaign in Table S1, it would be good to reference it in the main text.

- Ln 138 : «straight and level runs » not defined

- Ln 142 : why would a POC event not be relevant to this study ?

- Ln 145 : as written above, the instrument description could be shortened if the material is already published somewhere else. What would be interesting to add though is the typical size range of BBAs so that the readers can easily assess the suitability of the measurements.

- Ln 182-185 : the difference between the 0.1 g/m³ LWC and 0.2 g/m³ 'bulk' LWC thresholds was not clear to me

- Ln 190 : (CTO) instead of

- Ln 194 : I believe there is an error in the equation, it should be $COT^{1/2}$ and not $COT^{-1/2}$

- Ln 196 : Did you check the quality of your SEVIRI products close to sunrise and sunset times ? There might be biases around those time, this could impact your results if sunrise/sunset points are colocated to the backtrajectories.

- Ln 197 : The NAME description section got me a bit confused. I did not understand why there seems to be 2 sets of backtrajectories, one at 3-hr resolution, and the second one at 15-min resolution. What is the point of having both sets of trajectories ? why use NAME backtrajectories vs. using the already produced HYSPLIT ones from the Haywood et al., 2021 paper ?

- Ln 230 : here you introduce one definition for the inversion height, but later (Ln 246) you use a different one, why is that ?

- Ln 263 : The FT humidity comparison between clean and BB-impacted FT cannot be inferred from Figure 3, can it ?

- Ln 265 : what is the interpretation for this positive correlation between BBA and humidity in the FT?

- Ln 269 : Could the MBL CMD simply be lower because the BBAs are mixed with smaller aerosols (e.g. sulfate, sea salt, etc.) in the MBL but not in the FT ?

- Ln 279 : $CCN/CN_3$ and not

- Ln 288-291 : as per my comment above, it would be helpful to already include a description of observed cloud types during CLARIFY in the introduction, to contextualize the method and results.

- Ln 292 : near *cloud* top (at the end of the line)

- Ln 321 : what does a « «relatively » negative correlation mean ? Relative to what ?

- Ln 345-346 : could you provide the reference for these numbers again ?

- Ln 347 : maybe « indicated » could be replaced by another verb, like « hypothesized »

- Ln 353 : remove « recent »

- Ln 361 : how were these contact profiles selected ?

- Ln 362 : what does a « central » relationship mean ?

- Ln 365 : just a note that entrainment of warmer and drier FT air could also lead to cloud droplet evaporation and decreases in Nd (cases of extreme inhomogeneous mixing, see Hill et al., 2009)

- Ln 371 : Can you quantify the goodness of fit (for instance with the R2 for the linear regression between logNa and logNd ) ?
- Ln 372/372 : from the confidence intervals, it looks like the fit is quite uncertain. How should these parameters be used if there are so uncertain ?
- Ln 382 : Do you have a more detailed hypothesis of why SEA BBAs might have a better CCN ability compared to other regions ? (based on the literature ?)
- Ln 410 : in this paragraph, what can the given fit be interpreted, for instance does it tell us anything about the adiabaticity of the sampled clouds ?
- Ln 428 : maybe « linked » can be replaced with « co-located »
- Ln 450 : in the methods, it is said that the trajectories are initiated from an altitude of 341m, yet on Fig. 7, it looks like they come from an altitude of 2 km, why is that ?
- Ln 478: Have you explained what the air-density-weight transformation means for AOT?
- Ln 527: "observed larger droplets" is mostly true, but at -24h on Fig. 8, the droplet sizes are decreasing. Why is that?
- Ln 573: "a stronger relationship" could be made more precise with "a stronger linear correlation"
- Ln 576-577: Could it be interesting to add a quantification of relatively how much mass of aerosols is entrained along the transport vs at the place of observation?  Or how much percent of the mass is entrained in the identified "efficient mixing regions"?
- Ln 845: "Langitude" typo in the x-axis of Fig.1.
- Ln 845: why are the fire counts cumulated over August 2017 only, and not over the exact study period?
- Ln 860: the f) and g) labels have been swapped compared to what is shown in the figure's subplots

**References:**

Haywood, et al.: The CLoud–Aerosol–Radiation Interaction and Forcing: Year 2017 (CLARIFY-2017) measurement campaign, Atmos. Chem. Phys., 21, 1049–1084, https://doi.org/10.5194/acp-21-1049-2021, 2021.

Klein, S. A., & Hartmann, D. L. The seasonal cycle of low stratiform clouds. *Journal of climate*, 6(8), 1587-1606, 1993

Hill, A. A., Feingold, G., & Jiang, H. The influence of entrainment and mixing assumption on aerosol–cloud interactions in marine stratocumulus. *Journal of the Atmospheric Sciences*, 66(5), 1450-1464, 2009

---

## Author Comment (AC1)

**Reply to the Review of Manuscript EGUSphere-2024-3975:**

We would like to sincerely thank the editor and reviewer for their time, effort, and thoughtful feedback on our manuscript. The reviewer comments are shown in blue, with the authors' responses shown in **black** and any edited manuscript language shown in ***italicized black font***.

**General comments:**

Wu et al. presented aircraft observations on aerosol-cloud interactions over Ascension Island in the Southeast Atlantic, focusing on biomass burning aerosols from Africa. In-situ measurements of aerosol and cloud droplet properties, along with simulation experiments using the UK Met Office's Numerical Atmospheric Modelling Environment (NAME) and Unified Model, were conducted. This study also considered the effects of atmospheric conditions, such as in the marine boundary layer or free troposphere, on data analysis. Despite many language and presentation issues, the key findings are presented. In general, this study provides insights into cloud-aerosol interactions over remote oceans and shows the role of biomass burning aerosols under different atmospheric conditions. However, the manuscript requires substantial revision to improve the quality of data analysis and presentation. The main text should be better structured, and the results (see detailed comments below) need to be reanalyzed and better interpreted. Once these issues are addressed, the paper could be further considered for publication in Atmospheric Chemistry and Physics (ACP).

1.  The abstract is overly redundant and needs revision for clarity and conciseness. It should be carefully restructured to avoid repetition and follow the guidelines provided by ACP, particularly regarding word limit and structure. Please refer to the ACP author guidelines for the required format https://www.atmospheric-chemistry-and-physics.net/policies/guidelines_for_authors.html.

Thanks to the reviewer's suggestions on the abstract. The authors have revised the abstract for clarity and conciseness, following the guidelines provided by ACP. The revised abstract is:

*African biomass burning (BB) aerosols transported over the southeast Atlantic (SEA) strongly influence cloud properties but remain a major source of uncertainty in regional climate assessment. This study characterizes vertical profiles of thermodynamic conditions, aerosol properties, and cloud microphysics around Ascension Island during an aircraft campaign (August–September 2017). Backward-dispersion simulations evidence that observed pollution originated from long-range transported African BB plumes. In BB-polluted marine boundary layers (MBL), aerosol number concentrations ($N_a$) were substantially elevated relative to the clean MBL, driving increased cloud droplet number concentrations ($N_d$) and reduced cloud effective radii ($R_e$). Cloud-layer mean $N_d$ correlated strongly with aerosols below the cloud (sub-$N_a$) but weakly with free-tropospheric (FT) aerosols. Enhanced sub-$N_a$ was due to BB aerosols entrained from the FT into the MBL along long-range transport and/or locally. Droplet activation fractions were similar in clean and moderately BB-polluted (sub-$N_a < 700\ cm^{-3}$) clouds, while a weaker $N_d$-$N_a$ correlation was observed in more polluted clouds. Region-specific $N_d$-$N_a$ parameterizations are necessary for representing BB aerosol-cloud interactions over the remote SEA. A robust inverse $N_d$-$R_e$ relationship was observed, regardless of BB influence. By coupling backward simulations with satellite retrievals, this study indicates that FT-to-MBL entrainment of African BB aerosols over the SEA occurs several days before arrival at Ascension Island, predominantly west of 0° E for examined cases. These findings provide unique observational constraints for representing aerosol-cloud interactions and vertical*

*transport of Arican BB aerosols in climate models, offering improved assessments of African BB impacts over the SEA.*

2. The introduction is poorly developed and lacks a clear logical flow. In the first paragraph, the authors attempt to discuss the effects of biomass burning aerosols (BBAs) on cloud properties under different scenarios based on their vertical location relative to the cloud layer. However, the points are disorganized, and the lack of a clear leading sentence makes it difficult for readers, especially for those outside the field of cloud-aerosol interactions, to follow. The second and third paragraphs discuss the limitations of space-borne observations and model simulations respectively, but the authors fail to introduce specific research questions for the study. The statements in Lines 123-131 are too general and do not contribute enough detail. We would like to suggest the authors streamline the introduction and clarify the research questions to improve clarity and readability.

Thanks to the reviewer's suggestions on the introduction section. The authors have re-organized the introduction and added leading or/and conclusion sentences in each paragraph. For example:

*"Most previous studies have focused on assessing the direct effect (solar absorption and scattering by BB aerosols) and the semi-direct effect (cloud adjustments in response to the direct radiative effect) of transported African BB aerosols."*

*"In contrast, the effect of BB aerosols on clouds by acting as cloud condensation nuclei (CCN) (known as the indirect effect) has received less attention compared with their direct and semi-direct effects over the SEA."*

*"However, quantifying the cloud response to BB aerosols over the SEA remains uncertain due to observational limitations." "More accurate in-situ measurements of aerosol-cloud relationship are needed to quantify the response of clouds to BB aerosols in this region."*

*"Another uncertainty in assessing BB aerosol-cloud interactions over the SEA arises from model biases in simulating the vertical distribution of transported BB aerosols."*

The authors have briefly summarized the related findings that have already been published using the campaign data and expanded on the novel research questions this study will address. The revised manuscript is as follows:

*To address the aforementioned issues, aircraft in-situ measurements are essential to provide unique constraints on the vertical distribution of transported African BB aerosols over the SEA for climate models. Aircraft observations with continuous vertical sampling are also the most reliable source for accurately characterizing the correlations between aerosols and clouds. This study presents airborne observations of BB aerosols and clouds collected during the CLARIFY campaign (Cloud-Aerosol-Radiation Interactions and Forcing; August–September 2017), which was based around Ascension Island (7.96° S, 14.35° W) (Haywood et al., 2021). **The CLARIFY campaign addresses a key observational gap over the remote SEA, contributing to a more comprehensive understanding of BB aerosol vertical distributions and interactions with clouds across the SEA.** When integrated with complementary campaigns such as ORACLES (ObseRvations of Aerosols above CLouds and their intEractionS), which primarily focused on regions westward of the African continent and eastward of 0° E (Redemann et al., 2021), **the CLARIFY campaign provides an integrated wide-scale assessment of BB aerosol transport and its impact on cloud microphysics over the SEA.** Observations from these collaborative projects have indicated that the physicochemical properties of BB aerosols continuously evolve during weeklong transport, influenced by aging, cloud processing, and MBL environments, implying different CCN activity after long-range transport (Wu et al., 2020; Dobracki et al., 2023). Additionally, CLARIFY observations have shown*

*that there is often a complex vertical structure of BB aerosol relative to cloud layers, as well as vertical variability of aerosol chemical composition and size distributions (Wu et al., 2020; Haywood et al., 2021).* ***These complexities underscore the significance of investigating the mechanisms by which long-range transported BB aerosols modulate cloud microphysics over the remote SEA.***

*In this study, we **first present the pollution conditions over Ascension Island and trace the origins of air masses using backward-dispersion simulations** (Sect. 3.1). We then **characterize the vertical profiles of thermodynamic variables, aerosol properties, and cloud microphysics over Ascension Island** (Sect. 3.2). While the region is influenced by the transition from stratocumulus to cumulus clouds, associated with increasing sea surface temperatures (Gordon et al., 2018), this study **focuses on assessing the effects of BB aerosols on stratocumulus cloud microphysics** (Sect. 3.3). More details of cloud types (stratocumulus or cumulus clouds) collected during the CLARIFY are provided in Sect. 3.2. Finally, we integrate air parcel analysis with satellite observations (Spinning Enhanced Visible and Infrared Imager, SEVIRI), **to identify the efficient entrainment regions where FT air parcels from Africa are likely to enter the MBL over the SEA and to demonstrate their impact on cloud properties along transport** (Sect.3.4).*

3.  The Cloud Droplet Probe detects cloud droplets down to 2.0 μm. How does overlooking droplets smaller than 2.0 μm affect the investigation and conclusions?

    Thanks to the reviewer for raising the potential that droplets smaller than 2.0 μm may be overlooked. Typically cloud droplet diameters are much greater than this size. However, to check the potential influence of cloud droplets smaller than 2.0 μm—below the detection limit of the Cloud Droplet Probe (CDP), the authors did a statistical analysis of measured cloud droplet number and volume size distributions obtained from CDP measurements. Figure R1 shows the number (left axis) and volume (right axis) size distributions of cloud droplets sampled during the CLARIFY flights used in this study. The size distributions indicate that droplets smaller than 2.0 μm contributed minimally to cloud droplet volume concentrations and liquid water content.

[Figure]

**Figure R1. Droplet number (left axis) and volume (right axis) size distributions of clouds sampled during the CLARIFY flights used in this study. The line and shades represent medians, and ranges from 10 to 90 percentiles.**

4.  For Figure 3, the authors need to provide evidence or results to clearly classify biomass burning (BB)-impacted cases and those free of BB impacts. The manuscript mentions this classification, stating that "From 21st to 25th Aug, the MBL

became cleaner, with BB pollution predominantly in the FT. From 26th to 31st Aug, the MBL was BB-impacted again, with BBAs observed in both the MBL and FT." However, more detailed results and clear classification criteria should be unambiguously provided.

Thanks to the reviewer's suggestions on classification criteria. More details of the classification criteria for BB pollution have been added in the manuscript:

*Following established criteria used in previous CLARIFY studies, the presence of BB-pollution was defined using thresholds of carbon monoxide above 83 ppb and black carbon above 0.1 $\mu g\ m^{-3}$ (Wu et al., 2020; Haywood et al., 2021).*

5. For the results in Figure 4, further discussion is needed to gain more insights. For example, some above-$N_a$ values (2000-2600 $cm^{-3}$) are significantly higher than sub-$N_a$. Are these high above-$N_a$ data points due to $N_d$ activation being limited by updraft velocity or supersaturation? For lower $N_a$ values, both above- and below-cloud cases may be limited by aerosol particle concentrations. Thus, the differing correlation coefficients between $N_d$ and $N_a$ for sub-cloud and above-cloud cases may result from different $N_a$ ranges. What happens if the same $N_a$ range is compared for both cases? The coefficient difference may be smaller, and this should be further investigated.

Thanks to the reviewer's suggestion to further investigate $N_d$–$N_a$ relationships for above- and sub-cloud cases. The objective of comparing $N_d$–$N_a$ relationships between above- and sub-cloud cases is to investigate the activation mechanism (cloud-base activation and cloud-top activation through entrainment). For below-cloud cases, the $N_d$ exhibited a stronger positive correlation with sub-$N_a$ in the clean or moderately BB-impacted clouds (sub-$N_a$ < 700 $cm^{-3}$), compared to more polluted clouds (sub-$N_a$ > 700 $cm^{-3}$). Following the reviewer's suggestion, we have discussed correlation coefficients from different $N_a$ ranges for below-cloud cases. We have also discussed the possible reasons why the correlation between $N_d$ and sub-$N_a$ was weaker in more-polluted clouds, e.g., differences in updraft velocity and the strength of cloud-top entrainment processes. The overall correlation of $N_d$ with above-$N_a$ was weaker and more complex than sub-$N_a$ at the place/time of observation, indicating that cloud base activation of $N_a$ played a greater role as compared to cloud-top activation of $N_a$ through entrainment. Following the reviewer's suggestion, we have discussed correlation coefficients from different $N_a$ ranges for above-cloud cases. For polluted FT cases. the $N_d$ exhibited a fairly positive correlation with above-$N_a$ (r = 0.74, p <0.01) when above-$N_a$ was below 2000 $cm^{-3}$, while the correlation became weaker when above-$N_a$ was above 2000 $cm^{-3}$. We have also discussed the possible reasons for the weak correlation between $N_d$ and above-$N_a$ at the place/time of observation, e.g., entrainment rate and time duration. The variation in correlations between $N_d$ and above-$N_a$, e.g. a weaker correlation when above-$N_a$ was above 2000 $cm^{-3}$, may be due to differences in the strength of cloud-top entrainment.

We have rephrased the description of correlation coefficients as below:

*Figures 5a and 5b show the relationship between cloud-layer mean $N_d$ and $N_a$ concentrations for all the analyzed profiles, in terms of sub-$N_a$ and above-$N_a$ respectively The $N_d$ exhibited a strongly positive correlation with sub-$N_a$, with a Pearson correlation coefficient (r) of 0.89 (p <0.01, statistically significant) for all analyzed profiles. The positive correlation between $N_d$ and sub-$N_a$ was stronger (r = 0.93, p <0.01) in the clean or moderately BB-impacted clouds (sub-$N_a$ < 700 $cm^{-3}$), while a weaker correlation was observed in more polluted clouds (sub-$N_a$ > 700 $cm^{-3}$). In contrast, the overall correlation between $N_d$ and above-$N_a$ was weak but statistically significant when considering all analyzed*

*profiles (r = 0.41, p < 0.01). For polluted FT cases (circle markers in Fig. 5b), the $N_d$ exhibited a fairly positive correlation with above-$N_a$ (r = 0.74, p <0.01) when above-$N_a$ was <2000 $cm^{-3}$, while the correlation became weaker when above-$N_a$ was >2000 $cm^{-3}$. These results suggest that the influence of above-$N_a$ on cloud properties was weaker than sub-$N_a$ at the place/time of observation, indicating that cloud base activation of $N_a$ played a considerably greater role as compared to cloud-top activation of $N_a$ through entrainment. Particularly, the cloud base activation of $N_a$ could play a dominant role, on observation days characterized by a clean FT but BB impacted MBL (triangle markers in Fig. 5).*

6. The results section lacks coherence and a clear storyline. A more organized presentation would improve readability. For example, first classify conditions with and without BBA influences, providing evidence of BBA presence. Air mass footprint and trajectory results should be presented earlier to highlight differences in aerosol sources. Then, present the properties (aerosol and cloud) of each classified case. Next, show the relationship between cloud-layer mean $R_e$ and the average LWC/$N_d$ ratio, followed by the development of the parameterization and comparisons with the literature. This is just a suggestion; the authors may find other ways to better present the results.

Thanks to the reviewer's suggestion on the result section. The authors have re-organized the structure of result sections. We first added one section to present BB pollution conditions during the campaign period and to provide original airmass sources and evidence of BB presence by employing NAME simulations. The added section is as follows:

*3.1 BB pollution conditions and source analysis*

*Figure 2a shows complex vertical distributions of aerosol number concentrations ($N_a$, 0.1 – 3 μm) from PCASP measurements for the CLARIFY flights used in this study, alongside the estimated $z_i$. The $z_i$ estimates were derived from airborne measurements of θ and $q_l$, using Eq. (5). Following established criteria used in previous CLARIFY studies, the presence of BB-pollution was defined using thresholds of carbon monoxide above 83 ppb and black carbon above 0.1μg $m^{-3}$ (Wu et al., 2020; Haywood et al., 2021). Table S1 summarizes the BB pollution conditions in the FT and MBL respectively, for the 17 flights used in this study. During the CLARIFY campaign, BB pollution events were observed at different times to be solely in the MBL, or solely in the FT, or in both layers. Concurrent surface observations on Ascension Island (Zuidema et al., 2018) presented the same trend of MBL BB pollution as observed during the CLARIFY campaign. Based on the BB pollution conditions, the CLARIFY campaign can be divided into three periods: Period 1 from August 16 to 20, BB aerosols were observed to exist predominantly in the MBL, and the FT was mainly clean; Period 2 from August 21 to 25, the MBL became considerably clean, while the BB pollution existed predominantly in the FT; Period 3 from August 26 to 31, the MBL was BB-impacted again, and the BB aerosols were observed in both the MBL and FT.*

*NAME backward-dispersion fields are used to represent the horizontal footprint of air parcels transported over the past 7 days before arriving at the sampling area over Ascension Island. Figure S2 provides three examples of backward-dispersion fields (cases 1 to 3), with release times indicated by red markers in Fig. 2a. These cases illustrate distinct and representative transport pathways of original air parcels for three periods. Case 1 released tracers near the end of Period 1 (12:00 UTC, August 18). Cases 2 and 3 released tracers at the beginning of Periods 2 (12:00 UTC, August 21) and 3 (12:00 UTC, August 26) respectively, coinciding with shifts in MBL pollution conditions. FT dispersion*

*simulations (Figs. S2a–c) indicate that most FT air parcels over Ascension Island originated from westerly flow across the SEA and African continent. Polluted FT cases (Figs. S2b, c) show a substantially greater influence from African continental airmasses compared to the clean FT case (Fig. S2a). MBL dispersion simulations (Figs. S2d–f) suggest that MBL air parcels over Ascension Island mainly arose from clean oceanic flow that transported from the southeast to northwest over the SEA. Polluted MBL cases (Figs. S2d, f) also show contributions of westerly flow originated from African continent, while negligible continental influence in the clean MBL case (Fig. S2e). We calculated fractional contributions of airmass from African continent (20°S – 5°N, 9°W – 35°E) to Ascension Island area throughout the campaign period, differentiating between FT (Fig. 2b) and MBL (Fig. 2c) simulations. These contributions closely tracked the influence of BB plumes from African wildfire regions (Fig. 1, fire maps) throughout the campaign period. The temporal evolution of continental contributions aligns well with the observed FT and MBL pollutions over Ascension Island. NAME simulations evidence that the FT and MBL pollutions observed during the campaign were attributable to long-range transport of African BB plumes.*

[Figure]

**Figure S2. Examples of backward-dispersion fields from NAME simulations, illustrating representative horizontal footprints of original air parcels transported over the past 7 days before reaching Ascension Island area. a-c) Top and d-f) bottom panels are results from FT and MBL simulations respectively, for three example cases. Left two panels (a, d) are Case 1 released on 18 August 2017 at 12:00 UTC, middle two panels (b, e) are Case 2 released on 21 August 2017 at 12:00 UTC, and right two panels (c, f) are Case 3 released on 26 August 2017 at 12:00 UTC. The black boxes represent the release area around Ascension Island. The red box in a) represent the horizontal grids (20°S – 5°N, 9°W – 35°E) used for calculating fractional contributions of airmass from African continent to Ascension Island.**

[Figure]

*Figure 2. a) Vertical distributions of aerosol number concentrations ($N_a$, 0.1 – 3 μm) from PCASP measurements for the CLARIFY flights used in this study, alongside the estimated $z_i$. The red markers represent the release times for three NAME example cases. b, c) Fractional contributions of airmass from African continent (20°S – 5°N, 9°W – 35°E) to Ascension Island area throughout the campaign period, differentiating between b) FT and c) MBL simulations.*

Then, as in the previous manuscript, we focused on observational results.

*3.2 Vertical profiles of thermodynamics, aerosol and cloud*

*3.3 Aerosol-cloud interaction around Ascension*

    *3.3.1 Aerosol-$N_d$ relationship*

    *3.3.2 $R_e$ relationship with $N_a$ and $N_d$*

In the last section, we focused on the investigation of vertical transport and entrainment processes. We integrate air parcel analysis with satellite observations, to identify the efficient entrainment regions where FT air parcels from Africa are likely to enter the MBL over the SEA and to demonstrate their impact on cloud properties along transport.

*3.4 Vertical transport history of observed MBL pollutants*

    *3.4.1 Entrainment area*

    *3.4.2 Cloud fields along transport*

7.   As shown in Figure 2, the flight altitude can reach 6 km, where ice crystals may be sampled, meaning that the aircraft could fly through mixed-phase clouds. What is the impact of sampling ice crystals? Were measures taken for the sampling inlet to account for this or to evaluate the influence? The authors refer to previous studies for details on instrumentation and experiments, but this may not provide enough information for new readers.

Thanks to the reviewer for raising this useful point. To interpret the potential influence of ice crystals, the authors did a statistical analysis of measured ice particle number concentrations obtained from a Two- Dimensional Stereo Optical Array Probe (2DS). The 2D-S (SPEC Inc, 2024) provided high sample volume images of cloud droplets and ice particles

ranging in diameter from 10 to 1280 μm. Ice particle concentrations were derived and categorized according to morphology using circularity definitions as described by SPEC Inc (2024). Figure R2 shows the vertical profile of a) ice particle number concentrations ($N_{ice}$) and b) cloud droplet number concentrations ($N_d$) sampled from the flights used in this study. At high altitudes, there were negligible amounts of ice crystals and cloud droplets. At low altitudes, the amounts of ice crystals were also minor compared to cloud droplets. Additionally, aerosol measurements inside clouds are suggested to be unreliable and in-cloud aerosol data have been removed in this study. Overall, the influence of ice crystals on PCASP results is suggested to be negligible.

[Figure]

**Figure R2. Vertical profile of a) ice particle number concentrations ($N_{ice}$) and b) cloud droplet number concentrations ($N_d$) sampled from the flights used in this study.**

8. For high-speed aircraft in-situ sampling, correcting the sampled volume to the ambient volume (or standard conditions) is challenging due to air compression, which can cause significant volume differences. Did the authors account for the effects of air compression at high speeds? If so, how was it corrected or calculated?

   Thanks to the reviewer for raising this important point. We have accounted for the effects of air compression during high-speed in situ sampling aboard the FAAM BAE-146 research aircraft. The ambient static pressure and temperature were obtained from the Rosemount pitot-static system, and stagnation (inlet) conditions were recorded near the aerosol sampling inlet. To correct the sampled volume to ambient atmospheric conditions, we applied the volumetric correction a described in in previous studies (Trembath et al., 2013).

   We have added the brief description in the method section: "*Aerosol concentrations measured aboard the FAAM BAe-146 were corrected for ram air compression at inlet speed using the volumetric correction as described in previous studies (Trembath et al., 2013)*."

9. This study used a water condensation particle counter (CPC, model: 3786-LP). It is reported that water CPC undercounts hydrophobic organic-rich soot particles (Keller et al., 2013), potentially a component of BBAs. How does this undercounting affect the results and analysis in the study?

   Thanks to the reviewer for highlighting the potential limitations of water-based CPC in detecting hydrophobic organic-rich soot particles (Keller et al., 2013). The BB aerosols observed in this study have undergone week-long transport and were highly aged. As shown in previous studies, aging processes during transport can reduce the hydrophobicity of BB organics due to their oxidation and internally mixing with secondary species such as sulfates (i.e. Pósfai et al., 2003). Thus, aged BB aerosols in this study are likely to be more easily detected than fresh BB aerosols by water CPCs.

Additionally, CPC measurements were used primarily to show general aerosol loading and the CCN activation fractions of submicron aerosols (CCN/CN$_3$), possible undercounting would not significantly affect the core conclusions.

We have added the brief description in the manuscript: "*The CPC used in this study is water-based and may undercount hydrophobic particles, particularly soot-rich BB aerosols (Keller et al., 2013). However, most of the sampled BB plumes have undergone week-long transport. Aging processes may have reduced the hydrophobicity of BB organics due to their oxidation and internal mixing with secondary species such as sulfates (Pósfai et al., 2003). Therefore, undercounting is expected to be minimal for the conditions of this study.*"

10. Relevant statements are needed to introduce and briefly discuss the results in the supplementary tables and figures. Simply presenting a figure or table without context is too abrupt.

Thanks to the reviewer for the suggestion. The authors have added more descriptions regarding the supplementary tables and figures.

11. There are way too many abbreviations, which does not help the presentation but makes the reading more difficult. Please check through the manuscript and reduce the use of abbreviations.

Thanks to the reviewer for the suggestion. The authors have reduced the use of abbreviations throughout the manuscript.

12. The language needs revision to improve readability. Many statements are too long, increasing the likelihood of errors and making them hard to follow.

Thanks to the reviewer for the suggestion. The authors have reduced the use of long statements throughout the manuscript.

**Specific comments:**

Line 17: "vertical structures of thermodynamics, aerosol properties and cloud microphysics" sounds awkward

The revised manuscript is:

*This study characterizes vertical profiles of thermodynamic conditions, aerosol properties, and cloud microphysics around Ascension Island during an aircraft campaign (August–September 2017).*

Line 20-23: This sentence is too long and it is redundant.

The revised manuscript is:

*In BB-polluted marine boundary layers (MBL), aerosol number concentrations ($N_a$) were substantially elevated relative to the clean MBL, driving increased cloud droplet number concentrations ($N_d$) and reduced cloud effective radii ($R_e$).*

Line 23-24: The last part of the sentence can be more concise. And 'immediately' is not an appropriate adverb here.

The "immediately" has been deleted.

Line 47: It is stated that 'stratocumulus cloud (Sc) has been extensively studied'. However, the authors later on tried to bring

out the research question about the impacts of biomass burning aerosols (BBAs) from African wildfires on the Sc deck over the southeast Atlantic (SEA) Ocean. Isn't it self-contradictory?

The introduction has been revised. This sentence has been deleted.

Line 47-48: The statement in 'These clouds are climatically important, as they reflect a significant amount of solar radiation and exert only a small radiative effect in the longwave, leading to a net cooling effect.' Is not very clear. Is the small longwave radiative effect positive or negative? And relevant references are missing.

The revised manuscript is:

*The SEA region is climatically important, due to the significant net cooling effect of these stratocumulus clouds through their strong reflection of solar radiation but small positive longwave radiative effect (Wood, 2012).*

Line 68: should it be 'BB' or 'BBA layers'? Is it really necessary to abbreviate both BB and BBA?

The abbreviation of "BBA" has been replaced by "BB aerosols".

Line 69: What is 'sub-cloud relative humidity (RH)'? Please make a clear definition since it is not a standard or regular terminology and it is used quite often through the manuscript.

Line 69-70: The statement in 'In the smoky MBL, BBA heating tends to reduce the sub-cloud relative humidity (RH) and liquid water content, thereby decreasing the cloud cover Zhang and Zuidema (2019) may lead to misunderstanding that BBAs heat the cloud and thus decrease liquid water path and cloud cover. It would be better to specify it as BBA induced solar radiation absorption.

Regarding the above two comments, the revised manuscript is:

*Conversely, in the smoky MBL, **BB aerosol heating induced by absorbing solar radiation** tends to **reduce the humidity below the cloud**, thereby decreasing the cloud cover (Zhang and Zuidema, 2019).*

Line 74: The authors have defined the semi-direct effect in Line 67. It would be good to also define the indirect effect and shortly explain what is the difference between semi-direct effect and indirect effect.

The revised manuscript is:

*Most previous studies have focused on assessing the direct effect (solar absorption and scattering by BB aerosols) and the semi-direct effect (cloud adjustments in response to the direct radiative effect) of transported African BB aerosols.*

Line 79: how could the indirect effect of BBAs affect the cloud lifetime mentioned here? Is it a decrease or increase?

The revised manuscript is:

*Recent modeling studies show that BB aerosols can increase cloud droplet number concentrations ($N_d$) and reduce droplet sizes, which may enhance cloud lifetime and coverage by reducing drizzle formation (Gordon et al., 2018; Lu et al., 2018).*

Line 83: Specify the region since it is at the beginning of the paragraph.

The revised manuscript is:

*However, quantifying the cloud response to BB aerosols over the SEA remains uncertain due to observational limitations. Satellite-based observations have been widely used to examine the vertical distance between the smoke layer base and the cloud top, which plays a role in understanding the impact of BB aerosol layers on cloud properties (Costantino and Bréon, 2010, 2013; Painemal et al., 2014).*

Line 98-99: 'Recent models tend to show BBA layers descending rapidly when off the western coast of the African continent, resulting in BBAs layers that are too low in altitude over the SEA (Das et al., 2017; Gordon et al., 2018).' needs to be revised. Line 100: 'the models'. Which models? They are not introduced beforehand.

Regarding the above two comments, the revised manuscript is:

*Many simulations using Earth System models show that smoke layers descend too rapidly as they advect away from the African coast, thus the smoke layer heights were underestimated over the SEA (i.e. Das et al., 2017; Gordon et al., 2018; Shinozuka et al., 2020).*

Line 122-123: Is it only because aircraft observations over SEA region is lacking? What is advantages or advancements for aircraft observations in this study in comparison to those conducted before?

The revised manuscript is:

*The CLARIFY campaign addresses a key observational gap over the remote SEA, contributing to a more comprehensive understanding of BB aerosol vertical distributions and interactions with clouds across the SEA. When integrated with complementary campaigns such as ORACLES (ObseRvations of Aerosols above CLouds and their intEractionS), which primarily focused on regions westward of the African continent and eastward of 0° E (Redemann et al., 2021), the CLARIFY campaign provides an integrated wide-scale assessment of BB aerosol transport and its impact on cloud microphysics over the SEA.*

Line 129-130: It should be Spinning 130 Enhanced Visible and Infrared Imager (SEVIRI), instead of 'SEVIRI (Spinning 130 Enhanced Visible and Infrared Imager)'.

The revised manuscript is:

*We combine air parcel analysis with Spinning Enhanced Visible and Infrared Imager (SEVIRI) satellite observations…*

Line 136: It should be 7 September 2017 (British English) or September 7, 2017 (American English).

The time format consistently follows American English standards.

Line 137: What is the difference of 'temperature (T)' compared to 'potential temperature (θ)'? Is it ambient temperature?

The revised manuscript is:

*The aircraft was equipped with a range of instruments to derive aerosol and cloud properties, as well as meteorological variables (e.g. **ambient temperature (T, K)**, potential temperature (θ, K), and total water mixing ratio ($q_t$, g kg$^{-1}$, the total mass of vapor, liquid, and ice forms of water per unit mass of dry air)).*

Line 137: What is 'total water mixing ratio'? Definition is needed.

The revised manuscript is:

*The aircraft was equipped with a range of instruments to derive aerosol and cloud properties, as well as meteorological variables (e.g. ambient temperature (T, K), potential temperature (θ, K), and **total water mixing ratio ($q_t$, g kg$^{-1}$, the total mass of vapor, liquid, and ice forms of water per unit mass of dry air)).***

Line 138: Does it really help to abbreviate straight and level runs as SLRs? Maybe not very helpful for readers but one more abbreviation to keep in mind.

We have deleted the abbreviation of "SLRs".

Line 142: Again, is the abbreviation of 'POC' helpful and necessary?

We have deleted the abbreviation of "POC".

Line 151: Better delete 'accumulation mode' because some studies also define particles between 0.1 and 2.5 μm or between 0.1 and 1.0 μm as accumulation mode particles.

The "accumulation mode" has been deleted. The revised manuscript is:

*Aerosol number concentration in the size range of 0.1 – 3 μm ($N_a$) was obtained by integrating the PCASP distribution.*

Line 155: Manufacturer for the water CPC is missing.

The manufacturer for the water CPC has been added.

*A model 3786-LP water-filled condensation particle counter (CPC, TSI Inc.)…*

Line 161: Will the abbreviated 'STP' frequently used below? Please check through the manuscript for abbreviations and make sure only use it when it is really necessary.

We have deleted the abbreviation of "STP".

Line 166: Is the period the same for all observations on different days during the campaign?

Yes, the period for all SEVIRI observations on different days is the same during the campaign.

Line 176: To 'STP' or to the condition? Please use more scientific and formal language.

The revised manuscript is:

*Airborne aerosol and cloud measurements reported here were corrected to the standard temperature and pressure condition (273.15 K and 1013.25 hPa).*

Line 176-184: The introduction statements of Nd, Re, and liquid water content (LWC) using the three equations should be better organized. Simply putting all variables and parameters together seems too cursory.

The revised manuscript is:

*From the CDP's cloud droplet spectrum, three key cloud microphysical parameters were derived: $N_d$, $R_e$, and liquid water content (LWC). The $N_d$ was calculated by integrating the droplet size distribution, as described by Eq. (1).*

$$N_d = \int n(r)\, dr \approx \sum_{1}^{m} n(r_i) \tag{1}$$

*where $n(r_i)$ is the droplet number concentration in a particular size bin, and $r_i$ is the middle radius for each of the size bins.*

*The $R_e$ was calculated based on the Eq. (2).*

$$R_e = \frac{\int r^3\, n(r)\, dr}{\int r^2\, n(r)\, dr} \approx \frac{\sum_{1}^{m} r_i^3\, n(r_i)}{\sum_{1}^{m} r_i^2\, n(r_i)} \tag{2}$$

*The LWC was calculated based on the Eq. (3), assuming spherical droplets and a liquid water density ($\rho_{water}$) of 1 g cm$^{-3}$.*

$$LWC = \frac{4\pi}{3}\, \rho_{water} \int r^3\, n(r)\, dr \approx \frac{4\pi}{3}\, \rho_{water} \sum_{1}^{m} r_i^3\, n(r_i) \tag{3}$$

Line 190: How could there be two 'COT's? 'cloud optical thickness (COT)' and 'cloud-top height (COT)'

The revised manuscript is:

*We also obtained the above-cloud aerosol optical thickness (AOT), cloud optical thickness (COT), $R_e$ and **cloud-top height (CTH)** across the SEA region (20° W − 15° W; 30° S − 0° N) ...*

Line 195-196: This sentence is repeating the one in Lines 165-166.

The repeating part has been deleted.

Section 2.2: A clear definition for sub-cloud and above-cloud regions may be necessary.

The revised manuscript is: *Of relevance to this work are the saw-tooths and stepped profiles to characterize cloud regions. The definition for sub- and above-cloud are provided in Sect. 3.3.1. In each profile with a continuous cloud layer, we averaged the $N_a$ **within 200m above the cloud top** to obtain the **above-$N_a$**, and **within 200m below the cloud base** to obtain the **sub-$N_a$**.*

Line 206: How can people understand the 'reasonably well' performance of NWP if people do not go through the referred paper? It is too cursory to make such a statement.

The manuscript has been revised:

*NAME was chosen as an appropriate model for this study because it uses high-resolution meteorological fields (~10 km × 10 km). It can predict transport over distances ranging from a few kilometers to the whole globe, and it has been used successfully in similar studies (i.e. Panagi et al., 2020).*

Line 210-211: Isn't this sentence a repetition of the content introduced in the previous paragraph?

The NAME method section has been revised, and the repeated sentence has been deleted.

Line 225: Didn't the stratocumulus cloud defined as Sc in Line 46-47?

We have deleted the abbreviation of "Sc".

The revised manuscript is:

*Instead of using the BL depth in NAME, the estimated $z_i$ was employed to divide the column-integrated horizontal footprints into the FT and BL separately, distinguishing original air parcels as either from the FT or BL.*

The Unified Model (MetUM) is a Numerical Weather Prediction (NWP) model developed by the UK Met Office. We have made the description more concise. The revised manuscript is:

*The transport of released tracer particles was tracked backward over the past 7-days in the NAME, driven by the three-dimensional gridded (3D) meteorological fields from the UK Met Office's Unified Model (Brown et al., 2012).*

More details have been added to the manuscript:

*Following established criteria used in previous CLARIFY studies, the presence of BB-pollution was defined using thresholds of carbon monoxide above 83 ppb and black carbon above 0.1μg m⁻³ (Wu et al., 2020; Haywood et al., 2021). Table S1 summarizes the BB pollution conditions in the FT and MBL respectively, for the 17 flights used in this study.*

The error has been corrected throughout the manuscript.

The revised manuscript is:

*Figure 2a shows complex vertical distributions of aerosol concentrations ($N_a$, 0.1 – 3 μm) from PCASP measurements for the CLARIFY flights used in this study, alongside the estimated $z_i$.*

The revised manuscript is:

*Concurrent surface-based observations on Ascension Island (Zuidema et al., 2018) presented the same trend of MBL BB pollution as observed during the CLARIFY campaign.*

More details have been added to the manuscript:

*Here, we estimated the $z_i$ over the SEA, using the outputs of 3D meteorological parameters. The $z_i$ was quantified as the*

height at which the vertical gradient of liquid water potential temperature ($\theta_l$) is the largest, and there is also a steep decrease in humidity (Jones et al., 2011). The $\theta_l$ was estimated following Eq. (5).

$$\theta_l = \theta - \frac{L}{c_p} q_l \tag{5}$$

Where $\theta$ is the potential temperature, L is the latent heat of vaporization for water ($2.5 \times 10^6 \, J \, kg^{-1}$), $c_p$ is the specific heat of dry air at constant pressure ($1005 \, J \, kg^{-1} \, K^{-1}$), $q_l$ is the liquid water mixing ratio ($g \, kg^{-1}$).

Line 248: 'Fig. 3', the same comment as for Line 236.

The error has been corrected throughout the manuscript.

Line 251: What is lifting condensation level (LCL)? A clear definition is necessary since it shows up in the following discussions. It can also be clearly indicated in Fig.3a or 3b. What does it mean a surface layer? Whose surface? Or if it is a language issue, maybe better a flat layer or a layer with rather constant $z/z_i$.

More details have been added to the manuscript following the reviewer's suggestions:

*The MBL generally showed a well-mixed layer with near-constant $\theta_l$ and $q_t$ between the sea surface and the lifting condensation level (LCL, the height at which the relative humidity of an air parcel reaches 100% when lifted adiabatically), which is at an altitude of ~ 300 to 800 m ($z/z_i$ = ~ 0.2 – 0.5). Above the LCL, there was another layer that extended to the base of the inversion.*

Line 254: 'relatively' means the comparison to what?

The "relatively well-mixed" has been revised to "well-mixed".

Line 256: How can the decoupling parameters $\alpha_q$ and $\alpha_\theta$ be derived? It would be good to provide the equations and explain it with more details somewhere (e.g., in the supplement but properly referred in the main text).

More details have been added in the manuscript regarding the estimation of the decoupling parameters:

*The BL decoupling parameters $\alpha_q$ and $\alpha_\theta$, were also calculated from the respective profiles following Eq. (6) and (7) respectively, which represent the relative differences of the $q_t$ and $\theta_l$ between the surface and the upper part of the BL respectively (Zheng et al., 2011).*

$$\alpha_q = \frac{q_t(z_i^-) - q_t(0)}{q_t(z_i^+) - q_t(0)} \tag{6}$$

$$\alpha_\theta = \frac{\theta_l(z_i^-) - \theta_l(0)}{\theta_l(z_i^+) - \theta_l(0)} \tag{7}$$

*Where $\theta_l(z_i^+)$ and $q_t(z_i^+)$ are $\theta_l$ and $q_t$ at the level 50m above the inversion height, $\theta_l(z_i^-)$ and $q_t(z_i^-)$ are at the level 50m below the inversion height, $\theta_l(0)$ and $q_t(0)$ are at the lowest measured level near sea surface.*

Line 269: 'become' can be better replaced by 'act as'

The "become" has been revised to "act as".

Line 274: What is the definition of 'highly aged African BBAs in this study' and what are their property differences compared

More details have been added in the manuscript regarding the "'highly aged African BB aerosols":

*In this study, the sampled African BB aerosols have undergone week-long transport and are defined as highly aged African BB aerosols (Wu et al., 2020). Compared to freshly emitted African BB aerosols, these highly aged BB aerosols exhibited larger particle sizes, enhanced fractions of inorganics and the loss of organics after atmospheric aging such as chemical oxidation and cloud processing (Wu et al., 2020; Dobracki et al., 2023). Due to atmospheric aging, the CCN activation fractions of these highly aged African BB aerosols are generally higher than fresher BB aerosols sampled over the African continent ($CCN/N_a = 0.68$, at $SS = 0.3\%$) (Ross et al., 2003), but are comparable to transatlantic African BB aerosols observed over Amazon area during the dry season (the average ratio of CCN to aerosol concentrations in a range of 20 nm to 1μm is $0.83 \pm 0.06$ at $SS = 0.5\%$) (Holanda et al., 2020).*

Line 275: '$CN_{20}$' was not defined in the main text and only appeared once.

The revised manuscript is:

*the average ratio of CCN to aerosol concentrations in a range of 20 nm to 1μm is $0.83 \pm 0.06$ at $SS = 0.5\%$.*

Line 280: 'the majority of particles were in the size below 0.1 μm' refer to which kind of particles? Aerosol particles larger than 3 nm or CCN particles?

The revised manuscript is:

*The average ratio of $N_a/CN_3$ in the clean MBL was calculated to be $0.40 \pm 0.26$, also suggesting that the majority of submicron aerosols detected by the CPC (3 nm to 1 μm) in the clean MBL were Aitken mode particles (< 0.1 μm).*

Line 284: 'more sulfate'. Yes, it was stated before that 'The submicron aerosols from marine emissions in the clean MBL were previously reported to be dominated by sulfate (~ 60%) and organic (~ 24%).' However, there is no evidence provided to show that aerosols in the clean MBL contain more sulfate.

The revised manuscript is:

*Although the clean MBL aerosols contained **enhanced fractions** of sulfate with high hygroscopicity, the average $CCN/N_a$ ($0.83 \pm 0.20$) was close to those in the BB-impacted MBL ($0.82 \pm 0.17$), suggesting that the CCN activated fractions of accumulation-mode aerosols were similar between the clean and BB-impacted MBL conditions.*

Section 3.1:

1) It seems the second paragraph is mainly citing a previous study from Wu et al. (2020) but not presenting the results in Figure 3 in this study.

To avoid redundancy with Wu et al. (2020), vertical profiles of aerosol chemical composition under BB-impacted conditions have been provided in Supplementary Figure S2. In the main text, information of aerosol chemical composition in the clean MBL is referenced succinctly as "*The submicron aerosols from marine emissions in the clean MBL were previously reported to be dominated by sulfate (~ 60%) and organic (~ 24%) (Wu et al., 2020).*".

2) Figure 3d clearly shows that $CCN/CN_3$ under clean MBL conditions is lower than that under BB-impacted MBL

conditions and such a difference becomes smaller when it is in the FT. However, the CCN/$N_a$ in Fig. 3e did not show such a dependence on the atmospheric conditions (in the MBL or FT). Discussions on this finding is missing.

Since background aerosols in the clean FT during Period 1 mainly fell below instrumental detection limits, FT analysis of aerosol properties (CMD, CCN/$CN_3$, and CCN/$N_a$ ratios) only included those sampled during BB-impacted conditions. The red and black lines and shades in original Figs. 3c-e were summarized based on MBL pollution conditions, rather than indicating FT pollution conditions. To avoid misleading the reader, the authors have modified Figs. 3c-e and revised the manuscript.

*In this study, vertical profiles of aerosol properties (CMD, CCN/$CN_3$ and CCN/$N_a$ ratios) were summarized under clean (red line and shades in Figs. 3c-e) and BB-impacted (black line and shades in Figs. 3c-e) conditions separately. MBL analysis included both the clean and BB-impacted conditions. However, due to background aerosols in the clean FT during Period 1 being mainly below instrumental detection limits, the FT analysis (CMD, CCN/$CN_3$, and CCN/$N_a$ ratios) was restricted to BB-impacted conditions.*

[Figure]

***Figure 3.*** *a, b) Average vertical profiles of a) liquid water potential temperature ($\theta_l$, K), b) total water mixing ratio ($q_t$) for each flight used in this study. Blue, red and black lines represent measurements from Periods 1, 2 and 3 respectively. c-e) Summarized profiles of c) aerosol count median diameter derived from the PCASP (CMD, μm) and d,e) the ratio of CCN (~ 0.2%) to condensation nuclei ($D_a >$ 3 nm from the CPC) (CCN/$CN_3$) and accumulation aerosol concentration ($D_a > 0.1$ μm from the PCASP) (CCN/$N_a$) under polluted (black) and clean (red) conditions. Solid lines and shades represent median values and range from 10 to 90 percentiles. The red cross marker in Fig. 3c represents the average CMD in accumulation mode from measurements within the clean MBL. f-h) Vertical profiles of 1-hz, f) liquid water content (LWC, g $m^{-3}$), g) cloud droplet number concentration ($N_d$, # $cm^{-3}$) and h) cloud effective radius ($R_e$, μm) in sampled continuous cloud layers. Red lines represent cloud measurements in the clean MBL, and grey lines represent*

*cloud measurements in the BB-impacted MBL. It is noted that average vertical profiles of cloud properties from flight C032 are also provided in Figs. 3f-h (grey dashed lines). The y-axis uses a height scale normalized by inversion height ($z_i$).*

3) The role of particles between 3 and 100 nm can also be investigated, which is also overlooked.

We thank the reviewer for pointing out the potential importance of the 3–100 nm size range.

The revised manuscript is: *Under both clean and BB-impacted conditions, the $CCN/CN_3$ ratios remained consistently below unity, whereas $CCN/N_a$ occasionally exceeded unity, implying that a subset of Aitken mode particles (< 0.1 µm) contributed to CCN.*

4) In addition, why some data around $z/z_i$ in Figs. 3c, 3d and 3e is missing?

Since observations near top of the MBL ($z/z_i \sim 0.9$ to 1) were sampling predominantly cloudy conditions rather than clear-air aerosols, the aerosol data was deleted. Consequently, there are missing gaps in Figs. 3c–e.

5) Moreover, frequently using 'some days' (Line 253 and 255) or 'sometimes' (Line 292) makes the statement less convincible. Please use solid (clearly presented and referred) results to support the argument.

Related description in the manuscript has been revised following the reviewer's suggestions, to make the statement clearer and more convincing:

*During 6 flight cases (marked as type-d in Table S1), profiles above the LCL exhibited near-constant $\theta_l$ and $q_t$, indicating a fairly well-mixed upper layer and the presence of a decoupled stratocumulus deck. By contrast, in the remaining cases, the upper layer above the LCL was conditionally unstable, suggesting the presence of a stratocumulus-over-cumulus MBL (marked as type-e in Table S1) or a cumulus-capped MBL (marked as type-f in Table S1) structure.*

*The profile of $N_d$ (Fig. 3f) remains relatively constant, with few cases in the BB-impacted MBL (i.e. profiles in flights C029 and C038) presenting an increase near cloud top.*

Line 288-290: 'In Flight C032 with deep cumulus clouds, stepped profiles and SLRs were carried out and the average vertical profiles of cloud properties are provided.' Where was the result provided? Please refer to a figure or table.

The average vertical profiles of cloud properties for Flight C032 were provided in the original Figs. 3f-h. To make these profiles clearer, they are now shown as grey dashed lines in those panels.

The revised manuscript is: "*In Flight C032 with cumulus clouds, stepped profiles and SLRs were carried out. The average vertical profiles of cloud properties from flight C032 are also provided in Figs. 3f-h (grey dashed lines).*"

Line 299-301: Cloud condensation nuclei concentrations depend on cloud supersaturations. For the focus of this study, it is the contribution of $N_a$ to cloud condensation nuclei numbers that impacts cloud microphysics.

The revised manuscript is:

*Here, we use $N_a$ as the metric instead of CCN concentrations to establish the aerosol impact on cloud microphysics, since CCN concentrations depend on cloud supersaturations. In this study, we focus on aerosols in the size above 0.1 µm that behaved as the dominant CCN (as discussed in Sect. 3.2) and impacted cloud microphysics.*

Line 302: Should be subsection 3.2.1

The subsection has been added.

Line 310-311: What is the p value for the calculated Pearson correlation coefficients? It would be good to indicate the value in Figure 4 accordingly.

The p values have been added.

Line 312: Regarding the statement 'especially for the flights with clean FT but high Na in the BB-impacted MBL (dashed black box in Fig. 4b)', the data point with $N_d$ close to 400 cm$^{-3}$ in the black box looks like an outliner. There are only three data points in the box. It is far-fetched to treat it as a special case to investigate the correlations.

The dashed black box in the previous Fig. 4b highlights profiles sampled during Period 1, characterized by a clean FT but BB impacted MBL. In this case, activation of $N_a$ at the cloud base played a dominant role in determining $N_d$. Therefore, the weak correlation was observed between $N_d$ and above-$N_a$ in these profiles, which should not be considered as outliers.

Regarding the above two comments, the revised manuscript is:

*Figures 5a and 5b show the relationship between cloud-layer mean $N_d$ and $N_a$ concentrations for all the analyzed profiles, in terms of sub-$N_a$ and above-$N_a$ respectively The $N_d$ exhibited a strongly positive correlation with sub-$N_a$, with a Pearson correlation coefficient (r) of 0.89 (p <0.01, statistically significant) for all analyzed profiles. The positive correlation between $N_d$ and sub-$N_a$ was stronger (r = 0.93, p <0.01) in the clean or moderately BB-impacted clouds (sub-$N_a$ < 700 cm$^{-3}$), while a weaker correlation was observed in more polluted clouds (sub-$N_a$ > 700 cm$^{-3}$). In contrast, the overall correlation between $N_d$ and above-$N_a$ was weak but statistically significant when considering all analyzed profiles (r = 0.41, p < 0.01). For polluted FT cases (circle markers in Fig. 5b), the $N_d$ exhibited a fairly positive correlation with above-$N_a$ (r = 0.74, p <0.01) when above-$N_a$ was <2000 cm$^{-3}$, while the correlation became weaker when above-$N_a$ was >2000 cm$^{-3}$. These results suggest that the influence of above-$N_a$ on cloud properties was weaker than sub-$N_a$ at the place/time of observation, indicating that cloud base activation of $N_a$ played a greater role as compared to cloud-top activation of $N_a$ through entrainment. Particularly, the cloud base activation of $N_a$ could play a dominant role, on observation days characterized by a clean FT but BB impacted MBL (triangle markers in Fig. 5).*

Line 318-321: What is the relative position of FT, BB layer and the cloud? What is the cloud vertical extent? Without a clear statement or a schematic to show their relative altitudes, one can image different scenarios and can misunderstand the results.

The revised manuscript is:

*The bottom of the BB layer in the FT was defined as the lowest altitude of the plume where observed $N_a$ exceeded 500 cm$^{-3}$ (Gupta et al., 2021). The distance from the top of cloud layers in the MBL to the bottom of BB layers in the FT is referred to as Cloud Top to Aerosol Base (CTtoAB). Figure 6a shows the relationship between sub-$N_a$ and the CTtoAB. An overall negative correlation was observed between sub-$N_a$ and CTtoAB.*

Line 345-346: Please refer to the results in which Figure and which panel.

The revised manuscript is:

*During CLARIFY, the BB-impacted MBL had substantially enhanced sub-$N_a$ (212 – 1183 cm$^{-3}$, black markers in Fig. 5a) compared to the clean MBL (56 – 315 cm$^{-3}$, red markers in Fig. 5a).*

Line 377-379: why there is a difference? Please shortly explain it. Is it because of limitations in this study?

Line 381-382: What aerosol properties are responsible for the differences? Please clarify it. Simply leaving a very general (well-known) statement does not contribute to the advances of the field.

Regarding the above two comments, more details have been added to the manuscript. The revised manuscript is:

*Our observations suggest a small difference in the response of $N_d$ to $N_a$ between BB-impacted and clean MBL profiles. This is likely due to their comparable CCN activation abilities of accumulation-mode aerosols under two MBL conditions (as discussed in Sect 3.2). In contrast, a previous study reported higher droplet activation fractions for the cleaner MBL compared to the BB-impacted MBL over the Pacific Ocean (Mardi et al., 2019). The discrepancy likely stems from differences in their droplet activation behaviors of transported BB aerosols between the studies. The β value for BB-impacted MBL cases in this study (0.71 (0.42 – 0.92)) is in a higher range than that reported for BB-impacted areas off the California coast of North America (0.26 (0.15 – 0.42)) in Mardi et al. (2019). A key factor contributing to the different droplet activation behaviors of transported BB aerosols, may be the variability in aerosol chemical composition and CCN activity, which depends on source combustion conditions and aging process (Wu et al., 2020; Farley et al., 2025). Submicron BB aerosols from western U.S. wildfires have been reported to be dominated by organic (~90%) with minimal inorganic content (<2%) from near-source to regional scales (0.5 hours to several days) (Farley et al., 2025). In contrast, highly aged African BB aerosols were reported to contain ~35% inorganic mass (Wu et al., 2020). This implies that highly aged African BB aerosols are more hygroscopic, as inorganics typically have higher hygroscopicity than organics on a global scale (Pöhlker et al., 2023). Additionally, transported BB aerosols from western US wildfire presented similar accumulation-mode aerosol size distributions to this study (Laing et al., 2016). Consequently, the CCN activation ability of transported African BB aerosols in this study is broadly higher than that reported for aged BB aerosols from Western/Northern American wildfires (CCN/CN = 0.11 – 0.62, at SS = 0.2 – 0.5%) (Pratt et al., 2011; Zheng et al., 2020), leading to the observed differences in their $N_a$-$N_d$ relationship between these studies.*

The subsection has been added.

The reference has been added to the manuscript. The corresponding figures for the results have been referred to.

The revised manuscript is:

*The relationship between $R_e$ and sub-$N_a$ (Fig. 7a) shows a weaker correlation than that between $N_d$ and sub-$N_a$ (Fig. 5a), likely due to the influence of additional atmospheric factors such as the MBL thermodynamic structure, cloud depth, cloud-top entrainment process, etc (Wood, 2012; Herbert et al., 2020).*

Based on the BB pollution conditions, the CLARIFY campaign can be divided into three periods: Period 1 from August 16 to 20, BB aerosols were observed to exist predominantly in the MBL, and the FT was mainly clean; Period 2 from August 21 to 25, the MBL became considerably clean, while the BB pollution existed predominantly in the FT; Period 3 from August

26 to the end of the campaign, the MBL was again impacted by BB, and the BB aerosols were observed in both the MBL and FT. The three cases chosen were typical and representative examples for the three periods classified in this study. The revised manuscript is:

***These cases illustrate distinct and representative transport pathways of original air parcels for three periods****. Case 1 released tracers near the end of Period 1 (12:00 UTC, August 18). Cases 2 and 3 released tracers at the beginning of Periods 2 (12:00 UTC, August 21) and 3 (12:00 UTC, August 26) respectively, coinciding with shifts in MBL pollution conditions.*

Line 431: The journal does not allow the use of sub-panels like a1, a2, and a3. Instead, all panels should be listed following the alphabetic table.

The sub-panel labels have been corrected, following the alphabetic table.

Line 435: Why start with case 2 first but not case 1?

In the revised manuscript, cases 1 and 3 (BB polluted MBL) are described before the case2.

Line 439-440: the results in which previous figure supports the statement 'and the MBL was observed to be BB-impacted over Ascension Island' ? Please refer to it. If not, please provide relevant results to support it (for case 1 and 3).

The authors have reorganized the structure of this manuscript. The relevant description of this comment has been relocated to Section 3.1. The corresponding figures that can support this statement have been referred to.

Line 511-512: Why present the results in Fig. S9 but not in the main text? Since it is discussed in the main text.

Figure S9 has been moved to Sect. 3.2 (Fig. 4) in the main text.

Line 575: Better not to use 'primary activation' and 'secondary activation'. Instead, please directly state-out the activation pathway. This helps the readability.

The "primary activation" has been revised to "cloud-base activation" via updrafts carrying aerosols to the cloud base. The "secondary activation" has been revised to "cloud-top activation" via entrainment through turbulent mixing at the cloud top.

**Technical corrections:**

1. The comments on language and statements in the above specific comments are not exhausted. There are also many long sentences which generally have more grammatic errors and make it hard to digest. Please carefully go through the manuscript and improve the presentation.

   The authors have reduced the use of long statements and checked grammar throughout the manuscript.

2. Figure 1: It will be helpful to show the grid lines to guide naked eyes for reading flight paths.

   The grid lines have been added in Fig. 1.

3. Equations should be numbered. This is missing.

The numbers of Equations have been added.

 English for dates need to be checked through the manuscript. Please use either British or American style.

The date format now consistently follows American English standards.

5. Table S1: Full spelling for MBL and FT is necessary.

The full spelling for MBL (marine boundary layer) and FT (free troposphere) has been added to Table S1.

6. Figure 2: Axis ticks should be indicated for the ease of reading. Please also show the height for MBL as a function of time, which will help the discussions in Line 239-242.

The axis ticks have been added on Fig.2. The inversion height ($z_i$) as a function of time has also been added in Fig. 2.

7. Figure 3: Line 857 'shades represent 10% and 90% value'. The shading area represents a range.

The revised manuscript is: *Solid lines and shades represent median values and ranges from 10 to 90 percentiles.*

8. Figure 6: Revise the panel label and specify which panel corresponds to which case.

The labels have been revised in the previous Fig. 6 (now Fig. 8), and the cases have been correlated to the labels.

[Figure]

***Figure 8. The 7-days backward-dispersion of three cases from NAME simulations. Panels a-c) are column-integrated horizontal footprints for a) Case 1, b) Case 2 and c) Case 3. Panels d-f) are dispersion results attributed to FT for d) Case 1, e) Case 2 and f) Case 3. Panels g-i) are dispersion results attributed to BL for g) Case 1, h) Case 2 and i) Case 3. All plots are shown in the same colour scale. The black boxes represent the release locations of Ascension Island. The red boxes in Fig. e and g represent the horizontal grids (covering the area of 20°S – 0°N, 15°W – 12°E) used for integrating within each vertical layer to derive the vertical distribution of dispersed air parcels in Fig. 9.***

9.  Figure 7: The legend for cases should come earlier in panel (c). The two AOT for y-axis in panel (c) and (d) should be differentiated (further specified).

    The legend for cases has been moved to panel (c). The two AOT for y-axis in panel (c) and (d) have been differentiated ($AOT_{weighted-FT}$ and $AOT_{weighted-BL}$).

10. Figure 8: X-axis label is missing.

    The label of x-axis has been added in the previous Fig. 8 (now Fig. 10).

[revised manuscript text omitted]

---

## Author Comment (AC2)

**Reply to the Review of Manuscript EGUSphere-2024-3975:**

We would like to sincerely thank the editor and reviewer for their time, effort, and thoughtful feedback on our manuscript. The reviewer comments are shown in **blue**, with the authors' responses shown in **black** and any edited manuscript language shown in ***italicized black font***.

Review of Wu et al., 2025: The transport history of African biomass burning aerosols arriving in the remote Southeast Atlantic marine boundary layer and their impacts on cloud properties. This paper combines in-situ data from the CLARIFY-2017 campaign to satellite data and backtrajectory analyses to investigate aerosol-low-level cloud interactions around the Ascencion Island in the remote South-East Atlantic. The authors show that cloud droplet number concentrations are increased, and droplet radii are decreased by biomass burning aerosols, especially if the aerosols are in the sub-cloud layer. This agrees with published literature that highlights the important role of the respective positions of aerosol and cloud layers for ACI in this region, even though the exact functional relationship between aerosol and cloud properties seem to differ compared to other biomass burning regions. The authors then carry out backtrajectory analyses to estimate the exact timing of FT air entrainment into the marine boundary layer for 3 different case studies and identify efficient mixing regions 2-3 days upwind of the Ascencion Island, which could be used to constrain climate models that struggle to accurately simulate ACI in the SEA. I think these results are interesting and can be published after major revisions.

**Major comments:**

1. The novelty of the paper is not immediately obvious to me, but I believe this could be fixed with some reorganization of its content.

   1) - Many of the figures in the first half of the paper are reminiscent of the figures in Haywood et al., 2021. Since there are a lot of figures in the supplementary material, maybe it would make sense to remove redundant figures and instead move some of the supplementary material to the main text.

   We thank the reviewer for this suggestion. While Figures 1 (flight tracks) and 2 (aerosol vertical distributions) may appear similar to Figures 7 and 9a in Haywood et al. (2021), they serve a distinct purpose in our study. These panels provide essential context—defining the sampling region and summarizing biomass burning (BB) pollution conditions throughout the campaign—which underpin subsequent analyses. Therefore, we have retained Figures 1 and 2 in the revised manuscript.

   Regarding this suggestion:

   a. We have added more information on Fig.2. Using backward dispersion simulations, we calculated fractional contributions of airmasses from the African continent (20°S – 5°N, 9°W – 35°E) arriving at Ascension Island as a function of time throughout the campaign period, differentiating between FT (Fig. 2b) and MBL (Fig. 2c) simulations. This is to evidence that the observed pollution events around Ascension Island originated from long-range transported African BB plumes.

   b. We have moved the original Figs. S1 and S9 in the supplementary material to the main text.

   2) - I also wonder whether the instrument descriptions have previously been published in data papers, in which case it might not be necessary to have such a long method section.

Thanks to the reviewer for this suggestion. Original Sections 2.1 and 2.2 have been combined into one section "Section 2.1 Aerosol and cloud measurements". We have made the instrument descriptions more succinct.

3) Since it has been a few years since the CLARIFY 2017 campaign, it might be helpful to summarize for the reader the findings that have already been published using the campaign data, and mention questions that have yet to be answered, including the one(s) this study wants to address, which would make the novelty obvious.

Thanks to the reviewer's suggestions on making novelty obvious. The authors have briefly summarized the related findings that have already been published using the campaign data and expanded on the novel research questions this study will address. The revised manuscript is as follows:

*To address the aforementioned issues, aircraft in-situ measurements are essential to provide unique constraints on the vertical distribution of transported African BB aerosols over the SEA for climate models. Aircraft observations with continuous vertical sampling are also the most reliable source for accurately characterizing the correlations between aerosols and clouds. This study presents airborne observations of BB aerosols and clouds collected during the CLARIFY campaign (CLoudAerosol-Radiation Interactions and Forcing; August–September 2017), which was based around Ascension Island (7.96° S, 14.35° W) (Haywood et al., 2021). **The CLARIFY campaign addresses a key observational gap over the remote SEA, contributing to a more comprehensive understanding of BB aerosol vertical distributions and interactions with clouds across the SEA.** When integrated with complementary campaigns such as ORACLES (ObseRvations of Aerosols above CLouds and their intEractionS), which primarily focused on regions westward of the African continent and eastward of 0° E (Redemann et al., 2021), **the CLARIFY campaign provides an integrated wide-scale assessment of BB aerosol transport and its impact on cloud microphysics over the SEA.** Observations from these collaborative projects have indicated that the physicochemical properties of BB aerosols continuously evolve during weeklong transport, influenced by aging, cloud processing, and MBL environments, implying different CCN activity after long-range transport (Wu et al., 2020; Dobracki et al., 2023). Additionally, CLARIFY observations have shown that there is often a complex vertical structure of BB aerosol relative to cloud layers, as well as vertical variability of aerosol chemical composition and size distributions (Wu et al., 2020; Haywood et al., 2021). **These complexities underscore the significance of investigating the mechanisms by which long-range transported BB aerosols modulate cloud microphysics over the remote SEA.***

*In this study, we **first present the pollution conditions over Ascension Island and trace the origins of air masses using backward-dispersion simulations** (Sect. 3.1). We then **characterize the vertical profiles of thermodynamic variables, aerosol properties, and cloud microphysics over Ascension Island** (Sect. 3.2). While the region is influenced by the transition from stratocumulus to cumulus clouds, associated with increasing sea surface temperatures (Gordon et al., 2018), this study **focuses on assessing the effects of BB aerosols on stratocumulus cloud microphysics** (Sect. 3.3). More details of cloud types (stratocumulus or cumulus clouds) collected during the CLARIFY are provided in Sect. 3.2. Finally, we integrate air parcel analysis with satellite observations (Spinning Enhanced Visible and Infrared Imager, SEVIRI), **to identify the efficient entrainment regions where FT air parcels from Africa are likely to enter the MBL over the SEA and to demonstrate their impact on cloud properties along transport** (Sect.3.4).*

2.  The paper could be more succinct if the authors removed some redundancy in the explanations. For example, there are long paragraphs dedicated to reviewing the literature in the results section, but the relevance of the cited studies to this study might be more convincing if summarized into 1 or 2 sentences only. - It might also be helpful to make the abstract more succinct.

Thanks to the reviewer for this suggestion. We have streamlined the literature review in the Results section to improve clarity and conciseness. We have also revised the abstract, following the guidelines provided by ACP. The revised abstract is:

*African biomass burning (BB) aerosols transported over the southeast Atlantic (SEA) strongly influence cloud properties but remain a major source of uncertainty in regional climate assessment. This study characterizes vertical profiles of thermodynamic conditions, aerosol properties, and cloud microphysics around Ascension Island during an aircraft campaign (August–September 2017). Backward-dispersion simulations evidence that observed pollution originated from long-range transported African BB plumes. In BB-polluted marine boundary layers (MBL), aerosol number concentrations ($N_a$) were substantially elevated relative to the clean MBL, driving increased cloud droplet number concentrations ($N_d$) and reduced cloud effective radii ($R_e$). Cloud-layer mean $N_d$ correlated strongly with aerosols below the cloud (sub-$N_a$) but weakly with free-tropospheric (FT) aerosols. Enhanced sub-$N_a$ was due to BB aerosols entrained from the FT into the MBL along long-range transport and/or locally. Droplet activation fractions were similar in clean and moderately BB-polluted (sub-$N_a < 700\ cm^{-3}$) clouds, while a weaker $N_d$-$N_a$ correlation was observed in more polluted clouds. Region-specific $N_d$-$N_a$ parameterizations are necessary for representing BB aerosol-cloud interactions over the remote SEA. A robust inverse $N_d$-$R_e$ relationship was observed, regardless of BB influence. By coupling backward simulations with satellite retrievals, this study indicates that FT-to-MBL entrainment of African BB aerosols over the SEA occurs several days before arrival at Ascension Island, predominantly west of 0° E for examined cases. These findings provide unique observational constraints for representing aerosol-cloud interactions and vertical transport of Arican BB aerosols in climate models, offering improved assessments of African BB impacts over the SEA.*

3.  On Ln 540, you mention that your study has important implications for the radiative effects of aerosols in the region, but have you looked at how TOA radiative fluxes from SEVIRI change along the study period, depending on the aerosol load and location?

We thank the reviewer for this insightful comment. We agree with the reviewer that analysis of TOA radiative fluxes (e.g., from SEVIRI) can provide valuable information on the radiative effects of aerosols in this region. The evolution of TOA radiative fluxes during the study period can reflect the combined direct and indirect radiative effects of transported African BB aerosols over the remote SEA but is also affected by the cloud optical thickness that is largely influenced by meteorology.

This study focuses specifically on the transport process of African BB aerosols and quantifying their impacts on cloud microphysical properties over the SEA. The extension of the work to a consideration of the TOA radiative fluxes would add further complexity to the paper and hence is not carried out here. These results establish the foundation needed for subsequent investigations of their indirect radiative forcing. Accordingly, planned follow-on work from the project

will integrate our aerosol–cloud interaction parameterizations developed here to evaluate the TOA radiative impacts of transported BB aerosols in this region.

To address the broader implications, the revised manuscript is:

*Additionally, the modification of the cloud fields by high BB aerosol loadings, imply the important indirect radiative effects of transported African BB aerosols over the SEA. Future studies that integrate aerosol–cloud interaction parameterizations developed in this study will be conducted to improve the assessment of aerosol indirect radiative effects in this region.*

4. Section 4.2: In my opinion, this section needs to be rewritten.
   - I am concerned about the AOT analysis. Since AOT is a vertically integrated value, the analysis of AOT associated to either MBL or FT air parcels is misleading. For instance, an increase in AOT for the MBL air parcels could simply mean that the MBL is located under a polluted FT, but the MBL itself could still be pristine (and vice versa, for a clean FT over a polluted MBL). Could you address this caveat more explicitly?

   We agree with the reviewer that AOT represents a vertically integrated quantity. In this study, the SEVIRI-retrieved AOT is integrated only for the column above the cloud, as the MBL is cloud-filled. Thus, it does not provide information about aerosol presence within the boundary layer. Our analysis focuses on assessing whether the above-cloud AOT is co-located with regions of efficient entrainment as identified by NAME simulations, and thereby whether aerosols are present in the FT within these efficient entrainment regions.

   In the manuscript, we first highlight that the above-cloud AOT co-located with FT air parcels remained mostly high during efficient mixing periods indicated by NAME simulations (Cases 1 and 3). This indicates that FT BB aerosols existed over the efficient mixing area. BB aerosols may entrain from the FT into the MBL, if the bottom of FT BB layer is near the cloud top. Then, we highlight that the above-cloud AOT co-located with BL air parcels was enhanced when BL air parcels approached the efficient mixing area of FT air parcels (west of 0°E). This indicates that once FT BB aerosols could entrain into the MBL over efficient mixing periods, they could be subsequently advected by the MBL south-easterly winds toward Ascension Island area. Since the above-cloud AOT is a column integrated abundance and the aerosol layer may be vertically separated from the cloud top, this analysis does not unequivocally demonstrate entrainment of BB aerosols at these locations. Nevertheless, it is a necessary condition that elevated FT aerosol abudance is co-located with regions of efficient entrainment for the MBL to receive inputs of BB aerosols. Meteorological analysis of the SEA in August 2017 supports the connection between FT BB layers and the cloud top (Ryoo et al., 2022). The low-level easterly jet in early and late August allowed the transport of African BB plumes in the low FT and thus the possibility that BB aerosols were entrained into the MBL in efficient mixing regions.

5. Ln 490-510 is a very lengthy explanation mixing literature review and some interpretations. Some of the literature review could be moved to the introduction, which would allow the interpretations to stand out more convincingly.

   We thank the reviewer for this suggestion. We have made the explanation succinct regarding literature. The revised manuscript is:

   *Figs. 9c and 9d show the AOT_{weighted} along the FT and BL air parcel transport pathways respectively, for the three case*

*studies. High AOT_weighted values show the co-located and co-temporaneous abundance of African BB aerosols within the FT column above the cloud layer along the simulated transport pathway. For BB-impacted MBL cases (Cases 1 and 3), the AOT co-located with FT air parcels (black lines in Fig. 9c) remained mostly high during the periods of efficient mixing indicated by the NAME simulations. This indicates that FT BB aerosols existed over the efficient mixing area and so could be entrained from the FT into the MBL if the bottom of FT BB layer is near the cloud top. Along BL transport from southeast to northwest over the SEA, the AOT co-located with BL air parcels (black lines in Fig. 9d) was enhanced when approaching the region of efficient mixing of FT air parcels (west of 0°E). This further indicates that once FT BB aerosols could entrain into the MBL over efficient mixing periods, they could be subsequently advected by the MBL south-easterly winds toward Ascension Island area. Since the above-cloud AOT is a column integrated abundance and the aerosol layer may be vertically separated from the cloud top, this analysis does not unequivocally demonstrate entrainment of BB aerosols at these locations. Nevertheless, it is a necessary condition that elevated FT aerosol abudance is co-located with regions of efficient entrainment for the MBL to receive inputs of BB aerosols.* **Meteorological analysis of the SEA in August 2017 supports the connection between FT BB layers and the cloud top (Ryoo et al., 2022). The low-level easterly jet in early and late August allowed the transport of African BB plumes in the low FT and thus the possibility of BB aerosols entrainment into the MBL in efficient mixing area.** *In comparison, the AOT in the clean-MBL case (Case 2) was continuously low along both FT and BL transport (red lines in Fig. 9c and 9d), demonstrating a negligible contribution of BB pollution to the MBL over Ascension Island.* **This is due to a strong mid-level easterly jet in mid-August (Ryoo et al., 2022), leading to a disconnection between FT BB layers and the BL top, thereby suppressing the entrainment of African BB aerosols into the MBL.**

**Minor comments:**

- some choices of words are a bit vague and confusing, for instance. For example, on Ln 28: « a greater variability was noted in more polluted clouds» is imprecise. Same on Ln 352 (« a greater variation» ). Do you mean that the linear fit is weaker? Another example on Ln 295: « Larger ranges of Nd and LWC values … a smaller range of Re ». Do you mean that Nd and LWC values are larger on average and Re values are smaller on average, or do you mean that there is a larger/smaller standard deviation in the observed distributions (to me, « range » suggests the latter). There are several other occasions where « range » is used in this way in the text (e.g., Ln 404, 572). Could you use more precise mathematical language to help readers better understand your point?

We thank the reviewer for the suggestions. We have rephrased the related descriptions to make the manuscript more precise. The revised manuscript includes e.g.

*Droplet activation fractions were similar in clean and moderately BB-polluted (sub-$N_a$ < 700 cm$^{-3}$) clouds, while a weaker $N_d$-$N_a$ correlation was noted in more polluted clouds.*

*The relationship between $N_d$ and sub-$N_a$ follows a similar pattern in clean or moderately BB-impacted clouds (sub-$N_a$ < 700 cm$^{-3}$). The weaker correlation between $N_d$ and sub-$N_a$ under more polluted conditions (sub-$N_a$ > 700 cm$^{-3}$) may be partly due to the variability in the MBL updraft velocity.*

*In-situ and SEVIRI observations consistently indicate higher $N_d$ and smaller $R_e$ values in the BB-impacted MBL compared*

*with the clean MBL.*

*Cloud layers with negligible entrainment mixing (average LWC/aLWC values, ~ 1) present generally larger $R_e$ values compared to those with greater entrainment mixing (lower average LWC/aLWC, < 0.83).*

*This resulted in increased $N_d$ and LWC values but reduced $R_e$ within BB-impacted clouds compared to clean clouds.*

- Ln 50: affecting », not « affect ». The last part of the sentence (starting with « underscoring... ») sounds redundant and could probably be removed.

We have re-organized the introduction. The opening paragraph has been rewritten for conciseness and precision, and this requested sentence has been removed. The revised first paragraph now reads as follows:

*Every year from July to October, seasonal wildfires across central and southern Africa account for about one-third of global carbon emissions from biomass burning (BB) (Roberts et al., 2009). The emitted smoke is frequently transported westward over the southeast Atlantic (SEA), where it often resides in the free troposphere (FT) and may entrain into the marine boundary layer (MBL) during its subsiding transport (Painemal et al., 2014; Adebiyi and Zuidema, 2016; Das et al., 2017). These transported BB aerosols exert complex radiative effects by absorbing and reflecting solar radiations, and by interacting with one of the world's largest semi-permanent stratocumulus cloud decks over the SEA. The SEA region is climatically important, due to a significant net cooling effect of these stratocumulus clouds through their strong reflection of solar radiation but a small positive longwave radiative effect (Wood, 2012).*

- Ln 120: remove « within », correct « AfricaN continent »

The revised manuscript is:

*...which primarily focused on **regions westward of the African continent** and eastward of 0° E (Redemann et al., 2021)...*

- Ln 125: This study is obviously focused on stratocumulus clouds, and the study area is the Ascension Island, but traditionally, Sc are thought to form closer to the coast (e.g., areas defined by Klein and Hartmann, 1993), and then transition to cumulus clouds as they move westwards to Ascension Island, which might leave some readers confused. It would be good to address this. Since you actually already have a list of cloud types observed during the campaign in Table S1, it would be good to reference it in the main text.

Thanks to the reviewer's suggestions. More details regarding the suggestion have been added to the Introduction.

*In this study, we characterize the vertical profiles of thermodynamics, aerosol properties, and cloud microphysics over Ascension Island. While the region is influenced by the transition from stratocumulus to cumulus clouds, associated with increasing sea surface temperatures (Gordon et al., 2018), this study focuses on assessing the effects of BB aerosols on stratocumulus cloud microphysics. More details of cloud types (stratocumulus or cumulus clouds) collected during the CLARIFY are provided in Sect. 3.2.*

- Ln 138: «straight and level runs » not defined

We have added example flight patterns in Figure S1, to define "straight and level runs".

[Figure]

***Figure S1. Example flight patterns on flight C042 (top) and C032 (bottom). The example flights provide an illustration of straight and level runs for aerosol characterization and saw-tooth and stepped profiles for cloud samplings.***

- Ln 142: why would a POC event not be relevant to this study?

We have rephrased this part based on the suggestion.

*Transit flights (C040-C041, predominately at high altitudes) and flights with mainly limited cloud samplings are not included in this study. Specific events (pocket of open cell, C052-C054) have been characterized in Abel et al. (2020), which are also excluded from this study.*

- Ln 145: as written above, the instrument description could be shortened if the material is already published somewhere else. What would be interesting to add though is the typical size range of BBAs so that the readers can easily assess the suitability of the measurements.

Thanks to the reviewer for this suggestion. We have made the instrument descriptions more succinct.

The typical size range of BB aerosols has been added to the revised manuscript:

*These aerosol instruments covered the typical size range of BB aerosols, primarily within the submicron range.*

- Ln 182-185: the difference between the 0.1 g/m³ LWC and 0.2 g/m³ 'bulk' LWC thresholds was not clear to me

An LWC value over 0.01 g m$^{-3}$ for 1 Hz measurements was used to define the low threshold for the presence of cloud.

Yes, there are minor differences between 0.1 g/m³ and 0.2 g/m³ of LWC. However, following previous studies (e.g. Lance et al., 2010; Bretherton et al., 2010), to eliminate the inclusion of optically thin clouds, a threshold of $N_d > 5$ cm$^{-3}$ and bulk LWC > 0.02 g m$^{-3}$ was used to perform statistically robust cloud sample analysis.

- Ln 190: (CTO) instead of (COT) - Ln 194: I believe there is an error in the equation, it should be COT1/2 and not COT-1/2

Thanks to the reviewer for pointing out the typo. The revised manuscript is:

*We also obtained the above-cloud aerosol optical thickness (AOT), cloud optical thickness (COT), $R_e$ and **cloud-top height (CTH)** across the SEA region (20° W − 15° W; 30° S − 0° N) ...*

*The $N_d$ was calculated assuming an adiabatic-like vertical stratification (Painemal et al., 2012):*

$$N_d = 1.4067 \times 10^{-6} \left[ cm^{-\frac{1}{2}} \right] COT^{\frac{1}{2}} R_e^{-\frac{5}{2}} \qquad (4)$$

- Ln 196: Did you check the quality of your SEVIRI products close to sunrise and sunset times? There might be biases around those time, this could impact your results if sunrise/sunset points are colocated to the backtrajectories.

Yes, we have checked the quality of SEVIRI products close to UTC sunrise and sunset times. In addition, some of sunrise and sunset SEVIRI products were missing or unreliable when co-located with simulated transport, which were not included in the analysis.

- Ln 197: The NAME description section got me a bit confused. I did not understand why there seems to be 2 sets of backtrajectories, one at 3-hr resolution, and the second one at 15-min resolution. What is the point of having both sets of trajectories? Why use NAME backtrajectories vs. using the already produced HYSPLIT ones from the Haywood et al., 2021 paper?

Backward-dispersion and back-trajectory simulations both trace air parcel but differ fundamentally in how they treat transport and mixing:

|  | Back-trajectory | **Backward-dispersion (what we did)** |
|---|---|---|
| Purpose | Identify the origin and core transport path of air parcels arriving at a receptor | source region contributions to observed concentrations |
| Output | Trajectory lines | "footprint" fields showing where parcels most likely originated |
| Process | Advection only | Turbulence, deposition, chemistry |

In this study, we conducted backward-dispersion simulations. To investigate vertical distributions and the transport of original air parcels, the model output instantaneous 3D air parcel footprints **every 3h** during the 7-days backward dispersion simulations. In the original manuscript, we also conducted Back-trajectory simulations (output resolution = 15min) to examine the entrainment rates. To avoid misleading the readers, we have deleted the back-trajectory simulations in the revised manuscript and focus on the dispersion results.

Haywood et al. (2021) employed HYSPLIT back trajectories to identify main source regions. However, this approach is not able to answer the vertical transport and mixing processes in this study, which is captured by our NAME backward-dispersion simulations.

- Ln 230: here you introduce one definition for the inversion height, but later (Ln 246) you use a different one, why is that?

Thanks to the reviewer for pointing out the repeated definition and typo. To avoid misleading the readers, more details

have been added to the manuscript. The revised parts in Sects. 2.2 and 3.1 are:

Sect. 2.2: *Here, we estimated the $z_i$ over the SEA, using the outputs of 3D meteorological parameters. The $z_i$ was quantified as the height at which the vertical gradient of liquid water potential temperature ($\theta_l$) is the largest, and there is also a steep decrease in humidity (Jones et al., 2011). The $\theta_l$ was estimated following Eq. (5).*

$$\theta_l = \theta - \frac{L}{c_p} \, q_l \tag{5}$$

*Where $\theta$ is the potential temperature, L is the latent heat of vaporization for water ($2.5\times10^6$ J kg$^{-1}$), $c_p$ is the specific heat of dry air at constant pressure (1005 J kg$^{-1}$ K$^{-1}$), $q_l$ is the liquid water mixing ratio (g kg$^{-1}$).*

Sect. 3.1: *Figure 2a shows complex vertical distributions of aerosol number concentrations ($N_a$, 0.1 – 3 µm) from PCASP measurements for the CLARIFY flights used in this study, alongside the estimated $z_i$. The $z_i$ estimates were derived from airborne measurements of $\theta$ and $q_l$, using Eq. (5).*

-Ln 263: The FT humidity comparison between clean and BB-impacted FT cannot be inferred from Figure 3, can it?

-Ln 265: what is the interpretation for this positive correlation between BBA and humidity in the FT?

To differentiate the clean and BB-impacted FT, the vertical profiles of $q_t$ (Fig. 3b) are now colored by three periods. In addition, more details about the positive correlation between BB plumes and humidity in the FT have been added to the manuscript:

Regarding the above two comments, the revised version is:

*Figure 3b shows that FT humidity was generally higher on days with the presence of FT BB plumes (Periods 2 and 3, red and black profiles) compared to clean FT cases (Period 1, blue profiles). This pattern is consistent with previous studies, suggesting a positive correlation between BB plume strength and atmospheric water vapor content in the FT over the SEA (Pistone et al., 2021). The covariation between plume strength and humidity is attributed to the mixing between the moist, smoky continental boundary layer and the dry, clean FT. Consequently, humid BB plumes above the boundary layer can be advected to the SEA, enhancing FT humidity in the presence of transported smoke plumes (Pistone et al., 2021).*

[Figure]

*Figure 3. a, b) Average vertical profiles of a) liquid water potential temperature ($\theta_l$, K), b) total water mixing ratio ($q_t$) for each flight used in this study. Blue, red and black lines represent measurements from Periods 1, 2 and 3 respectively. c-e) Summarized profiles of c) aerosol count median diameter derived from the PCASP (CMD, μm) and d,e) the ratio of CCN (~ 0.2%) to condensation nuclei ($D_a > 3$ nm from the CPC) (CCN/CN$_3$) and accumulation aerosol concentration ($D_a > 0.1$ μm from the PCASP) (CCN/Na) under polluted (black) and clean (red) conditions. Solid lines and shades represent median values and range from 10 to 90 percentiles. The red cross marker in Fig. 3c represents the average CMD in accumulation mode from measurements within the clean MBL. f-h) Vertical profiles of 1-hz f) cloud droplet number concentration ($N_d$, # cm$^{-3}$), g) liquid water content (LWC, g m$^{-3}$) and h) cloud effective radius ($R_e$, μm) in sampled continuous cloud layers. Red lines represent cloud measurements in the clean MBL, and grey lines represent cloud measurements in the BB-impacted MBL. It is noted that average vertical profiles of cloud properties from flight C032 are also provided in Figs. 3f-h (grey dashed lines). The y-axis uses a height scale normalized by inversion height ($z_i$).*

- Ln 269: Could the MBL CMD simply be lower because the BBAs are mixed with smaller aerosols (e.g. sulfate, sea salt, etc.) in the MBL but not in the FT?

Thanks to the reviewer for the suggestion. The revised manuscript is:

*Figure. 3c shows that the CMD of aerosols in the BB-impacted MBL was 10 – 15 % lower than in the FT BB plumes, which is probably attributable to some processes occurring in the MBL such as **mixing with marine emitted small particles** (i.e. marine sulfate) and aerosol removal by drizzle (Wu et al., 2020).*

- Ln 279: CCN/CN3 and not Na/CN3

Here, we want to show that the majority of aerosols detected in the clean MBL were Aitken mode particles in the size

below 0.1 μm. Thus, it is **the ratio of $N_a/CN_3$** rather than $CCN/CN_3$.

To make the description clearer, the revised manuscript is:

*The aerosols in the clean MBL were smaller than in the BB-impacted MBL during CLARIFY. The average ratio of $N_a/CN_3$ in the clean MBL was calculated to be $0.40 \pm 0.26$, also suggesting that the majority of submicron aerosols detected by the CPC (3 nm to 1 μm) in the clean MBL were Aitken mode particles (< 0.1 μm).*

- Ln 288-291: as per my comment above, it would be helpful to already include a description of observed cloud types during CLARIFY in the introduction, to contextualize the method and results.

Thanks to the reviewer's suggestions. More details regarding the suggestion have been added to the Introduction.

*While the region is influenced by the transition from stratocumulus to cumulus clouds, associated with increasing sea surface temperatures (Gordon et al., 2018), this study focuses on assessing the effects of BB aerosols on stratocumulus cloud microphysics. More details of cloud types (stratocumulus or cumulus clouds) collected during the CLARIFY are provided in Sect. 3.2.*

- Ln 292: near *cloud* top (at the end of the line)

The revised manuscript is:

*The profiles of $N_d$ (Fig. 3f) remain fairly constant, with few cases in the BB-impacted MBL (i.e. profiles in flights C029 and C038) presenting an increase **near cloud top**.*

- Ln 321: what does a « «relatively » negative correlation mean? Relative to what?

The revised manuscript is:

*An **overall** negative correlation was observed between sub-$N_a$ and CTtoAB.*

- Ln 345-346: could you provide the reference for these numbers again?

The Figures have been referred to for these numbers.

*During CLARIFY, the BB-impacted MBL had substantially enhanced sub-$N_a$ ($212 – 1183$ $cm^{-3}$, black markers in Fig. 5a) compared to the clean MBL ($56 – 315$ $cm^{-3}$, red markers in Fig. 5a).*

- Ln 347: maybe « indicated » could be replaced by another verb, like « hypothesized »

The "indicated" has been replaced by "hypothesized".

- Ln 353: remove « recent »

The "recent" has been deleted.

- Ln 361: how were these contact profiles selected?

To make the description clearer, the revised manuscript is:

*When contact profiles had average LWC/aLWC values close to 1, this suggests near-adiabatic profiles with negligible*

*mixing between cloudy and non-cloudy air. ... Notably, some of the near-adiabatic profiles (blue dashed box highlighted in Fig. 5a) displayed lower droplet activation fractions at similar sub-$N_a$ levels, compared to other profiles.*

To make the description clearer, the revised manuscript is:

*Notably, some of the near-adiabatic profiles (blue dashed box highlighted in Fig. 5a) displayed lower droplet activation fractions at similar sub-$N_a$ levels, compared to other profiles.*

- Ln 365: just a note that entrainment of warmer and drier FT air could also lead to cloud droplet evaporation and decreases in Nd (cases of extreme inhomogeneous mixing, see Hill et al., 2009)

Thanks to the reviewer for the insightful comment. Yes, we agree that entrainment of warmer and drier FT air could also lead to cloud droplet evaporation and may decrease $N_d$. However, our observations suggest that contact profiles with greater entrainment generally promoted additional droplet nucleation and eventually enhanced $N_d$, compared to near-adiabatic profiles. We have added the reviewer's suggestion to the revised manuscript:

*When contact profiles had average LWC/aLWC values close to 1, this suggests near-adiabatic profiles with negligible mixing between cloudy and non-cloudy air. In contrast, other contact profiles had lower average LWC/aLWC values (0.34 – 0.83), indicating greater entrainment mixing of aerosols from above-cloud into the cloud layer at the place of observation. Notably, some of the near-adiabatic profiles (blue dashed box highlighted in Fig. 5a) displayed lower droplet activation fractions at similar sub-$N_a$ levels, compared to other profiles. **Although cloud-top entrainment of warmer, drier FT air could cause cloud droplet evaporation, our observations suggest that contact profiles with greater entrainment generally promoted additional droplet activation and enhanced $N_d$, compared to near-adiabatic profiles.***

- Ln 371: Can you quantify the goodness of fit (for instance with the R2 for the linear regression between logNa and logNd)?
- Ln 372/373: from the confidence intervals, it looks like the fit is quite uncertain. How should these parameters be used if there are so uncertain ?

Thanks to the reviewer for the above two suggestions. The positive correlation between $N_d$ and sub-$N_a$ was strong ($r = 0.93$, $p < 0.01$) in clean or moderately BB-impacted clouds (sub-$N_a < 700$ cm$^{-3}$), while a weaker correlation was observed in more polluted clouds (sub-$N_a > 700$ cm$^{-3}$). The relationship between $N_d$ and sub-$N_a$ follows a similar pattern in clean or moderately BB-impacted clouds, which is regarded as an aerosol-limited regime. Following the reviewer's suggestions, we quantified the power law fit ($N_d \sim \alpha N_a^\beta$, with tight confidence intervals) and its goodness in aerosol-limited regime, for application in future studies.

*Since droplet activation displayed similar behavior in clean and moderately BB-polluted clouds (sub-$N_a < 700$ cm$^{-3}$) (Fig. 5a), a strong linear correlation ($r^2 = 0.87$) was observed between log($N_d$) and log(sub-$N_a$). Data in this aerosol-limited droplet activation regime (sub-$N_a < 700$ cm$^{-3}$), yielded $\alpha = 0.64$ (0.20 – 1.28) and $\beta = 0.93$ (0.79 – 1.04), which we recommend as representative parameters for application in future studies.*

- Ln 382: Do you have a more detailed hypothesis of why SEA BBAs might have a better CCN ability compared to other

regions? (based on the literature?)

Regarding the comment, more discussions of CCN ability between different BB regions have been added to the manuscript. The revised manuscript is:

*Our observations suggest a small difference in the response of $N_d$ to $N_a$ between BB-impacted and clean MBL profiles. This is likely due to their comparable CCN activation abilities of accumulation-mode aerosols under two MBL conditions (as discussed in Sect 3.2). In contrast, a previous study reported higher droplet activation fractions for the cleaner MBL compared to the BB-impacted MBL over the Pacific Ocean (Mardi et al., 2019). The discrepancy likely stems from differences in their droplet activation behaviors of transported BB aerosols between the studies. The $\beta$ value for BB-impacted MBL cases in this study (0.71 (0.42 – 0.92)) is in a higher range than that reported for BB-impacted areas off the California coast of North America (0.26 (0.15 – 0.42)) in Mardi et al. (2019). A key factor contributing to the different droplet activation behaviors of transported BB aerosols, may be the variability in aerosol chemical composition and CCN activity, which depends on source combustion conditions and aging process (Wu et al., 2020; Farley et al., 2025). Submicron BB aerosols from western U.S. wildfires have been reported to be dominated by organic (~90%) with minimal inorganic content (<2%) from near-source to regional scales (0.5 hours to several days) (Farley et al., 2025). In contrast, highly aged African BB aerosols were reported to contain ~35% inorganic mass (Wu et al., 2020). This implies that highly aged African BB aerosols are more hygroscopic, as inorganics typically have higher hygroscopicity than organics on a global scale (Pöhlker et al., 2023). Additionally, transported BB aerosols from western US wildfire presented similar accumulation-mode aerosol size distributions to this study (Laing et al., 2016). Consequently, the CCN activation ability of transported African BB aerosols in this study is broadly higher than that reported for aged BB aerosols from Western/Northern American wildfires (CCN/CN = 0.11 – 0.62, at SS = 0.2 – 0.5%) (Pratt et al., 2011; Zheng et al., 2020), leading to the observed differences in their $N_a$-$N_d$ relationship between these studies.*

- Ln 410: in this paragraph, what can the given fit be interpreted, for instance does it tell us anything about the adiabaticity of the sampled clouds?

Thanks to the reviewer for the insightful comment. More discussions have been added to the manuscript:

*Figure 7b shows the relationship between cloud-layer mean $R_e$ versus the average ratio of LWC/$N_d$ for all analyzed profiles. BB-impacted and clean MBL profiles yielded similar exponent (b) values of (0.34 ± 0.01) and (0.33 ± 0.01) respectively, which are close to the exponent (~0.33) validated in previous BB aerosol-cloud studies such as in Amazon (e.g. Reid et al., 1999). Previous studies have reported a theoretical exponent of b =1/3 for adiabatic clouds, where droplet growth is dominated by condensation without entrainment (Burnet and Brenguier, 2007). In this study, the estimated ratios of LWC/aLWC (~ 0.3 – 1, Fig. S5c) suggest the occurrence of entrainment mixing, nevertheless, the empirical b values near or slightly exceed ~1/3. This indicates that entrainment processes were likely dominated by homogeneous mixing, which could proportionally reduce LWC and $N_d$ and thus preserve b~1/3, particularly in clean cases (Burnet and Brenguier, 2007). In BB-impacted cases, entrainment of aerosol-rich air may also supply additional CCN, increasing $N_d$ disproportionately to LWC reduction. This entrainment of additional CCN could yield cases with b>1/3 in this study.*

- Ln 428: maybe « linked » can be replaced with « co-located »

The "linked" has been replaced with "co-located".

- Ln 450: in the methods, it is said that the trajectories are initiated from an altitude of 341m, yet on Fig. 7, it looks like they come from an altitude of 2 km, why is that?

As described in the Method section, for NAME MBL simulations, tracer particles were released within a height range in the MBL (341±300 m) over Ascension Island. The original Fig. 7 (now Fig. 9) displays only the vertical distributions of source air parcels **originating from the FT region** along the backward-simulation time. Therefore, the figure specifically illustrates the vertical contribution of original FT air parcels along transport, before arriving at release area. This is designed to study **vertical transport and evolution of original FT air parcels**, and then the exchange or entrainment from the FT to the MBL. If the vertical distributions of original MBL air parcels were included in the figure, the starting point at time = 0 would appear at an altitude of ~ (341±300 m).

- Ln 478: Have you explained what the air-density-weight transformation means for AOT?

The definition of air-density-weight transformation has been added to the manuscript. The revised manuscript is:

*The air-density-weighted AOT ($AOT_{weighted}$) was calculated for the co-located areas following Eg. (8).*

$$AOT_{weighted} = \sum_{i=1}^{N} \left( \frac{Mass_i}{\sum_{i=1}^{N} Mass_i} \times AOT_i \right) \qquad (8)$$

*Where N is the total number of horizontal air parcel grids for co-located areas, $Mass_i$ is air parcel concentration ($g\ m^{-3}$) in each grid, $\frac{Mass_i}{\sum_{i=1}^{N} Mass_i}$ represents fractional air parcel concentration in each grid relative to the total air parcel density, $AOT_i$ is the above-cloud AOT co-located with each air parcel grid.*

- Ln 527: "observed larger droplets" is mostly true, but at -24h on Fig. 8, the droplet sizes are decreasing. Why is that?

Thanks to the reviewer for this insightful observation. When approaching Ascension Island, the MBL deepens and the deeper MBL also tends to be decoupled (Abel et al., 2020). The decoupled MBL is likely associated with an increased occurrence of drizzle due to the presence of larger droplets (Jones et al., 2011), which may result in slightly decreased $N_d$ and $R_e$ when approaching Ascension Island. The revised manucript is:

*In Case 2 (red line in Fig. 10c), the CTH shows an increasing trend along the southeast to northwest transport path over the SEA. Concurrent MBL deepening and enhanced CTH could promote condensational growth, yielding larger droplets at the cloud top (Painemal et al., 2014). This agrees with the observed overall increase in $R_e$ along the BL transport.* **However, the deeper MBL also tends to be decoupled, and the decoupled MBL is likely associated with an increased occurrence of drizzle due to the presence of larger droplets (Jones et al., 2011). The occurrence of drizzle and associated deposition may result in reduced $N_d$ and $R_e$, corresponding to the observed slightly reduction in the vicinity of Ascension Island.**

- Ln 573: "a stronger relationship" could be made more precise with "a stronger linear correlation"

The "a stronger relationship" has been rephrase to "a strong correlation".

- Ln 576-577: Could it be interesting to add a quantification of relatively how much mass of aerosols is entrained along the transport vs at the place of observation? Or how much percent of the mass is entrained in the identified "efficient mixing regions"?

Thanks to the reviewer for this suggestion.

NAME simulations can provide the contribution and transport of original air parcels before arrival at the release location. However, it is not easy to distinguish the entrainment of aerosol at the place/time of observation, since the entrainment process is time dependent. Thus, we couldn't quantify the relative contributions of aerosols entrained along the transport vs. at the place of observation from NAME simulations. However, in Sect. 3.3, using aircraft datasets, we have implied the influence of entrained BB aerosols to sub-$N_a$ at the place of observation. In future work, it will be useful to quantify the relative contribution of long-range transport and local mixing.

In Fig. S8, we have provided cumulative exchange amounts of air parcels between the FT and the MBL along backward simulations, which could indicate the total entrained FT-to-MBL amounts.

- Ln 845: "Langitude" typo in the x-axis of Fig.1.

The typo has been corrected.

- Ln 845: why are the fire counts cumulated over August 2017 only, and not over the exact study period?

The fire counts have been cumulated over the campaign period, and Fig. 1 has been modified accordingly.

[Figure]

***Figure 1. Tracks of the CLARIFY flights used in this study, cloud sampling periods during tracks ($N_d > 5\ cm^{-3}$ and bulk LWC > 0.02 g m$^{-3}$) are highlighted in purple colour. Fire density maps are counted over the African continent, showing 2° 2° bins of the number of MODIS-detected fire during the campaign period.***

- Ln 860: the f) and g) labels have been swapped compared to what is shown in the figure's subplots

The labels have been revised.

*f-h) Vertical profiles of 1-hz f) cloud droplet number concentration ($N_d$, # cm$^{-3}$), g) liquid water content (LWC, g m$^{-3}$) and h) cloud effective radius ($R_e$, μm) in sampled continuous cloud layers.*

[revised manuscript text omitted]

---

## Referee Report (RR1)

**Review of Manuscript EGUSphere-2024-3975:**

**The transport history of African biomass burning aerosols arriving in the remote Southeast Atlantic marine boundary layer and their impacts on cloud properties, by Wu et al.**

**General comments:**

Wu et al. presented a study investigating the impacts of African biomass burning aerosols (BBAs) on liquid clouds over the southeast Atlantic region. This study combines both airborne observations on aerosol properties and cloud microphysic properties and numerical simulations for air mass sources and properties. It reveals that the presence of BBAs can increase the observed cloud droplet number but decrease cloud droplet size. In addition, this study evaluates the BBA-cloud interaction activities under both free troposphere and marine boundary layer conditions. Overall, the revised manuscript is improved compared with the original submission. If comments below can be addressed, we would like to suggest acceptance for publishing in ACP.

**Major comments:**

- 1. In addition to presenting NAME simulations for three cases (Figure S2) as representatives for three periods in Figure 2, the authors should provide one simulation result for each flight observation (e.g., releasing air parcels at the middle time of each flight). This will make the paper stronger. One would worry about whether the three cases presented in Figure S2 are the best cases. For example, one would wonder if one of the flight observations during period 2 with similar fractions of airmass from African continent to those of case 2 in Figure S2 also shows similar airmass back trajectories. In addition, rationales for selecting case 2 are not clearly provided in discussions on Figure 2. For example, it should be directly noted that it is a reference case as clean MBL for comparing with BBA polluted case 1 and 3.
- 2. This study focuses on the impacts of BBAs from Africa on SAE clouds. It is stated that 'the presence of BB-pollution was defined using thresholds of carbon monoxide above 83 ppb and black carbon above 0.1µg m-3' (Line 223-225 in Section 3). However, Figure 2 and Figure S2 present results about all airmasses from African continent which may include all kinds of continental pollutions. To clearly indicate the presence and influence of BBAs, the abundance of carbon monoxide and black carbon as a function of time should also be provided in Figure 2 or in the supplement material.
- 3. Relevant text statements for introducing each figure and table in the Supplementary document should be provided.

**Minor comments:**

- 1. In the caption of Figure 8 and Figure S2, it should be clarified that Figure 8 is based on the output of 3D air parcel footprints whereas Figure S2 is based on the method introduced in the second paragraph in section 2.2.
- 2. Line 47: to reduce the relative humidity
- 3. Better to also provide the flight numbers for flights with mainly limited cloud samplings that are not included in this study
- 4. Line 128-129 (and Line 133): PCASP is missing here for Passive Cavity Aerosol Spectrometer Probes
- 5. Line 145-146: the general size range of BBAs observed in this region should be provided here (or refer to the literature)
- 6. Line 161: Why 0.01 g m-3 is used as the threshold LWC? Any rationale or reference?
- 7. Line 201: what is BL? Only MBL was defined before this.
- 8. Line 236: isn't case 1 at the middle of period 1?
- 9. Figure 2: why do not choose case 2 and case 3 at a point where flight observations are available?
- 10. Line 248-149: it might be too arbitrarily to use three cases to conclude the pollution source of the whole campaign. It is necessary to provide back trajectory results for each flight observation. Same as major comment 1.
- 11. Line 276-278: please refer to Figure 3a. Also, please check through the manuscript to make sure relevant figures/tables are correctly and properly referred in the discussion. Some of in-text references are missing, which makes it a bit hassle to follow the flow.
- 12. Line 300: Also provide the value of the CCN activation fractions of these highly aged African BB aerosols
- 13. Line 304: A sentence for clear definition of clean MBL is missing in the manuscript. It should be already introduced in Section 1 or Section 2
- 14. Line 305: 'The aerosols in the clean MBL were smaller', is the number or size of aerosols small?
- 15. Line 312-316: based on the results, the two individual statements seem right. However, it is awkward when putting them together
- 16. Figure 3: provide legend in each panel. A missing legend makes it not convenient to read it
- 17. Line 406: ambiguous statement. Enhanced Nd value or larger Nd distribution range?
- 18. Line 406-407: Figure 5a shows what kind of effects? It should be clearly stated out what the figure/result show but not only say 'affect'
- 19. Figure 8: indicate the atmospheric condition and case number in the legend for each panel
- 20. Line 609-611: the continuous increasing of CTH cannot explain why there is a decrease in Re at -24 hours. Further details/results should be provided to explain this?

---

## Author Response (AR2)

**Reply to the Review of Manuscript EGUSphere-2024-3975:**

We would like to sincerely thank the editor and reviewer for their time, effort, and thoughtful feedback on our manuscript. The reviewer comments are shown in **blue**, with the authors' responses shown in **black** and any edited manuscript language shown in *italicized black font*.

**Reply to the Reviewer #1**

Wu et al. presented a study investigating the impacts of African biomass burning aerosols (BBAs) on liquid clouds over the southeast Atlantic region. This study combines both airborne observations on aerosol properties and cloud micro physic properties and numerical simulations for air mass sources and properties. It reveals that the presence of BBAs can increase the observed cloud droplet number but decrease cloud droplet size. In addition, this study evaluates the BBA-cloud interaction activities under both free troposphere and marine boundary layer conditions. Overall, the revised manuscript is improved compared with the original submission. If comments below can be addressed, we would like to suggest acceptance for publishing in ACP.

**Major comments:**

1. In addition to presenting NAME simulations for three cases (Figure S2) as representatives for three periods in Figure 2, the authors should provide one simulation result for each flight observation (e.g., releasing air parcels in the middle time of each flight). This will make the paper stronger. One would worry about whether the three cases presented in Figure S2 are the best cases. For example, one would wonder if one of the flight observations during period 2 with similar fractions of airmass from African continent to those of case 2 in Figure S2 also shows similar airmass back trajectories.

In addition, rationales for selecting case 2 are not clearly provided in discussions on Figure 2. For example, it should be directly noted that it is a reference case as clean MBL for comparing with BBA polluted case 1 and 3.

Thanks to the reviewer's suggestions.

1) As described in the method section (Sect.2.2), we performed backward-dispersion simulations at 3-hour (h) intervals throughout the campaign period. In each simulation, a certain amount of hypothetical tracer particles was released from a 2° × 2° grid box centered around the Ascension Island observation site (14.35°W, 7.96°S). To distinguish source origins of MBL and FT airmass over Ascension Island, we also performed simulations that released tracer particles within the MBL (341±300 m) and FT (2.5-4 km) altitude ranges separately. In the original Sect. 3.1, we first present three representative backward-dispersion fields (Cases 1 to 3), that illustrate the major transport pathways of air parcels during the three defined periods. To assess the contributions of polluted airmass originating from the African continent to the Ascension Island area, we calculated the fractional contributions of original airmass over the African domain (20° S-5° N, 9° W-35° E) based on the outputs of 3-hourly NAME back-dispersion fields. This continental domain is closely associated with BB pollutions identified in satellite wildfire observations (Figure 1). The resulting analysis provides a quantitative indication of BB plume influence on the Ascension Island area throughout the campaign period, as shown in Fig. 2b (FT simulations) and Fig. 2c (MBL simulations). Therefore, the conclusions regarding pollution sources are supported not only by the three representative examples of dispersion fields shown in the original Fig. S2, but also by the statistical analysis of the overall African source contributions presented in Fig. 2b (FT simulations) and Fig. 2c (MBL simulations). To address the reviewer's concern, we have further included back-dispersion fields corresponding to each flight observation, which are now provided in Figure S3-4 of the revised Supplementary. The corresponding description in Sect 3.1 has also been rephrased as below:

"Example backward-dispersion fields corresponding to release times within the flight sampling period are presented, distinguishing between FT (Fig. S3) and MBL (Fig. S4) simulations. FT dispersion simulations (Fig. S3) indicate that most FT air parcels over Ascension Island originated from westerly flow across the SEA and African continent. Compared to clean FT cases from Period 1, polluted FT cases from Periods 2 and 3 show a substantially greater influence from African continental airmasses. MBL dispersion simulations (Fig. S4) suggest that MBL air parcels over Ascension Island mainly arose from clean oceanic flow that transported from the southeast to northwest over the SEA. Polluted MBL cases from Periods 1 and 3 also show contributions of westerly flow originated from the African continent, while negligible continental influence in clean MBL cases from Period 2. To assess the contributions of polluted airmass originating from the African continent to the Ascension Island area, we calculated the fractional contributions of original airmass over the African domain (20° S-5° N, 9° W-35° E) based on the outputs of 3-hourly NAME back-dispersion fields. This continental domain is closely associated with BB pollutions identified in satellite wildfire observations (Fig. 1). The resulting analysis provides a quantitative indication of BB plume influence on the Ascension Island area throughout the campaign period, as shown in Fig. 2b (FT simulations) and Fig. 2c (MBL simulations). The temporal evolution of continental contributions aligns well with the observed FT and MBL pollution patterns over Ascension Island. NAME simulations evidence that the FT and MBL pollutions observed during the campaign were attributable to long-range transport of African continental airmass which brought BB plumes."

- 2) We have added in the manuscript: "Case 2 is a reference clean-MBL case for comparison with BB-polluted MBL cases (Cases 1 and 3).".
- 2. This study focuses on the impacts of BBAs from Africa on SAE clouds. It is stated that 'the presence of BB-pollution was defined using thresholds of carbon monoxide above 83 ppb and black carbon above 0.1µg m-3 (Line 223-225 in Section 3). However, Figure 2 and Figure S2 present results about all airmasses from African continent which may include all kinds of continental pollutions. To clearly indicate the presence and influence of BBAs, the abundance of carbon monoxide and black carbon as a function of time should also be provided in Figure 2 or in the supplement material.

Thanks to the reviewer's suggestion.

We have added Figure S2 in the supplement material, including the measurements of carbon monoxide (CO) and black carbon (BC) from the CLARIFY campaign and surface observations on Ascension Island.

The revised manuscript includes:

Figure S2 presents vertical distributions of CO (Fig. S2a) and BC (Fig. S2b) for the CLARIFY flights (C028 to C051). Concurrent surface observations of CO and BC on Ascension Island (Fig. S2) presented the same trend of MBL BB pollution as observed during the CLARIFY campaign (Zuidema et al., 2018).

Figure S2. a) Vertical distributions of CO (ppbv) from the CLARIFY flights (C028 to C051), alongside the estimated  $z_i$  from each flight (black circles). The right axis is the surface CO (red line) observed on Ascension Island (Zuidema et al., 2018). a) Vertical distributions of BC ( $\mu$ g m-3) from the CLARIFY flights (C028 to C051), alongside the estimated  $z_i$  from each flight (black circles). The right axis is the surface BC (red line) observed on Ascension Island (Zuidema et al., 2018).

**3. Relevant text statements for introducing each figure and table in the Supplementary document should be provided.**

Thanks to the reviewer's suggestion.

We have included text statements for introducing Figures/Table in the supplementary. In the revised supplementary, two sections are added: "S1 Observations of aerosol components and carbon monoxide" and "S2 NAME backward-dispersion fields".

**Minor comments:**

1. In the caption of Figure 8 and Figure S2, it should be clarified that Figure 8 is based on the output of 3D air parcel footprints whereas Figure S2 is based on the method introduced in the second paragraph in section 2.2.

We have added the related description to the captions.

**2. Line 47: to reduce the relative humidity**

Accepted.

**3. Better to also provide the flight numbers for flights with mainly limited cloud samplings that are not included in this study.**

The revised manuscript is:

Transit flights (C040-C041, predominately at high altitudes) and flights with mainly limited cloud samplings (C028, C030, C034, C035, and C043) are not included in this study.

**4. Line 128-129 (and Line 133): PCASP is missing here for Passive Cavity Aerosol Spectrometer Probes.**

Aerosol size distributions were measured at 1-Hz resolution via two wing-mounted Passive Cavity Aerosol Spectrometer

Probes (PCASP), which resolved number concentrations in 30 diameter bins between 0.1 and 3 μm.

**5. Line 145-146: the general size range of BBAs observed in this region should be provided here (or refer to the literature).**

The related description has been added to the manuscript.

These aerosol instruments covered the typical size range of BB aerosols in this region, primarily within the submicron range.

**6. Line 161: Why is 0.01 g m-3 used as the threshold LWC? Any rationale or reference?**

A reference of (Heymsfield and McFarquhar, 2001) has been added to the manuscript.

Heymsfield, A. J. and McFarquhar, G. M.: Microphysics of INDOEX clean and polluted trade wind cumulus clouds, J. Geophys. Res., 106, 28653–28673, 2001.

**7. Line 201: what is BL? Only MBL was defined before this.**

The definition of "BL" has been added.

Over the SEA, there is typically a strong thermodynamic capping inversion that inhibits turbulent mixing between the **boundary layer (BL)** and overlying FT air...

**8. Line 236: isn't case 1 at the middle of period 1?**

Case 1 released tracers in the middle of Period 1 (12:00 UTC, August 18).

**9. Figure 2: Why not choose case 2 and case 3 at a point where flight observations are available?**

As shown in the original Fig. 2 and the newly added Fig. S2, the MBL over Ascension Island became considerably clean from August 21 to 25 (Period 2), and became BB-polluted again from August 26 to September 5 (Period 3). Accordingly, we selected Cases 2 and 3 to release tracers near the start of Periods 2 (12:00 UTC, August 21) and 3 (12:00 UTC, August 26), respectively, aligning with these shifts in MBL pollution conditions.

The revised manuscript includes:

"Cases 2 and 3 released tracers near the start of Periods 2 (12:00 UTC, August 21) and 3 (12:00 UTC, August 26), respectively, coinciding with shifts in MBL pollution conditions observed over Ascension Island."

**10. Line 248-249: it might be too arbitrarily to use three cases to conclude the pollution source of the whole campaign. It is necessary to provide back trajectory results for each flight observation. Same as major comment 1.**

Based on the outputs of 3-hourly NAME back-dispersion fields, we calculated the time series of the fractional contributions of airmass originating from the African continent (20°S–5°N, 9°W–35°E) to the Ascension Island area. The resulting analysis provides a quantitative indication of BB plume influence on the Ascension Island area throughout the campaign period, as shown in Fig. 2b (FT simulations) and Fig. 2c (MBL simulations). Therefore, the conclusions regarding pollution sources are supported not only by the three representative examples of dispersion fields shown in the original Fig. S2, but also by the statistical analysis of the overall African source contributions presented in Fig. 2b (FT simulations) and Fig. 2c (MBL simulations).

11. Line 276-278: please refer to Figure 3a. Also, please check through the manuscript to make sure relevant figures/tables are correctly and properly referred to in the discussion. Some of in-text references are missing, which makes it a bit hassle to follow the flow.

The in-text references of figures/tables have been checked and referred to relevant discussions throughout the manuscript.

**12. Line 300: Also provide the value of the CCN activation fractions of these highly aged African BB aerosols**

The fraction value has been added.

Due to atmospheric aging, the CCN activation fractions of these highly aged African BB aerosols (CCN/Na =  $0.82 \pm 0.17$ , at SS = 0.2%) are generally higher than fresher BB aerosols sampled over the African continent (CCN/Na = 0.68, at SS = 0.3%)...

**13. Line 304: A sentence for clear definition of clean MBL is missing in the manuscript. It should be already introduced in Section 1 or Section 2**

The selection of clean BL air masses was added in the manuscript.

Clean BL air masses were selected when CO < 66 ppbv (53  $\mu$ g m-3), which corresponds to the lowest 5th percentile of all CO data collected in the MBL (Wu et al., 2020).

**14. Line 305: "The aerosols in the clean MBL were smaller", is the number or size of aerosols small?**

The aerosol sizes in the clean MBL were smaller than in the BB-polluted MBL during CLARIFY.

**15. Line 312-316: based on the results, the two individual statements seem right. However, it is awkward when putting them together**

The description of CCN/CN3 has been removed in the revised manuscript.

"However, average CCN/CN3 in the clean MBL (0.38 ± 0.18) was much smaller than in the BB impacted MBL (0.76 ± 0.10), as the dominant Aitken mode particles (

Figure 3. a, b) Average vertical profiles of a) liquid water potential temperature ( $\theta_l$ , K), b) total water mixing ratio ( $q_l$ ) for each flight used in this study. Blue, red and black lines represent measurements from Periods 1, 2 and 3, respectively. c-e) Summarized profiles of c) aerosol count median diameter derived from the PCASP (CMD,  $\mu$ m) and d,e) the ratio of CCN (~ 0.2%) to condensation nuclei ( $D_a > 3$  nm from the CPC) (CCN/CN3) and accumulation aerosol concentration ( $D_a > 0.1$   $\mu$ m from the PCASP) (CCN/Na) under polluted (black) and clean (red) conditions. Solid lines and shades represent median values and range from 10 to 90 percentiles. The red cross marker in Fig. 3c represents the average CMD in accumulation mode from measurements within the clean MBL. f-h) Vertical profiles of 1-hz, f) cloud droplet number concentration ( $N_d$ , # cm-3), g) liquid water content (LWC, g m-3) and h) cloud effective radius ( $R_e$ ,  $\mu$ m) in sampled continuous cloud layers. Red lines represent cloud measurements in the clean MBL, and grey lines represent cloud measurements in the BB-polluted MBL. It is noted that average vertical profiles of cloud properties from flight C032 are also provided in Figs. 3f-h (grey dashed lines). The y-axis uses a height scale normalized by inversion height ( $z_i$ ).

**17. Line 406: ambiguous statement. Enhanced Nd value or larger Nd distribution range?**

The revised manuscript is:

The enhanced values and broader range of sub- $N_a$  in the BB-polluted MBL led to substantially higher and more variable  $N_d$  values as compared to the clean MBL (Fig. 5a).

18. Line 406-407: Figure 5a shows what kind of effects? It should be clearly stated out what the figure/result show but not only say 'affect'.

The revised manuscript is:

The enhanced values and broader range of sub- $N_a$  in the BB-polluted MBL led to substantially higher and more variable  $N_d$  values as compared to the clean MBL (Fig. 5a), showing that **transported BB aerosols promote droplet activation and cloud** formation in this region.

**19. Figure 8: indicate the atmospheric condition and case number in the legend for each panel.**

The case numbers (Case 1, Case 2, and Case 3) and attribution column ranges (whole column, FT, and BL) of dispersion results have been added in Fig. 8.

20. Line 609-611: the continuous increase of CTH cannot explain why there is a decrease in Re at -24 hours. Further details/results should be provided to explain this?

The revised manuscript has explained that: "Concurrent MBL deepening and enhanced CTH could promote condensational growth, yielding larger droplets at the cloud top (Painemal et al., 2014). This agrees with the observed overall increase in  $R_e$  along the BL transport. However, as the MBL continues to deepen and the entrainment of dry FT air strengthens, LWC may decrease, explaining the subsequent decline in  $R_e$  observed from -24 h (Wood, 2012; Ryoo et al., 2022)."

**Reply to the Reviewer #3**

**General comments:**

The authors sufficiently addressed the referee's comments. I recommend publication after minor edits (see below) and additional proofreading.

**Minor edits:**

Line 36. Change "radiations" to singular.

Accepted

Line 39. Add period. Also, I do not see a paper by Wood in 2012 in your references. Please add it and make sure he is the sole author on that paper, otherwise an "et al." should be added to the citation.

The citation has been checked:

Wood, R.: Stratocumulus clouds, Mon. Weather Rev., 140, 2373–2423, https://doi.org/10.1175/MWR-D-11-00121.1, 2012.

Line 48. Specify what about the semi-direct effect is "negative".

The revised manuscript is:

Modeling studies suggest that BB aerosols exert an overall negative semi-direct radiative forcing (cooling) over the SEA during the fire season.

Line 185. "e.g." is more appropriate than "i.e." because you are providing an example of a study. Other uses of "i.e." in the manuscript also do not seem appropriate and should be checked.

The uses of "i.e." have been checked and replaced by "e.g." throughout the manuscript.

Line 225. "Respectively" should be deleted.

Accepted

Line 238. When using "respectively" in a sentence, a comma must be added before. Please fix this issue throughout the manuscript. Also, double check the abbreviation "Figs." rather than "Fig." is correct under ACP guidelines.

A comma has been added before "respectively" in related sentences.

The use of "Figure" and its abbreviation have been checked throughout the manuscript. They now follow the ACP guidelines, e.g. "Figure 1", "Fig. 1", "Figure 3f-h", "Figures 9c and 9d" and "Figs. 9c and 9d".

Lines 326 and 344. Change "samling" to "sampling".

Accepted

Line 346. Change "wiskers" to "whiskers".

Accepted

Line 464. "CTtoAB" is different that the variable on the x-axis in panel a.

The variable on the x-axis has been revised in Fig. 6a.

Figure 6. Relationships a) between sub- $N_a$  and the distance from the top of cloud layers in the MBL to the bottom of BB layers in the FT (Cloud Top to Aerosol Base, CTtoAB); b) between sub- $N_a$  and above- $N_a$ . The markers and error bars represent the average values and standard deviation for each profile. Black triangle and circle markers are from the BB-polluted MBL in Periods 1 and 3 respectively, and red circle markers are from the clean MBL in Period 2.

Line 573. Change "abudance" to abundance".

Accepted

Line 605. Change the word "Figures." to correct abbreviation.

Accepted

**References**

Heymsfield, A. J. and McFarquhar, G. M.: Microphysics of INDOEX clean and polluted trade wind cumulus clouds, J. Geophys. Res., 106, 28653–28673, 2001.

Painemal, D., Kato, S., and Minnis, P.: Boundary layer regulation in the southeast Atlantic cloud microphysics during the biomass burning season as seen by the A-train satellite constellation, J. Geophys. Res.-Atmos., 119, 11288–11302, https://doi.org/10.1002/2014JD022182, 2014.

Ryoo, J.-M., Pfister, L., Ueyama, R., Zuidema, P., Wood, R., Chang, I., and Redemann, J.: A meteorological overview of the ORACLES (ObseRvations of Aerosols above CLouds and their intEractionS) campaign over the southeastern Atlantic during 2016–2018: Part 2 – Daily and synoptic characteristics, Atmos. Chem. Phys., 22, 14209–14241, https://doi.org/10.5194/acp-22-14209-2022, 2022.

Wood, R.: Stratocumulus clouds, Mon. Weather Rev., 140, 2373–2423, https://doi.org/10.1175/MWR-D-11-00121.1, 2012. Wu, H., Taylor, J. W., Szpek, K., Langridge, J. M., Williams, P. I., Flynn, M., Allan, J. D., Abel, S. J., Pitt, J., Cotterell, M. I., Fox, C., Davies, N. W., Haywood, J., and Coe, H.: Vertical variability of the properties of highly aged biomass burning aerosol transported over the southeast Atlantic during CLARIFY-2017, Atmos. Chem. Phys., 20, 12697–12719, https://doi.org/10.5194/acp-20-12697-2020, 2020.

Zuidema, P., Sedlacek III, A. J., Flynn, C., Springston, S., Delgadillo, R., Zhang, J., Aiken, A. C., Koontz, A., and Muradyan, P.: The Ascension Island Boundary Layer in the Remote Southeast Atlantic is Often Smoky, Geophys. Res. Lett., 45, 4456–4465, https://doi.org/10.1002/2017GL076926, 2018.